# Disentanglement via Mechanism Sparsity Regularization: A New Principle for Nonlinear ICA

**Sébastien Lachapelle**                                                    SEBASTIEN.LACHAPELLE@UMONTREAL.CA
*Mila & DIRO, Université de Montréal*

**Pau Rodríguez López**
*ServiceNow Research*

**Yash Sharma**
*Tübingen AI Center, University of Tübingen*

**Katie Everett**
*Google Research*

**Rémi Le Priol**
*Mila & DIRO, Université de Montréal*

**Alexandre Lacoste**
*ServiceNow Research*

**Simon Lacoste-Julien**
*Mila & DIRO, Université de Montréal, Canada CIFAR AI Chair*

**Editors:** Bernhard Schölkopf, Caroline Uhler and Kun Zhang

## Abstract

This work introduces a novel principle we call *disentanglement via mechanism sparsity regularization*, which can be applied when the latent factors of interest depend sparsely on past latent factors and/or observed auxiliary variables. We propose a representation learning method that *induces* disentanglement by *simultaneously* learning the latent factors and the sparse causal graphical model that relates them. We develop a rigorous identifiability theory, building on recent nonlinear independent component analysis (ICA) results, that formalizes this principle and shows how the latent variables can be recovered up to permutation if one regularizes the latent mechanisms to be sparse and if some graph connectivity criterion is satisfied by the data generating process. As a special case of our framework, we show how one can leverage unknown-target interventions on the latent factors to disentangle them, thereby drawing further connections between ICA and causality. We propose a VAE-based method in which the latent mechanisms are learned and regularized via binary masks, and validate our theory by showing it learns disentangled representations in simulations.

**Keywords:** Causal representation learning, disentanglement, nonlinear ICA, causal discovery

## 1. Introduction

It has been proposed that causal reasoning will be central to move modern machine learning algorithms beyond their current shortcomings, such as their lack of *robustness*, *transferability* and *interpretability* (Pearl, 2019; Schölkopf, 2019; Schölkopf et al., 2021; Goyal and Bengio, 2021). However, it is still unclear how to reconcile the causal graphical model (CGM) formalism (Pearl, 2009; Peters et al., 2017), which operates on semantically meaningful high-level variables, with deep neural networks (Goodfellow et al., 2016), which excel on unstructured, low-level, high-dimensional

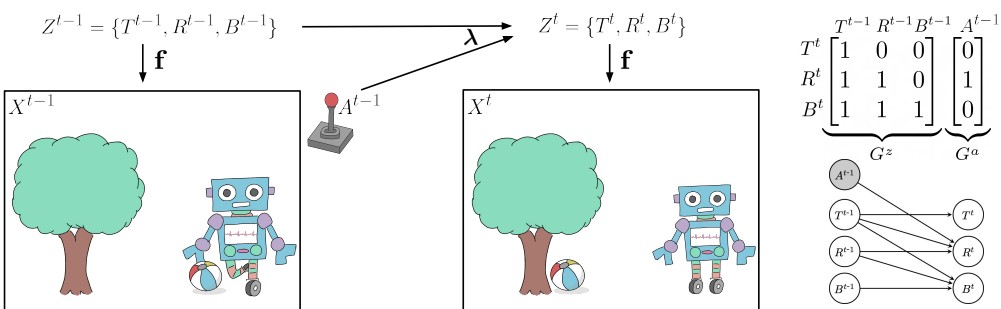

Figure 1: A minimal motivating example. The variables $T^t$, $R^t$ and $B^t$ represent the $x$-positions of the tree, the robot and the ball at time $t$, respectively. Only the image of the scene $X^t$ and the action $A^{t-1}$ are observed. See end of Sec. 2.1 for details.

observations, e.g. images. One way forward would be a two-step approach in which we first *disentangle* the high-level variables from low-level observations (Bengio et al., 2013; Locatello et al., 2019), then learn a CGM that relates them. Instead, this work proposes a method to do *both steps simultaneously*, and provides a rigorous theory that shows how doing so can *induce* disentanglement when the CGM is regularized to be sparse.

Our contribution is based on recent theoretical results in the nonlinear ICA literature (Hyvärinen et al., 2019; Khemakhem et al., 2020a,b) that assume the data is explained by *unobserved* and *meaningful* latent variables, or factors, $Z$, which are transformed by a *decoder*, or *mixing function*, $\mathbf{f}$, to produce the observation $X$. The problem of disentanglement can then be formulated as recovering, or reconstructing, the latent variables from the observation.

This problem is plagued by the difficult question of *identifiability*. Indeed, Hyvärinen and Pajunen (1999) showed that this task is impossible with general nonlinear mixing under the standard assumption of independent latent factors. Nevertheless, recent theoretical developments have shown identifiability of the latent factors is possible in the nonlinear setting, assuming the latent variables are *conditionally independent given an observed auxiliary variable $A$* (Hyvärinen et al., 2019; Khemakhem et al., 2020a,b). This auxiliary variable can be, for instance, a time or an environment index, an action in an interactive environment, or even a previous observation if the data has temporal structure, as long as its effect on the latent factors is "sufficiently strong".

The present paper introduces *mechanism sparsity regularization* as a new path to disentanglement. By building on the recent theoretical developments in ICA, we show that if the high-level variables have a *sparse* temporal structure and/or an action is observed and affects the high-level variables *sparsely*, then the latent variables can be recovered by regularizing the inferred graphical model to have sparse dependencies (Thm. 5). In estimating the latent variables, the presented methodology estimates the causal graph describing them and their relation to the action $A$ (when available). A very similar disentanglement method based on graph sparsity was proposed independently by Volodin (2021), but this concurrent work does not analyze identifiability formally (Sec. 3). In contrast, our theory provides precise conditions, e.g. on the ground-truth graph, to ensure identifiability, thus extending the domain of known cases where latent variables can be recovered.

The hypothesis that *high-level concepts can be described by a sparse dependency graph* has been described and leveraged for out-of-distribution generalization originally by Bengio (2019) and Goyal et al. (2021b), which were early sources of inspiration for this work. To the best of

our knowledge, our theory is the first to show formally that this inductive bias can sometimes be enough to recover the latent factors. As a special case, it also shows formally how *unknown-target interventions* on the latent factors can be leveraged to disentangle them (Sec. 2.5), which is closely related to the *sparse mechanism shift* hypothesis described by Schölkopf et al. (2021).

Fig. 1 shows a minimal motivating example in which our approach could be used to extract the high-level variables (such as the $x$-position of the three objects) and learn their dynamics (how the objects move and affect one another) from a time series of images, $X^t$, and agent actions, $A^t$. Thm. 5 shows how the sparse dependencies between the objects can be leveraged to estimate the latent variables as well as the graph describing their dynamics. The learned CGM could subsequently be used to simulate interventions on semantic variables (Pearl, 2009; Peters et al., 2017), such as changing the torque of the robot or the weight of the ball. Interventions allow an agent to imagine situations it has never seen before, which would not be possible without a disentangled representation (Schölkopf, 2019). Moreover, disentanglement could be useful for interpretability by allowing for the extraction of a causal graph of the agent actions (Pearl, 2019).

**Contributions:**
1. A new principle to achieve disentanglement based on *mechanism sparsity regularization* motivated by a rigorous and novel identifiability theory (Thm. 5).
2. A formal connection to the *sparse mechanism shift* hypothesis introduced by Schölkopf et al. (2021) via the notion of unknown-target interventions on the latent factors (Sec. 2.5).
3. An estimation procedure which relies on variational autoencoders (VAEs) (Kingma and Welling, 2014) and learned causal mechanisms regularized for sparsity via binary masks.
4. An illustration of our theoretical predictions being satisfied in practice by our estimation procedure on synthetic datasets.

The paper is structured as follows. Sec. 2.1 introduces the model under consideration. Sec. 2.2 defines the notions of linear and permutation equivalence between representations. Sec. 2.3 provides conditions to identify the model up to linear equivalence. Sec. 2.4 shows how mechanism sparsity regularization can induce permutation-identifiability, i.e. disentanglement. Sec. 2.6 proposes a VAE-based approach to model estimation, as well as a practical way to induce sparsity based on binary masks. Sec. 3 situates this paper in the related literature. Sec. 4 illustrates the proposed identifiability theory and learning methods on synthetic data.

## 2. Disentanglement via Mechanism Sparsity Regularization

### 2.1. An identifiable latent causal model

We now specify the setting under consideration. Assume we observe the realization of a sequence of $d_x$-dimensional random vectors $\{X^t\}_{t=1}^T$ and a sequence of $d_a$-dimensional auxiliary vectors $\{A^t\}_{t=0}^{T-1}$. The coordinates of $A^t$ are either discrete or continuous and can potentially represent, for example, an action taken by an agent, or the index of the environment the corresponding observation was taken from. Going forward, we will refer to $A^t$ as the action vector. The observations $\{X^t\}$ are assumed to be explained by a sequence of hidden $d_z$-dimensional continuous random vectors $\{Z^t\}_{t=1}^T$ via the equation $X^t = \mathbf{f}(Z^t) + N^t$ where $N^t \sim \mathcal{N}(0, \sigma^2 I)$ are mutually independent across time and independent of all $Z^t$ and $A^t$. Throughout, we assume $d_z \leq d_x$ and that $\mathbf{f} : \mathcal{Z} \to \mathcal{X}$ is a diffeomorphism[1] where $\mathcal{Z}$ is the support of $Z^t$ for all $t$, and $\mathcal{X} := \mathbf{f}(\mathcal{Z})$, i.e. the image of

---

1. A *diffeomorphism* is a differentiable bijection with a differentiable inverse.

$\mathcal{Z}$ under $\mathbf{f}$. App. A.5 discusses the implications of the diffeomorphism assumption. We suppose that each factor $Z_i^t$ contains semantic information about the observation, e.g. for high-dimensional images, the coordinates $Z_i^t$ might be the position of an object, its color, or its orientation in space. We denote $Z^{\leq t} := [Z^1 \cdots Z^t] \in \mathbb{R}^{d_z \times t}$ and analogously for $Z^{<t}$ and other random vectors.

Similar to previous work on nonlinear ICA (Hyvärinen et al., 2019; Khemakhem et al., 2020a), we assume the variables $Z_i^t$ are mutually independent given $Z^{<t}$ and $A^{<t}$

$$p(z^t \mid z^{<t}, a^{<t}) = \prod_{i=1}^{d_z} p(z_i^t \mid z^{<t}, a^{<t}) \,. \tag{1}$$

Our theory holds for a rich family of conditional densities $p(z_i^t \mid z^{<t}, a^{<t})$ called the *exponential family* (Wainwright and Jordan, 2008), which has the following form:

$$p(z_i^t \mid z^{<t}, a^{<t}) = h_i(z_i^t) \exp\{\mathbf{T}_i(z_i^t)^\top \boldsymbol{\lambda}_i(G_i^z \odot z^{<t}, G_i^a \odot a^{<t}) - \psi_i(z^{<t}, a^{<t})\} \,. \tag{2}$$

Well-known distributions which belong to this family include the Gaussian and beta distribution. In the Gaussian case, the *sufficient statistic* is $\mathbf{T}_i(z) := (z, z^2)$ and the *base measure* is $h_i(z) := \frac{1}{\sqrt{2\pi}}$. The function $\boldsymbol{\lambda}_i(G_i^z \odot z^{<t}, G_i^a \odot a^{<t})$ outputs the *natural parameter* vector for the conditional distribution and can be itself parametrized, for instance, by a multi-layer perceptron (MLP) or a recurrent neural network (RNN). We will refer to the functions $\boldsymbol{\lambda}_i$ as the *mechanisms* or the *transition functions*. In the Gaussian case, the natural parameter is two-dimensional and is related to the usual parameters $\mu$ and $\sigma^2$ via the equation $(\lambda_1, \lambda_2) = (\frac{\mu}{\sigma^2}, -\frac{1}{\sigma^2})$. We will denote by $k$ the dimensionality of the natural parameter and that of the sufficient statistic (which are equal). Thus, $k = 2$ in the Gaussian case. It will be useful to keep this example in mind throughout the paper. The remaining term $\psi_i(z^{<t}, a^{<t})$ acts as a normalization constant. The binary vectors $G_i^z \in \{0,1\}^{d_z}$ and $G_i^a \in \{0,1\}^{d_a}$ act as masks selecting the direct parents of $z_i^t$. The Hadamard product $\odot$ is applied element-wise and broadcasted along the time dimension.[2] We define

$$G^z := [G_1^z \cdots G_{d_z}^z]^\top \in \mathbb{R}^{d_z \times d_z} \,, \qquad G^a := [G_1^a \cdots G_{d_a}^a]^\top \in \mathbb{R}^{d_z \times d_a} \,, \tag{3}$$

as well as $G := [G^z \ G^a]$ which is the adjacency matrix of the causal graph.[3] Indeed, (1) & (2) describe a CGM over the unobserved variables $Z^{\leq T}$ conditioned on the auxiliary variables $A^{<T}$.

We define $\boldsymbol{\lambda}(z^{<t}, a^{<t}) \in \mathbb{R}^{k d_z}$ to be the concatenation of all $\boldsymbol{\lambda}_i(G_i^z \odot z^{<t}, G_i^a \odot a^{<t})$ and similarly for $\mathbf{T}(z^t) \in \mathbb{R}^{k d_z}$. Note that $\boldsymbol{\lambda}(z^{<t}, a^{<t})$ depends on $G$, implicitly to simplify the notation.

The learnable parameters are $\theta := (\mathbf{f}, \boldsymbol{\lambda}, G)$, which induce a conditional probability distribution $\mathbb{P}_{X^{\leq T} \mid a; \theta}$ over $X^{\leq T}$, given $A^{<T} = a$. Let $\mathcal{A} \subset \mathbb{R}^{d_a}$ be the set of possible values $A^t$ can take. We assume $p(a^{<T})$ has probability mass over all $\mathcal{A}^T$. This could arise, for instance, when $A^t$ is sampled from a policy $\pi(a^t \mid z^t)$ distribution with probability mass everywhere in $\mathcal{A}$.

**A motivating example.** Fig. 1 represents a minimal example where our theory applies. The environment consists of three objects: a tree, a robot and a ball with $x$-positions $T^t$, $R^t$ and $B^t$, respectively. Together, they form the vector $Z^t$ of high-level latent variables, i.e. $Z^t = (T^t, R^t, B^t)$.

---

2. This implies that if $z_i^t$ is connected to $z_j^{t-1}$, it is also connected to $z_j^{<t-1}$. Our theory may be generalizable to $G$ having different connectivity structure across time; but we keep this convention to simplify the notation.

3. Interpreting $G$ as "causal", meaning it can predict the effects of interventions, is natural in a temporal setting, since the future cannot affect the past. However, the following theory does not strictly require this causal interpretation.

A remote controls the direction in which the wheels of the robot turn. The vector $A^t$ records these actions, which might be taken by a human or an artificial agent trained to accomplish some goal. The only observations are the actions $A^t$ and the images $X^t$ representing the scene which is given by $X^t = \mathbf{f}(Z^t) + N^t$. The dynamics of the environment is governed by the transition function $\boldsymbol{\lambda}$. Assuming a Gaussian model with fixed variance for the latent factors $Z^{\leq T}$, $\boldsymbol{\lambda}$ would output the expected position of every object given their previous positions. Plausible connectivity graphs $G^z$ and $G^a$ are given in Fig. 1 showing how the latent factors are related, and how the controller affects them. For every object, its position at time step $t$ depends on its position at $t - 1$. The position of the tree, $T^t$, is not affected by anything, since neither the robot nor the ball can change its position. The robot, $R^t$, changes its position based on both the action, $A^{t-1}$ and the position of the tree, $T^{t-1}$ (in case of collision). The ball position, $B^t$, is affected by both the robot, which can kick it around by running into it, and the tree, on which it can bounce. The key observations here are that (i) the different objects interact *sparsely* with one another and (ii) the action $A^t$ affects very few objects (in this case, only one). Thm. 5 will show how one can leverage this sparsity for disentanglement.

## 2.2. Identifiability and model equivalence

To formalize the problem of disentanglement, we will rely on the notion of *identifiability*, which is a property a model has when its parameters can be uniquely determined by the distribution that it represents. Formally, given some distribution $\mathbb{P}_\theta$ parameterized by $\theta$, this means

$$\forall \theta, \tilde{\theta}, \; \mathbb{P}_\theta = \mathbb{P}_{\tilde{\theta}} \implies \theta = \tilde{\theta} \, . \tag{4}$$

For a model as flexible as the one described in the previous section, identifying the exact parameter $\theta$ is too strong a demand. Instead, we will be interested in identifying the parameter $\theta$ *up to an equivalence class*, which amounts to substituting some equivalence relation $\theta \sim \tilde{\theta}$ for $\theta = \tilde{\theta}$ in (4). We now present two equivalence relations for the model presented in Sec. 2.1 adapted from Khemakhem et al. (2020a): linear and permutation equivalence. The latter will help us formalize disentanglement. In what follows, we overload the notation by defining $\mathbf{f}^{-1}(z^{<t}) := [\mathbf{f}^{-1}(z^1) \; \cdots \; \mathbf{f}^{-1}(z^{t-1})]$.

**Definition 1 (Linear equivalence)**  *Let $\mathcal{X} := \mathbf{f}(\mathcal{Z})$ and $\tilde{\mathcal{X}} := \tilde{\mathbf{f}}(\mathcal{Z})$, i.e., the image of the support of $Z^t$ under $\mathbf{f}$ and $\tilde{\mathbf{f}}$, respectively. We say $\theta$ is **linearly equivalent** to $\tilde{\theta}$ if and only if $\mathcal{X} = \tilde{\mathcal{X}}$ and there exists an invertible matrix $L \in \mathbb{R}^{kd_z \times kd_z}$ as well as vectors $b, c \in \mathbb{R}^{kd_z}$ such that*

*1. $\mathbf{T}(\mathbf{f}^{-1}(x)) = L\mathbf{T}(\tilde{\mathbf{f}}^{-1}(x)) + b, \forall x \in \mathcal{X}$; and*

*2. $L^\top \boldsymbol{\lambda}(\mathbf{f}^{-1}(x^{<t}), a^{<t}) + c = \tilde{\boldsymbol{\lambda}}(\tilde{\mathbf{f}}^{-1}(x^{<t}), a^{<t}), \forall t \in \{1, ..., T\}, x^{<t} \in \mathcal{X}^{t-1}, a^{<t} \in \mathcal{A}^t.$*

*In this case, we write $\theta \sim_L \tilde{\theta}$.*

Hence, two models are linearly equivalent if they entail the same data manifold $\mathcal{X}$ and their respective representations $\mathbf{f}^{-1}(x)$ and $\tilde{\mathbf{f}}^{-1}(x)$ transformed through the element-wise sufficient statistic $\mathbf{T}$ are the same everywhere on $\mathcal{X}$ *up to an affine transformation.*

In the Gaussian case, with variance fixed to one, $\mathbf{T}(z) := z$ and $\boldsymbol{\lambda}$ outputs the usual mean parameter $\mu$ (here, $k = 1$). The first condition therefore means we can go from one representation to another via an affine transformation.

Suppose $\theta$ corresponds to the data generating process, while $\hat{\theta}$ is some learned model. Both being linearly equivalent is not enough to declare the learned representation disentangled, since the

matrix $L$ might still "mix up" the variables i.e. one component of $\mathbf{f}^{-1}$ corresponding to multiple components of $\hat{\mathbf{f}}^{-1}$. However, if $L$ happens to have a (block-)permutation structure, we have a one-to-one correspondence between the ground truth latent factors of the data and the coordinates of the learned representation.

**Definition 2 (Permutation equivalence)** *We say $\theta$ is **permutation-equivalent** to $\tilde{\theta}$ if and only if $\theta \sim_P \tilde{\theta}$ (Def. 1) where $P$ has a block-permutation structure respecting $\mathbf{T}$, i.e. there are $d_z$ invertible $k \times k$ matrices $L_1, ..., L_{d_z}$ and a $d_z$-permutation $\pi$ such that for all $y = [y_1 \dots y_{d_z}]^\top \in \mathbb{R}^{kd_z}$, $Py = [y_{\pi(1)} L_1^\top \dots y_{\pi(d_z)} L_{d_z}^\top]^\top$.*

In the Gaussian case with a fixed variance, permutation equivalence implies that each coordinate $i$ of one representation is equal to the scaled and shifted coordinate $\pi(i)$ of the other, for some permutation $\pi$. Inspired by previous works on nonlinear ICA, we define *disentanglement* as follows.

**Definition 3 (Disentanglement)** *Given a ground-truth model $\theta$, we say a learned model $\hat{\theta}$ is **disentangled** when $\theta$ and $\hat{\theta}$ are permutation-equivalent.*

### 2.3. Conditions for linear identifiability

From now on, it will be useful to think of $\theta$ as the *ground-truth parameter* and $\hat{\theta}$ as a *learned parameter*. The following theorem provides conditions that ensure *linear identifiability*, which is defined as

$$\mathbb{P}_{X^{\leq T}|a;\theta} = \mathbb{P}_{X^{\leq T}|a;\hat{\theta}} \ \forall a \in \mathcal{A}^T \implies \theta \sim_L \hat{\theta}, \tag{5}$$

where $L$ is some invertible matrix (not necessarily with a block-permutation structure). This theorem is an adaptation and minor extension of Thm. 1 from Khemakhem et al. (2020a), which we elaborate upon in Sec. 3, and is central to the stronger permutation-identifiability (disentanglement) theorems of the following section. A proof can be found in App. A.

**Theorem 4 (Conditions for linear identifiability - Extended from Khemakhem et al. (2020a))**
*Suppose we have two models as described in Sec. 2.1 with parameters $\theta = (\mathbf{f}, \boldsymbol{\lambda}, G)$ and $\hat{\theta} = (\hat{\mathbf{f}}, \hat{\boldsymbol{\lambda}}, \hat{G})$ for a fixed sequence length $T$. Suppose the following assumptions hold:*

1. *For all $i \in \{1, ..., d_z\}$, the sufficient statistic $\mathbf{T}_i$ is minimal (see next paragraph below).*

2. ***[Sufficient variability]** There exist $(z_{(p)}, a_{(p)})_{p=0}^{kd_z}$ in their respective supports such that the $kd_z$-dimensional vectors $(\boldsymbol{\lambda}(z_{(p)}, a_{(p)}) - \boldsymbol{\lambda}(z_{(0)}, a_{(0)}))_{p=1}^{kd_z}$ are linearly independent.*

*Then, we have linear identifiability: $\mathbb{P}_{X^{\leq T}|a;\theta} = \mathbb{P}_{X^{\leq T}|a;\hat{\theta}}$ for all $a \in \mathcal{A}^T$ implies $\theta \sim_L \hat{\theta}$.*

The first assumption is a standard one saying that $\mathbf{T}_i$ is defined appropriately to ensure that the parameters of the exponential family are identifiable (see e.g. Wainwright and Jordan (2008, p. 40)). See Def. 9 for a formal definition of minimality. The second assumption is sometimes called the *assumption of variability* (Hyvärinen et al., 2019), and requires that the conditional distribution of $Z^t$ depends "sufficiently strongly" on $Z^{<t}$ and/or $A^{<t}$. We stress the fact that this assumption concerns the ground-truth data generating model $\theta$. Notice that the $z_{(p)}$ represent values of $Z^{<t}$ for potentially different values of $t$ and can thus have different dimensions.

In the Gaussian case with variance fixed to one, the sufficient variability assumption requires that the values of the conditional mean $\mathbb{E}[Z^t|z^{<t}, a^{<t}]$ are not all contained in a *proper*[4] affine subspace of $\mathbb{R}^{d_z}$. This can be interpreted as having a *sufficiently complex* transition model.

### 2.4. Permutation-identifiability via mechanism sparsity regularization

We are now ready to present the core contribution of this work, i.e. a novel permutation-identifiability result based on mechanism sparsity regularization (Thm. 5). The intuition for this result is that, under appropriate assumptions (that are satisfied in the motivating example of Fig. 1), models that have an entangled representation also have a denser adjacency matrix $\hat{G}$. Thus, by regularizing $\hat{G}$ to be sparse, we exclude entangled models, leaving us with only the disentangled ones. Thm. 5 gives precise conditions about the data-generating model $\theta$ under which fitting the model $\hat{\theta}$ and regularizing the graph $\hat{G} = [\hat{G}^z \; \hat{G}^a]$ to be sparse will be sufficient to obtain a disentangled model (Def. 3). Recall that $\hat{G}^z$ controls the connectivity between the latent variables from one time step to another and that $\hat{G}^a$ controls the connectivity between the action $A^{<t}$ and the latent variable $Z^t$. Sec. 3 will contrast these results with those introduced in the recent literature on nonlinear ICA.

**Theorem 5 (Disentanglement via mechanism sparsity)** *Suppose we have two models as described in Sec. 2.1 with parameters $\theta = (\mathbf{f}, \boldsymbol{\lambda}, G)$ and $\hat{\theta} = (\hat{\mathbf{f}}, \hat{\boldsymbol{\lambda}}, \hat{G})$ for a fixed $T$ representing the same distribution, i.e. $\mathbb{P}_{X^{\leq T}|a;\theta} = \mathbb{P}_{X^{\leq T}|a;\hat{\theta}}$ for all $a \in \mathcal{A}^T$. Suppose the assumptions of Thm. 4 hold and:*

1. *The sufficient statistic $\mathbf{T}$ is $d_z$-dimensional ($k = 1$) and is a diffeomorphism from $\mathcal{Z}$ to $\mathbf{T}(\mathcal{Z})$.*

2. *[**Sufficient time-variability**] The Jacobian of the ground-truth transition function $\boldsymbol{\lambda}$ with respect to $z$ varies "sufficiently", as formalized in Assumption 1 in the next section.*

*Then, there exists a permutation matrix $P$ such that $PG^zP^\top \subset \hat{G}^z$.[5] Further assume that*

3. *[**Sufficient action-variability**] The ground-truth transition function $\boldsymbol{\lambda}$ is affected "sufficiently strongly" by each individual action $a_\ell$, as formalized in Assumption 2 in the next section.*

*Then $PG^a \subset \hat{G}^a$. Further assume that*

4. *[**Sparsity**] $||\hat{G}||_0 \leq ||G||_0$.*

*Then, $PG^zP^\top = \hat{G}^z$ and $PG^a = \hat{G}^a$. Further assume that*

5. *[**Graphical criterion**] For all $p \in \{1, ..., d_z\}$, there exist sets $\mathcal{I}, \mathcal{J} \subset \{1, ..., d_z\}$ and $\mathcal{L} \subset \{1, ..., d_a\}$ such that*

$$\left(\bigcap_{i \in \mathcal{I}} \mathbf{Pa}_i^z\right) \cap \left(\bigcap_{j \in \mathcal{J}} \mathbf{Ch}_j^z\right) \cap \left(\bigcap_{\ell \in \mathcal{L}} \mathbf{Ch}_\ell^a\right) = \{p\},$$

*where $\mathbf{Pa}_i^z$ and $\mathbf{Ch}_i^z$ are the sets of parents and children of node $z_i$ in $G^z$, respectively, while $\mathbf{Ch}_\ell^a$ is the set of children of $a_\ell$ in $G^a$.*

*Then $\theta$ and $\hat{\theta}$ are permutation-equivalent (Def. 2), i.e. the model $\hat{\theta}$ is disentangled.*

The first assumption is satisfied for example by the Gaussian case with variance fixed to one since $\mathbf{T}(z) = z$ is a diffeomorphism. In contrast, it is not satisfied in the Gaussian case with fixed mean since $\mathbf{T}(z) = z \odot z$ is not invertible.

---

4. A subset $A \subset B$ is *proper* when $A \neq B$.

5. Given two binary matrices $M^1$ and $M^2$ with equal shapes, we say $M^1 \subset M^2$ when $M_{i,j}^1 = 1 \implies M_{i,j}^2 = 1$.

**Sufficient variability.** Thm. 5 has also two sufficient variability assumptions, one for $G^z$ and one for $G^a$. Rigorous statements of these are delayed until Sec. 2.4.1. Intuitively, both assumptions require that the ground-truth transition function $\boldsymbol{\lambda}$ is complex enough.

**Sparsity.** The first three assumptions imply that the learned graph $\hat{G}$ is a supergraph of some permutation of the ground-truth graph $G$. By adding the *sparsity* assumption, we have that the learned graph $\hat{G}$ *is exactly* a permutation of the ground-truth graph $G$. This assumption is satisfied if $\hat{G}$ is a minimal graph among all graphs that allow the model to exactly match the ground-truth generative distribution. In Sec. 2.6, we suggest achieving this by regularizing $\hat{G}$ to be sparse.

**Graphical criterion.** The very last assumption is a *graphical criterion* that guarantees disentanglement. This criterion is trivially satisfied when $G^z$ is diagonal, since $\{i\} = \mathbf{Pa}_i^z$ for all $i$ (actions are not necessary here). This simple case amounts to having mutual independence between the sequences $Z_i^{\leq T}$, which is a standard assumption in the ICA literature (Tong et al., 1990; Hyvarinen and Morioka, 2017; Klindt et al., 2021). The illustrative example we introduced in Fig. 1 has a more interesting "non-diagonal" graph satisfying our criterion. Indeed, we have that $\{T\} = \mathbf{Pa}_T^z$, $\{R\} = \mathbf{Ch}_R^z \cap \mathbf{Pa}_R^z$ and $\{B\} = \mathbf{Ch}_B^z$. This example is actually part of an interesting family of graphs that satisfy our criterion:

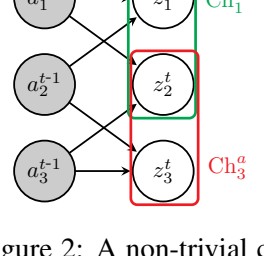

Figure 2: A non-trivial case where actions $a$ are enough to satisfy the graphical criterion of Thm. 5. Indeed, we have $\{z_1\} = \mathbf{Ch}_1^a \cap \mathbf{Ch}_2^a$, $\{z_2\} = \mathbf{Ch}_1^a \cap \mathbf{Ch}_3^a$ and $\{z_3\} = \mathbf{Ch}_2^a \cap \mathbf{Ch}_3^a$.

**Proposition 6 (Sufficient condition for the graphical criterion)** *If $G_{i,i}^z = 1$ for all $i$ (all nodes have a self-loop) and $G^z$ has no 2-cycles, then $G$ satisfies the graphical criterion of Thm. 5.*

**Proof** Self-loops guarantee $i \in \mathbf{Pa}_i^z \cap \mathbf{Ch}_i^z$ for all $i$. Suppose $j \in \mathbf{Pa}_i^z \cap \mathbf{Ch}_i^z$ for some $i$ and $j \neq i$. This implies $i$ and $j$ form a 2-cycle, which is a contradiction. Thus $\{i\} = \mathbf{Pa}_i^z \cap \mathbf{Ch}_i^z$ for all $i$. ∎

Prop. 6 implies that actions are not necessary for permutation-identifiability. Conversely, it is also possible to satisfy the graphical criterion without temporal dependencies, e.g., see Fig. 2, which highlights the fact that the condition of Prop. 6 is not necessary.

In App. A.3.2 & A.3.3, we present two theorems which are versions of Thm. 5 specialized to time-sparsity and action-sparsity, respectively. Note that they are not special cases of Thm. 5.

### 2.4.1. SUFFICIENT VARIABILITY ASSUMPTIONS

We now present the two technical variability assumptions of Thm. 5. Intuitively, both assumptions require that the data generating model has a "sufficiently complex" transition function $\boldsymbol{\lambda}$.

**Notation.** Let $\mathbb{R}_{G^z}^{d_z \times d_z}$ be the set of matrices $M \in \mathbb{R}^{d_z \times d_z}$ such that $M_{i,j} = 0$ whenever $G_{i,j}^z = 0$. Similarly, let $\mathbb{R}_{\mathbf{Ch}_\ell^a}^{d_z}$ be the subspace of $\mathbb{R}^{d_z}$ where all coordinates outside $\mathbf{Ch}_\ell^a$ are zero.

**Assumption 1 (Sufficient time-variability)** *There exist $\{(z_{(p)}, a_{(p)}, \tau_{(p)})\}_{p=1}^{||G^z||_0}$ belonging to their respective support such that*

$$\mathrm{span}\left\{ D_z^{\tau_{(p)}} \boldsymbol{\lambda}(z_{(p)}, a_{(p)}) D_z \mathbf{T}(z_{(p)}^{\tau_{(p)}})^{-1} \right\}_{p=1}^{||G^z||_0} = \mathbb{R}_{G^z}^{d_z \times d_z},$$

*where $D_z^{\tau_{(p)}} \boldsymbol{\lambda}$ and $D_z \mathbf{T}$ are Jacobians with respect to $z^{\tau_{(p)}}$ and $z$, respectively.*

Notice the Jacobian $D_z^{\tau(p)}\boldsymbol{\lambda}$ is always in $\mathbb{R}_{G^z}^{d_z \times d_z}$ because of how $G^z$ masks the input of $\boldsymbol{\lambda}_i$ in (2). The *sufficient time-variability* assumption further requires that the Jacobian varies "enough" so that it cannot be contained in a *proper* subspace of $\mathbb{R}_{G^z}^{d_z \times d_z}$. The following *sufficient action-variability* assumption has an analogous interpretation.

**Assumption 2 (Sufficient action-variability)** *For all $\ell \in \{1, ..., d_a\}$, there exist* $\{(z_{(p)}, a_{(p)}, \epsilon_{(p)}, \tau_{(p)})\}_{p=1}^{|\mathbf{Ch}_\ell^a|}$ *belonging to their respective support such that*

$$\text{span}\left\{\Delta_\ell^{\tau(p)}\boldsymbol{\lambda}(z_{(p)}, a_{(p)}, \epsilon_{(p)})\right\}_{p=1}^{|\mathbf{Ch}_\ell^a|} = \mathbb{R}_{\mathbf{Ch}_\ell^a}^{d_z},$$

*where $\Delta_\ell^\tau \boldsymbol{\lambda}(z^{<t}, a^{<t}, \epsilon)$ is a partial difference defined by*

$$\Delta_\ell^\tau \boldsymbol{\lambda}(z^{<t}, a^{<t}, \epsilon) := \boldsymbol{\lambda}(z^{<t}, a^{<t} + \epsilon E_{\ell,\tau}) - \boldsymbol{\lambda}(z^{<t}, a^{<t}), \tag{6}$$

*where $\epsilon \in \mathbb{R}$ and $E_{\ell,\tau} \in \mathbb{R}^{d_a \times t}$ is the one-hot matrix with the entry $(\ell, \tau)$ set to one. Thus, (6) is the discrete analog of a partial derivative w.r.t. $a_\ell^\tau$.*

In App. A.7, we provide a plausible transition function $\boldsymbol{\lambda}$ based on the illustrative example of Fig. 1 and show that it has sufficient variability. Given the complex interactions which abound in the real world, we conjecture that "realistic" transition functions are "complex enough" to satisfy both assumptions.

### 2.5. Actions as interventions with unknown targets and sparse mechanism shifts

An important special case of Thm. 5 is when $A^{t-1}$ corresponds to a one-hot vector indexing an *intervention with unknown targets* on the latent variables $Z^t$. This specific kind of intervention has been explored previously in the context of causal discovery where the intervention occurs on *observed* variables instead of *latent* variables like in our case (Eaton and Murphy, 2007; Mooij et al., 2020; Squires et al., 2020; Jaber et al., 2020; Brouillard et al., 2020; Ke et al., 2019). Specifically, assume $A^{t-1} \in \{\mathbf{0}, e_1, ..., e_{d_a}\}$, where each $e_\ell$ is a one-hot vector. The action $A^{t-1} = \mathbf{0}$ corresponds to the *observational setting*, i.e. when no intervention occurred, while $A^{t-1} = e_\ell$ corresponds to the $\ell$th intervention. In that context, the unknown graph $G^a$ describes which latents are targeted by the intervention, i.e. $G_{i,\ell}^a = 1$ if and only if $z_i$ is targeted by the $\ell$th intervention. Here, the partial difference $\Delta_\ell^{t-1}\boldsymbol{\lambda}(z^{<t}, \mathbf{0}, 1)$ measures the difference of natural parameters between the observational setting and the $\ell$th intervention.

In this context, the assumption that $G^a$ is sparse corresponds precisely to the *sparse mechanism shift* hypothesis from Schölkopf et al. (2021), i.e. that *only a few mechanisms change at a time*. Thm. 5 thus provides precise conditions for when sparse mechanism shifts induce disentanglement.

### 2.6. Regularized model estimation

In order to estimate from data the model presented in previous sections, we propose to use a maximum likelihood approach based on the well-known framework of variational autoencoders (VAEs) (Kingma and Welling, 2014) in which the decoder neural network corresponds to the mixing function $\mathbf{f}$. We consider an approximate posterior of the form

$$q(z^{\leq T} \mid x^{\leq T}, a^{<T}) := \prod_{t=1}^T q(z^t \mid x^t), \tag{7}$$

where $q(z^t \mid x^t)$ is a Gaussian distribution with mean and diagonal covariance outputted by a neural network $\texttt{encoder}(x^t)$. In our experiments, the transition functions $\boldsymbol{\lambda}_i$ are parameterized by fully connected neural networks that look only at a fixed window of $s$ lagged latent variables.[6] In all experiments, $\hat{p}(z_i^t \mid z^{<t}, a^{<t})$ is Gaussian with a learned variance that does not depend on $(z^{<t}, a^{<t})$ (see App. B.2 for details). This variational inference model induces the following evidence lower bound (ELBO) on $\log \hat{p}(x^{\leq T} | a^{<T})$:

$$\sum_{t=1}^{T} \mathop{\mathbb{E}}_{Z^t \sim q(\cdot | x^t)} [\log \hat{p}(x^t \mid Z^t)] - \mathop{\mathbb{E}}_{Z^{<t} \sim q(\cdot | x^{<t})} KL(q(Z^t \mid x^t) || \hat{p}(Z^t \mid Z^{<t}, a^{<t})). \tag{8}$$

We derive this fact in App. A.8. The learned distribution will exactly match the ground truth distribution if (i) the model has enough capacity to express the ground-truth generative process, (ii) the approximate posterior has enough capacity to express the ground-truth posterior $p(z^t | x^{\leq T}, a^{<T})$, (iii) the dataset is sufficiently large and (iv) the optimization finds the global optimum. If, in addition, the ground truth generative process satisfies the assumptions of Thm. 4, we can guarantee that the learned model $\hat{\theta}$ will be linearly equivalent to the ground truth model $\theta$.

To go from linear identifiability (Def. 1) to permutation-identifiability (Def. 2), Thm. 5 suggests we should not only fit the data, but also choose the model such that $\hat{G}$ is sparse (or minimal). To achieve this in practice, we add regularization terms $-\alpha_z ||\hat{G}^z||_0$ and $-\alpha_a ||\hat{G}^a||_0$ to the ELBO objective, where $\alpha_z \geq 0$ and $\alpha_a \geq 0$ are hyperparameters. To make the objective amenable to gradient-based optimization, we treat $\hat{G}_{i,j}^z$ and $\hat{G}_{i,\ell}^a$ as independent Bernoulli random variables with probabilities of success $\texttt{sigmoid}(\gamma_{i,j}^z)$ and $\texttt{sigmoid}(\gamma_{i,\ell}^a)$ and optimize the continuous parameters $\gamma^z$ and $\gamma^a$ using the Gumbel-Softmax gradient estimator (Jang et al., 2017; Maddison et al., 2017). This strategy has been used successfully in previous work to enable gradient-based causal discovery (Ng et al., 2019; Brouillard et al., 2020). A regularization that is too weak or too strong will result in graphs that are too dense or too sparse, respectively. In Sec. 4, we select $\alpha_z$ and $\alpha_a$ using an adaptation of the unsupervised model selection criterion proposed by Duan et al. (2020).

## 3. Related work

Recent theoretical results have shown that nonlinear ICA is possible when leveraging additional assumptions, e.g., nonstationarity (Hyvarinen and Morioka, 2016) and temporal dependencies (Hyvarinen and Morioka, 2017). Hyvärinen et al. (2019) generalized these works by introducing the notion of auxiliary variables (which correspond to $A$ in our work). All of these methods rely on noise contrastive estimation (NCE) (Gutmann and Hyvärinen, 2012), which underlies the state-of-the-art in self-supervised representation learning (Oord et al., 2018; Chen et al., 2020), in which identifiability has been used as an analysis tool (Roeder et al., 2021; Zimmermann et al., 2021; Von Kügelgen et al., 2021). Subsequent works have shown similar results using VAEs (Khemakhem et al., 2020a; Locatello et al., 2020; Klindt et al., 2021), normalizing flows (Sorrenson et al., 2020) and energy-based models (Khemakhem et al., 2020b).

Khemakhem et al. (2020a), which introduced iVAE, is likely the closest to the present work. Thm. 4 is quite similar to Thm. 1 from Khemakhem et al. (2020a), but iVAE's notion of linear equivalence is different in that it does not characterize the relationship between $\boldsymbol{\lambda}$ and $\hat{\boldsymbol{\lambda}}$, which

---

6. The theory we developed would allow for a $\boldsymbol{\lambda}$ function that depends on all previous time steps, not only the $s$ previous ones. This could be achieved with a recurrent neural network, but we leave this to future work.

is crucial for our proof of Thm. 5. The most significant distinction between the theory of (Khemakhem et al., 2020a) and ours is how *permutation-identifiability* is obtained: Thm. 2 & 3 from iVAE shows that if the assumptions of their Thm. 1 are satisfied and $\mathbf{T}_i$ has dimension $k > 1$ or is non-monotonic, then the model is not just linearly, but permutation-identifiable. In contrast, our theory covers the case where $k = 1$ and $\mathbf{T}_i$ is monotonic, like in the Gaussian case with fixed variance. Interestingly, Khemakhem et al. (2020a) mentioned this specific case as a counterexample to their theory in their Prop. 3. The extra power of our theory comes from the extra *structure* in the dependencies of the latent factors coupled with sparsity regularization. In App. A.6, we argue that the assumptions of iVAE for disentanglement are less plausible in an environment like the one depicted in Fig. 1, thus highlighting the importance of the case $k = 1$ with monotonic $\mathbf{T}_i$ of Thm. 5.

Similar to our theory, PCL (Hyvarinen and Morioka, 2017) and SlowVAE (Klindt et al., 2021) leverage temporal dependence, but always assume mutual independence of the sequences $\{Z_i^t\}$. In our notation, this amounts to assuming the graph $G^z$ is diagonal. Our theory allows for more flexibility by accounting for a variety of dependency structures, like a triangular graph $G^z$. However, we do not claim our theory is a strict generalization of these works, since, for instance, the latent Laplacian transition model assumed by SlowVAE is not in the exponential family.

Locatello et al. (2020) also leverages temporal dependence, but assumes each pair $(Z^t, Z^{t+1})$ shares a random subset $S$ of its components. Our theory allows for every latent factor to change constantly and, thus, does not make this assumption. Interestingly, they assume that, for all $i$, $P(S \cap S' = \{i\}) > 0$ (for i.i.d $S$ and $S'$), which resembles our graphical criterion.

While there is a significant amount of interest in learning probabilistic or causal graphs between high-level latent variables extracted from low-level observations (Bengio, 2019; Schölkopf, 2019; Schölkopf et al., 2021; Goyal and Bengio, 2021; Ke et al., 2021), there have been comparatively few practical solutions contributed to the literature. Of works which learn CGMs, a number assume the causal graph structure is known (Kocaoglu et al., 2018; Shen et al., 2021; Nair et al., 2019). The concurrent work of Volodin (2021) independently proposed a very similar approach to jointly disentangle the latent factors by learning a sparse causal graph relating them using binary masks, but focuses more on exploring various algorithm-specific decisions than on formal identifiability proofs and does not use a VAE-based approach to estimate their model. Bengio et al. (2020) suggests using adaptation speed as a heuristic objective to disentangle latent factors and their causal relationship in the bivariate case. Yang et al. (2021) learns the causal graph by incorporating a "causal model layer" into iVAE, but does not rely on mechanism sparsity to disentangle, does not apply to time-series and is limited to linear CGMs. The assumption that high-level variables are sparsely related to one another has been leveraged also by Goyal et al. (2021b,a); Madan et al. (2021) via attention mechanisms. Although these works are, in part, motivated by the same core assumption as ours, their focus is more on empirically verifying out-of-distribution generalization than it is on disentanglement (Def. 3) and formal identifiability theory.

The assumption that individual actions often affect only one factor of variation has been leveraged for disentanglement by Thomas et al. (2017). Loosely speaking, the theory we developed in the present work can be seen as a formal justification for such an approach.

## 4. Experiments

To illustrate Thm. 5 and the benefit of mechanism sparsity regularization for disentanglement, we apply the regularized VAE method of Section 2.6 on synthetic datasets that both satisfy (Fig. 3) and

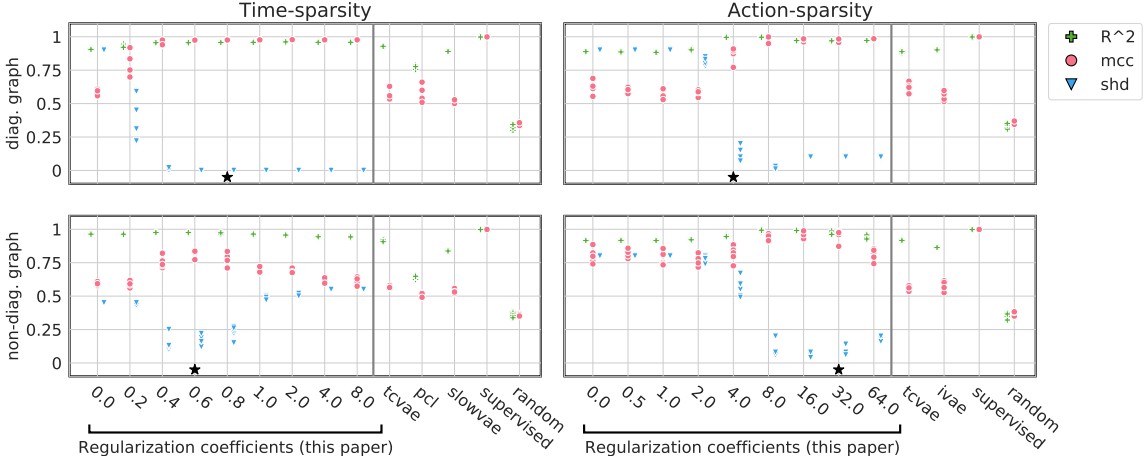

Figure 3: Top row datasets have a diagonal graph and bottom row datasets have a non-diagonal graph satisfying the graphical criterion of Thm. 5. Sufficient variability is always satisfied. In the left column, only $\hat{G}^z$ is learned and we vary $\alpha_z$, and in the right column, only $\hat{G}^a$ is learned and we vary $\alpha_a$. For more details on the synthetic datasets, see App. B.1. The black star indicates which regularization parameter is selected by the filtered UDR procedure (see App. B.7). For $R^2$ and MCC, higher is better. For SHD, lower is better. Performance is reported on 5 random seeds.

violate (Fig. 8, in App. B.4) the assumptions of Thm. 5. Details about the implementation of our approach are provided in App. B.2 and the code used to run these experiments can be found here: https://github.com/slachapelle/disentanglement_via_mechanism_sparsity.

**Synthetic datasets.** The datasets we considered are separated in two groups: *time-sparsity* and *action-sparsity* datasets. The former group has only temporal dependence without actions, we thus fix $\hat{G}^a = \mathbf{0}$, while the latter has only actions without temporal dependence, we thus fix $\hat{G}^z = \mathbf{0}$. In each dataset, the ground-truth mixing function $\mathbf{f}$ is a randomly initialized neural network. The dimensionality of $Z$ and $X$ are $d_z = 10$ and $d_x = 20$, respectively. In the action-sparsity datasets, the dimensionality of $A$ is $d_a = 10$. The ground-truth transition model $p(z^t \mid z^{<t}, a^{<t})$ is always a Gaussian with covariance $\sigma_z^2 I$ and a mean outputted by some function $\mu_G(z^{t-1}, a^{t-1})$ (the data is *Markovian*). Hence, each dataset has a 1d sufficient statistic ($k = 1$) that is also monotonic and, thus, is not covered by the theory of Khemakhem et al. (2020a). App. B.1 provides a more detailed descriptions of the datasets including the explicit form of $\mu$ and $G$ in each case. Note that the learned transition model $\hat{p}(z^t \mid z^{t-1}, a^{t-1})$ is also Gaussian where the mean is outputted by a MLP.

**Performance metrics.** To evaluate disentanglement, we will use encoder($x$) as a proxy for the learned $\hat{\mathbf{f}}^{-1}(x)$. To assess *linear identifiability*, we perform linear regression to predict the ground-truth latent factors from the inferred ones, and report the *coefficient of determination* $R^2$. To assess *permutation-identifiability*, i.e. disentanglement, we report the *mean correlation coefficient* (MCC), which has been used in similar contexts, e.g. Khemakhem et al. (2020a). This metric is obtained by first computing the Pearson correlation matrix $C \in \mathbb{R}^{d_z \times d_z}$ between the learned representation and the ground truth latent variables. Then, MCC $= \max_{\pi \in \text{permutations}} \frac{1}{d_z} \sum_{i=1}^{d_z} |C_{i,\pi(i)}|$. For our method, which is the only one learning a graph, we also report the *structural hamming distance* (SHD) between the ground-truth graph and the learned graph permuted by $\pi^*$, the optimal permutation found when computing MCC. We normalize SHD by the maximal number of edges

to ensure it is always between 0 and 1. The normalized SHD is thus the proportion of incorrectly estimated edges in the graph.

**Baselines.** On the temporal-sparsity datasets, we compare our approach with TCVAE (Chen et al., 2018), PCL (Hyvarinen and Morioka, 2017) and SlowVAE (Klindt et al., 2021). On the action-sparsity datasets, we compare with TCVAE and iVAE (Khemakhem et al., 2020a). We also report the performance of a randomly initialized encoder (Random) and one trained via least-square regression directly on the ground-truth latent factors (Supervised). See App. B.6 for details.

**Unsupervised hyperparameter selection.** The hyperparameters of the baselines were selected via *unsupervised disentanglement ranking* (UDR) (Duan et al., 2020). For our approach, Fig. 3 & 8 show performance for a range of regularization coefficients $\alpha_z$ and $\alpha_a$. We suggest selecting it using UDR and excluding coefficients that yield graphs with less than $d_z = 10$ edges, as the graphical criterion cannot be achieved in that case. Fig. 3 & 8 show this *unsupervised* procedure selects a reasonable regularization coefficient (as indicated by the black star). See App. B.7 for details.

**Effect of regularization.** Fig. 3 shows that the right amount of mechanism sparsity regularization leads to improved disentanglement (as measured by MCC), which is in line with Thm. 5. Improvements in SHD indicate that regularization allows for estimation of the causal graph $G$ (see App. B.5 for visualizations). When $\alpha_z$ and $\alpha_a$ are selected with the filtered UDR procedure, our approach outperforms the baselines by a significant margin, while without regularization, it performs similarly. Most baselines obtain a high $R^2$ but a low MCC, indicating linear identifiability without disentanglement. These observations are consistent across all four datasets. Similar observations also hold for randomly sampled graphs (Fig. 5), different noise-levels on the latent variables (Fig. 6) and different noise-levels on the observations (Fig. 7). See App. B.3 for details and caveats.

**Violating assumptions.** In App. B.4, Fig. 8 shows experiments on data violating either the sufficient variability assumption or the graphical criterion. Additionally, Fig. 9 shows data with a sufficient statistic $\mathbf{T}_i$ of dimension $k = 2$, thus violating the first assumption of Thm. 5. On all these datasets, except the time-sparsity data with insufficient variability, regularization improved MCC, although by a smaller margin than when assumptions are met. This suggests some of these assumptions might be relaxed.

## 5. Conclusion

This work proposed a novel principle for disentanglement based on *mechanism sparsity regularization*. The idea is based on the assumption that the mechanisms that govern the dynamics of high-level concepts are often sparse: objects usually interact sparsely with each other and actions usually affect only a few entities. Building on recent developments in nonlinear ICA, we constructed a rigorous theory which provides precise conditions, e.g. on the structure of the ground-truth dependency graph, for when regularizing the mechanisms to be sparse will result in disentanglement. A special case of our framework shows how one can leverage *unknown-target interventions* on the latent factors, or *sparse mechanism shifts*, for disentanglement. We proposed a regularized VAE-based approach and demonstrated that it can improve disentanglement in controlled synthetic settings, thereby preparing the stage for more realistic scenarios, e.g. interactive environments. We believe this work opens up new possibilities at the intersection of causality and disentanglement that leverage *structural assumptions*. For instance, we posit that contextual sparsity, i.e., the assumption that objects only interact with each other in particular situations, could be formalized and leveraged for disentanglement using the tools developed in this work.

## Acknowledgments

This research was partially supported by the Canada CIFAR AI Chair Program, by an IVADO excellence PhD scholarship, by a Google Focused Research award, the German Federal Ministry of Education and Research (BMBF): Tübingen AI Center, FKZ: 01IS18039A, and by Mitacs through the Mitacs Accelerate program. The experiments were in part enabled by computational resources provided by Calcul Quebec and Compute Canada. The authors would like to thank Yoshua Bengio for inspiring mechanism sparsity regularization through various talks and discussions. The authors would also like to thank the International Max Planck Research School for Intelligent Systems (IMPRS-IS) for supporting Yash Sharma. Simon Lacoste-Julien is a CIFAR Associate Fellow in the Learning in Machines & Brains program.

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

## Contents

## Appendix A. Theory

### A.1. Technical Lemmas and definitions

In this section, we prove a technical lemma which will be useful for central results of this work. However, this section can be safely skipped at first read.

**Definition 7** *(Support of a random variable) Let $X$ be a random variable with values in $\mathbb{R}^n$ with measure $\mathbb{P}_X$. Let $\mathcal{O}_n$ be the standard topology of $\mathbb{R}^n$ (i.e. the set of open sets of $\mathbb{R}^n$). The support of $X$ is defined as*

$$supp(X) := \left\{ x \in \mathbb{R}^n \mid x \in O \in \mathcal{O}_n \implies \mathbb{P}_X(O) > 0 \right\}. \tag{9}$$

**Lemma 8** *Let $Z$ be a random variable with values in $\mathbb{R}^m$ with distribution $\mathbb{P}_Z$ and $Y = f(Z)$ where $f : supp(Z) \subset \mathbb{R}^m \to \mathcal{Y} \subset \mathbb{R}^n$ is a homeomorphism. Then*

$$f(supp(Z)) = supp(Y). \tag{10}$$

*Proof.* We first prove that $f(\text{supp}(Z)) \subset \text{supp}(Y)$. Let $y \in f(\text{supp}(Z)) \subset \mathcal{Y}$ and $N$ be an open neighborhood of $y$, i.e. $y \in N \in \mathcal{O}_m$. Note that there exists $z \in \text{supp}(Z)$ such that $f(z) = y$. Note that $z \in f^{-1}(\{y\}) \subset f^{-1}(N \cap \mathcal{Y})$ and that, by continuity of $f$, $f^{-1}(N \cap \mathcal{Y})$ is an open neighborhood of $z$. Since $z \in \text{supp}(Z)$, we have

$$0 < \mathbb{P}_Z(f^{-1}(N \cap \mathcal{Y})) \tag{11}$$
$$= \mathbb{P}_Z \circ f^{-1}(N \cap \mathcal{Y}) \tag{12}$$
$$= \mathbb{P}_Y(N \cap \mathcal{Y}) \tag{13}$$
$$\leq \mathbb{P}_Y(N). \tag{14}$$

Hence $y \in \text{supp}(Y)$, which concludes the "$\subset$" part.

To prove the other direction, we notice that $Z = f^{-1}(Y)$ with $f^{-1}$ continuous. We can thus apply the same argument as above to show $f^{-1}(\text{supp}(Y)) \subset \text{supp}(Z)$, which implies that $\text{supp}(Y) \subset f(\text{supp}(Z))$. ∎

We recall the definition of a minimal sufficient statistic in an exponential family, which can be found in Wainwright and Jordan (2008, p. 40).

**Definition 9 (Minimal sufficient statistic)** *Given a parameterized distribution in the exponential family, as in (2), we say its sufficient statistic $\mathbf{T}_i$ is minimal when there is no $v \neq 0$ such that $v^\top \mathbf{T}_i(z)$ is constant for all $z \in \mathcal{Z}$.*

The following Lemma gives a characterization of minimality which will be useful in the proof of Thm. 4.

**Lemma 10 (Characterization of minimal T)** *A sufficient statistic $\mathbf{T} : \mathcal{Z} \to \mathbb{R}^k$ is minimal if and only if there exists $z_{(0)}, z_{(1)}, ..., z_{(k)}$ belonging to the support $\mathcal{Z}$ such that the following $k$-dimensional vectors are linearly independent:*

$$\mathbf{T}(z_{(1)}) - \mathbf{T}(z_{(0)}), ..., \mathbf{T}(z_{(k)}) - \mathbf{T}(z_{(0)}). \tag{15}$$

*Proof.* We start by showing the "if" part of the statement. Suppose there exist $z_{(0)}, ..., z_{(k)}$ in $\mathcal{Z}$ such that the vectors of (15) are linearly independent. By contradiction, suppose that $\mathbf{T}$ is not minimal, i.e. there exist a nonzero vector $v$ and a scalar $b$ such that $v^\top \mathbf{T}(z) = b$ for all $z \in \mathcal{Z}$. Notice that $b = v^\top \mathbf{T}(z_{(0)})$. Hence, $v^\top (\mathbf{T}(z_{(i)}) - \mathbf{T}(z_{(0)})) = 0$ for all $i = 1, ..., k$. This can be rewritten in matrix form as

$$v^\top [\mathbf{T}(z_{(1)}) - \mathbf{T}(z_{(0)}) \, ... \, \mathbf{T}(z_{(k)}) - \mathbf{T}(z_{(0)})] = 0 \,, \tag{16}$$

which implies that the matrix in the above equation is not invertible. This is a contradiction.

We now show the "only if" part of the statement. Suppose that there is no $z_{(0)}, ..., z_{(k)}$ such that the vectors of (15) are linearly independent. Choose an arbitrary $z_{(0)} \in \mathcal{Z}$. We thus have that $U := \text{span}\{\mathbf{T}(z) - \mathbf{T}(z_{(0)}) \mid z \in \mathcal{Z}\}$ is a proper subspace of $\mathbb{R}^k$. This means the orthogonal complement of $U$, $U^\perp$, has dimension 1 or greater. We can thus pick a nonzero vector $v \in U^\perp$ such that $v^\top (\mathbf{T}(z) - \mathbf{T}(z_0)) = 0$ for all $z \in \mathcal{Z}$, which is to say that $v^\top \mathbf{T}(z)$ is constant for all $z \in \mathcal{Z}$, and thus, $\mathbf{T}$ is not minimal. ∎

### A.2. Proof of linear identifiability (Thm. 4)

The following theorem and its proof are a minor extension of that of Khemakhem et al. (2020a). The key differences are (i) the fact that the sufficient statistics $\mathbf{T}_i$ do not have to be differentiable, which allows us to cover discrete latent variables (even though this is not highlighted in the main text), (ii) the notion of linear equivalence does not say anything about the link between $\boldsymbol{\lambda}$ and $\hat{\boldsymbol{\lambda}}$, which is crucial for the proof of Thm. 5, and (iii) allowing $\boldsymbol{\lambda}$ to depend on $z^{<t}$. Strictly speaking, point (iii) was not covered by previous nonlinear ICA frameworks since $z^{<t}$ is not observed and, thus, cannot be treated as an auxiliary variable (which must be observed).

**Theorem 4 (Conditions for linear identifiability - Extended from Khemakhem et al. (2020a))**
*Suppose we have two models as described in Sec. 2.1 with parameters $\theta = (\mathbf{f}, \boldsymbol{\lambda}, G)$ and $\hat{\theta} = (\hat{\mathbf{f}}, \hat{\boldsymbol{\lambda}}, \hat{G})$ for a fixed sequence length $T$. Suppose the following assumptions hold:*

1. *For all $i \in \{1, ..., d_z\}$, the sufficient statistic $\mathbf{T}_i$ is minimal (Def. 9).*

2. *[Sufficient variability] There exist $(z_{(p)}, a_{(p)})_{p=0}^{kd_z}$ in their respective supports such that the $kd_z$-dimensional vectors $(\boldsymbol{\lambda}(z_{(p)}, a_{(p)}) - \boldsymbol{\lambda}(z_{(0)}, a_{(0)}))_{p=1}^{kd_z}$ are linearly independent.*

*Then, we have linear identifiability: $\mathbb{P}_{X^{\leq T}|a;\theta} = \mathbb{P}_{X^{\leq T}|a;\hat{\theta}}$ for all $a \in \mathcal{A}^T$ implies $\theta \sim_L \hat{\theta}$.*

*Proof.*

**Equality of Denoised Distributions.** Define $Y^t := \mathbf{f}(Z^t)$. Given an arbitrary $a \in \mathcal{A}^T$ and a parameter $\theta = (\mathbf{f}, \boldsymbol{\lambda})$, let $\mathbb{P}_{Y^{\leq T}|a;\theta}$ be the conditional probability distribution of $Y^{\leq T}$, let $\mathbb{P}_{Z^{\leq T}|a;\theta}$ be the conditional probability distribution of $Z^{\leq T}$ and let $\mathbb{P}_{N^{\leq T}}$ be the probability distribution of $N^{\leq T}$ (the Gaussian noises added on $\mathbf{f}(Z^{\leq T})$, defined in Sec. 2.1). First, notice that

$$\mathbb{P}_{X^{\leq T}|a;\theta} = \mathbb{P}_{Y^{\leq T}|a;\theta} * \mathbb{P}_{N^{\leq T}} \,, \tag{17}$$

where $*$ is the convolution operator between two measures. We now show that if two models agree on the observations, i.e. $\mathbb{P}_{X^{\leq T}|a;\theta} = \mathbb{P}_{X^{\leq T}|a;\hat{\theta}}$, then $\mathbb{P}_{Y^{\leq T}|a;\theta} = \mathbb{P}_{Y^{\leq T}|a;\hat{\theta}}$. The argument makes use

of the Fourier transform $\mathcal{F}$ generalized to arbitrary probability measures. This tool is necessary to deal with measures which do not have a density w.r.t either the Lebesgue or the counting measure, as is the case of $\mathbb{P}_{Y^{\leq T}|a;\theta}$ (all its mass is concentrated on the set $\mathbf{f}(\mathbb{R}^{d_z})$). See Pollard (2001, Chapter 8) for an introduction and useful properties.

$$\mathbb{P}_{X^{\leq T}|a;\theta} = \mathbb{P}_{X^{\leq T}|a;\hat{\theta}} \tag{18}$$

$$\mathbb{P}_{Y^{\leq T}|a;\theta} * \mathbb{P}_{N^{\leq T}} = \mathbb{P}_{Y^{\leq T}|a;\hat{\theta}} * \mathbb{P}_{N^{\leq T}} \tag{19}$$

$$\mathcal{F}(\mathbb{P}_{Y^{\leq T}|a;\theta} * \mathbb{P}_{N^{\leq T}}) = \mathcal{F}(\mathbb{P}_{Y^{\leq T}|a;\hat{\theta}} * \mathbb{P}_{N^{\leq T}}) \tag{20}$$

$$\mathcal{F}(\mathbb{P}_{Y^{\leq T}|a;\theta})\mathcal{F}(\mathbb{P}_{N^{\leq T}}) = \mathcal{F}(\mathbb{P}_{Y^{\leq T}|a;\hat{\theta}})\mathcal{F}(\mathbb{P}_{N^{\leq T}}) \tag{21}$$

$$\mathcal{F}(\mathbb{P}_{Y^{\leq T}|a;\theta}) = \mathcal{F}(\mathbb{P}_{Y^{\leq T}|a;\hat{\theta}}) \tag{22}$$

$$\mathbb{P}_{Y^{\leq T}|a;\theta} = \mathbb{P}_{Y^{\leq T}|a;\hat{\theta}}\,, \tag{23}$$

where (20) & (23) use the fact the Fourier transform is invertible, (21) is an application of the fact that the Fourier transform of a convolution is the product of their Fourier transforms and (22) holds because the Fourier transform of a Normal distribution is nonzero everywhere. Note that the latter argument holds because we assume $\sigma^2$, the variance of the Gaussian noise added to $Y^t$, is the same for both models. For an argument that takes into account the fact that $\sigma^2$ is learned and assumes $d_z < d_x$, see Appendix A.4.1. We can further derive that, by Lemma 8, we have that

$$\mathbf{f}(\mathcal{Z}^T) = \mathrm{supp}(\mathbb{P}_{Y^{\leq T}|a;\theta}) = \mathrm{supp}(\mathbb{P}_{Y^{\leq T}|a;\hat{\theta}}) = \hat{\mathbf{f}}(\mathcal{Z}^T)\,, \tag{24}$$

where we overloaded the notation by defining $\mathbf{f}(z^{\leq T}) := (\mathbf{f}(z^1), ..., \mathbf{f}(z^T))$ and analogously for $\hat{\mathbf{f}}(z^{\leq T})$. Equation (24) shows that that the data manifolds are the same for both models, which is part of the linear equivalence definition we want to show (Def. 1).

**Equality of densities.** Continuing with (23),

$$\mathbb{P}_{Y^{\leq T}|a;\theta} = \mathbb{P}_{Y^{\leq T}|a;\hat{\theta}} \tag{25}$$

$$\mathbb{P}_{Z^{\leq T}|a;\theta} \circ \mathbf{f}^{-1} = \mathbb{P}_{Z^{\leq T}|a;\hat{\theta}} \circ \hat{\mathbf{f}}^{-1} \tag{26}$$

$$\mathbb{P}_{Z^{\leq T}|a;\theta} = \mathbb{P}_{Z^{\leq T}|a;\hat{\theta}} \circ \hat{\mathbf{f}}^{-1} \circ \mathbf{f} \tag{27}$$

$$\mathbb{P}_{Z^{\leq T}|a;\theta} = \mathbb{P}_{Z^{\leq T}|a;\hat{\theta}} \circ \mathbf{v}\,, \tag{28}$$

where $\mathbf{v} := \hat{\mathbf{f}}^{-1} \circ \mathbf{f}$, with $\mathbf{v} : \mathcal{Z} \to \hat{\mathcal{Z}}$. Note that this composition is well defined because $\mathbf{f}(\mathcal{Z}) = \hat{\mathbf{f}}(\hat{\mathcal{Z}})$. We chose to work directly with measures (functions on sets), as opposed to manifold integrals in (Khemakhem et al., 2020a), because it simplifies the derivation of (28) and avoids having to define densities w.r.t. measures concentrated on a manifold. We now derive the density of $\mathbb{P}_{Z^{\leq T}|a;\hat{\theta}} \circ \mathbf{v}$

w.r.t. to the Lebesgue measure $m$. Let $E \subset \mathcal{Z}^T$ be an event, we then have

$$\mathbb{P}_{Z^{\leq T}|a;\hat{\theta}} \circ \mathbf{v}(E)$$

$$= \int_{\mathbf{v}(E)} d\mathbb{P}_{Z^{\leq T}|a;\hat{\theta}} \tag{29}$$

$$= \int_{\mathbf{v}(E)} \prod_{t=1}^{T} \hat{p}(z^t \mid z^{<t}, a^{<t}) dm(z^{\leq T}) \tag{30}$$

$$= \int_{E} \prod_{t=1}^{T} \left[ \hat{p}(\mathbf{v}(z^t) \mid \mathbf{v}(z^{<t}), a^{<t}) \right] |\det D\mathbf{v}(z^{\leq T})| dm(z^{\leq T}) \tag{31}$$

$$= \int_{E} \prod_{t=1}^{T} \left[ \hat{p}(\mathbf{v}(z^t) \mid \mathbf{v}(z^{<t}), a^{<t}) |\det D\mathbf{v}(z^t)| \right] dm(z^{\leq T}), \tag{32}$$

where $D\mathbf{v}$ is the Jacobian matrix of $\mathbf{v}$ and $\hat{p}$ refers to the conditional density of the model with parameter $\hat{\theta}$. If $Z^t$ are discrete random variables, we can do the same except replacing $m$ by the counting measure and forgetting about the Jacobian of $\mathbf{v}$. We will present the rest of the argument in the case where $Z^t$ is continuous. The reader should keep in mind that the discrete case is exactly the same, except without the Jacobian of $\mathbf{v}$ appearing. Since $\mathbb{P}_{Z^{\leq T}|a;\theta}$ and $\mathbb{P}_{Z^{\leq T}|a;\hat{\theta}} \circ \mathbf{v}$ are equal, they must have the same density:

$$\prod_{t=1}^{T} p(z^t \mid z^{<t}, a^{<t}) = \prod_{t=1}^{T} \hat{p}(\mathbf{v}(z^t) \mid \mathbf{v}(z^{<t}), a^{<t}) |\det D\mathbf{v}(z^t)|, \tag{33}$$

where $p$ refers to the conditional density of the model with parameter $\theta$. For a given $t_0$, we have

$$\prod_{t=1}^{t_0} p(z^t \mid z^{<t}, a^{<t}) = \prod_{t=1}^{t_0} \hat{p}(\mathbf{v}(z^t) \mid \mathbf{v}(z^{<t}), a^{<t}) |\det D\mathbf{v}(z^t)|, \tag{34}$$

by integrating first $z^T$, then $z^{t-1}$, then ..., up to $z^{t_0+1}$. Note that we can integrate $z^{t_0}$ and get

$$\prod_{t=1}^{t_0-1} p(z^t \mid z^{<t}, a^{<t}) = \prod_{t=1}^{t_0-1} \hat{p}(\mathbf{v}(z^t) \mid \mathbf{v}(z^{<t}), a^{<t}) |\det D\mathbf{v}(z^t)|. \tag{35}$$

By dividing (34) by (35), we get

$$p(z^{t_0} \mid z^{<t_0}, a^{<t_0}) = \hat{p}(\mathbf{v}(z^{t_0}) \mid \mathbf{v}(z^{<t_0}), a^{<t_0}) |\det D\mathbf{v}(z^{t_0})|. \tag{36}$$

**Linear relationship between $\mathbf{T}(\mathbf{f}^{-1}(x))$ and $\mathbf{T}(\hat{\mathbf{f}}^{-1}(x))$.** Recall that we gave an explicit form to these densities in Sec. 2.1 Equations (1) & (2). By taking the logarithm on each sides of (36) we get

$$\sum_{i=1}^{d_z} \log h_i(z_i^t) + \mathbf{T}_i(z_i^t)^\top \boldsymbol{\lambda}_i(G_i^z \odot z^{<t}, G_i^a \odot a^{<t}) - \psi_i(z^{<t}, a^{<t}) \tag{37}$$

$$= \sum_{i=1}^{d_z} \log h_i(\mathbf{v}_i(z^t)) + \mathbf{T}_i(\mathbf{v}_i(z^t)))^\top \hat{\boldsymbol{\lambda}}_i(\hat{G}_i^z \odot \mathbf{v}(z^{<t}), \hat{G}_i^a \odot a^{<t}) - \hat{\psi}_i(\mathbf{v}(z^{<t}), a^{<t})$$

$$+ \log|\det D\mathbf{v}(z^t)|$$

Note that (37) holds for all $z^{<t}$ and $a^{<t}$. In particular, we evaluate it at the points given in the assumption of sufficient variability of Thm. 4. We evaluate the equation at $(z^t, z_{(p)}, a_{(p)})$ and $(z^t, z_{(0)}, a_{(0)})$ and take the difference which yields[7]

$$
\begin{aligned}
\sum_{i=1}^{d_z} \mathbf{T}_i(z_i^t)^\top [\boldsymbol{\lambda}_i(G_i^z \odot z_{(p)}, G_i^a \odot a_{(p)}) - \boldsymbol{\lambda}_i(G_i^z \odot z_{(0)}, G_i^a \odot a_{(0)})] - \psi_i(z_{(p)}, a_{(p)}) + \psi_i(z_{(0)}, a_{(0)}) \\
= \sum_{i=1}^{d_z} \mathbf{T}_i(\mathbf{v}_i(z^t))^\top [\hat{\boldsymbol{\lambda}}_i(\hat{G}_i^z \odot \mathbf{v}(z_{(p)}), \hat{G}_i^a \odot a_{(p)}) - \hat{\boldsymbol{\lambda}}_i(\hat{G}_i^z \odot \mathbf{v}(z_{(0)}), \hat{G}_i^a \odot a_{(0)})] \\
- \hat{\psi}_i(\mathbf{v}(z_{(p)}), a_{(p)}) + \hat{\psi}_i(\mathbf{v}(z_{(0)}), a_{(0)})
\end{aligned}
\tag{38}
$$

We regroup all normalization constants $\psi$ into a term $d(z_{(p)}, z_{(0)}, a_{(p)}, a_{(0)})$ and write

$$
\begin{aligned}
\mathbf{T}(z^t)^\top [\boldsymbol{\lambda}(z_{(p)}, a_{(p)}) - \boldsymbol{\lambda}(z_{(0)}, a_{(0)})] \\
= \mathbf{T}(\mathbf{v}(z^t))^\top [\hat{\boldsymbol{\lambda}}(\mathbf{v}(z_{(p)}), a_{(p)}) - \hat{\boldsymbol{\lambda}}(\mathbf{v}(z_{(0)}), a_{(0)})] + d(z_{(p)}, z_{(0)}, a_{(p)}, a_{(0)}),
\end{aligned}
\tag{39}
$$

where we used $\mathbf{T}$ and $\boldsymbol{\lambda}$ as defined in Sec. 2.1. Define

$$
w_{(p)} := \boldsymbol{\lambda}(z_{(p)}, a_{(p)}) - \boldsymbol{\lambda}(z_{(0)}, a_{(0)})
\tag{40}
$$

$$
\hat{w}_{(p)} := \hat{\boldsymbol{\lambda}}(\mathbf{v}(z_{(p)}), a_{(p)}) - \hat{\boldsymbol{\lambda}}(\mathbf{v}(z_{(0)}), a_{(0)})
\tag{41}
$$

$$
d_{(p)} := d(z_{(p)}, z_{(0)}, a_{(p)}, a_{(0)}),
\tag{42}
$$

which yields

$$
\mathbf{T}(z^t)^\top w_{(p)} = \mathbf{T}(\mathbf{v}(z^t))^\top \hat{w}_{(p)} + d_{(p)}.
\tag{43}
$$

We can regroup the $w_{(p)}$ into a matrix and the $d_{(p)}$ into a vector:

$$
W := [w_{(1)} ... w_{(kd_z)}] \in \mathbb{R}^{kd_z \times kd_z}
\tag{44}
$$

$$
\hat{W} := [\hat{w}_{(1)} ... \hat{w}_{(kd_z)}] \in \mathbb{R}^{kd_z \times kd_z}
\tag{45}
$$

$$
d := [d_{(1)} ... d_{(kd_z)}] \in \mathbb{R}^{1 \times kd_z}.
\tag{46}
$$

Since (43) holds for all $1 \le p \le kd_z$, we can write

$$
\mathbf{T}(z^t)^\top W = \mathbf{T}(\mathbf{v}(z^t))^\top \hat{W} + d.
\tag{47}
$$

Note that $W$ is invertible by the assumption of variability, hence

$$
\mathbf{T}(z^t)^\top = \mathbf{T}(\mathbf{v}(z^t))^\top \hat{W} W^{-1} + d W^{-1}.
\tag{48}
$$

Let $b := (d W^{-1})^\top$ and $L := (\hat{W} W^{-1})^\top$. We can thus rewrite as

$$
\mathbf{T}(z^t) = L\mathbf{T}(\mathbf{v}(z^t)) + b.
\tag{49}
$$

---

7. Note that $z_{(0)}$ and $z_{(p)}$ can have different dimensionalities if they come from different time steps. It is not an issue to combine equations from different time steps, since (37) holds for all values of $t$, $z^t$, $z^{<t}$ and $a^{<t}$.

**Invertibility of** $L$**.** We now show that $L$ is invertible. By Lemma 10, the fact that the $\mathbf{T}_i$ are minimal (Assumption 1) is equivalent to, for all $i \in \{1, ..., d_z\}$, having elements $z_i^{(0)}, ..., z_i^{(k)}$ in $\mathcal{Z}$ such that the family of vectors

$$\mathbf{T}_i(z_i^{(1)}) - \mathbf{T}_i(z_i^{(0)}), \ ... \ , \mathbf{T}_i(z_i^{(k)}) - \mathbf{T}_i(z_i^{(0)}) \tag{50}$$

is independent. Define

$$z^{(0)} := [z_1^{(0)} \ldots z_{d_z}^{(0)}]^\top \in \mathbb{R}^{d_z} \tag{51}$$

For all $i \in \{1, ..., d_z\}$ and all $p \in \{1, ..., k\}$, define the vectors

$$z^{(p,i)} := [z_1^{(0)} \ldots z_{i-1}^{(0)} \ z_i^{(p)} \ z_{i+1}^{(0)} \ldots z_{d_z}^{(0)}]^\top \in \mathbb{R}^{d_z} \ . \tag{52}$$

For a specific $1 \le p \le k$ and $i \in \{1, ..., d_z\}$, we can take the following difference based on (49)

$$\mathbf{T}(z^{(p,i)}) - \mathbf{T}(z^{(0)}) = L[\mathbf{T}(\mathbf{v}(z^{(p,i)})) - \mathbf{T}(\mathbf{v}(z^{(0)}))] \,, \tag{53}$$

where the left hand side is a vector filled with zeros except for the block corresponding to $\mathbf{T}_i(z_i^{(p,i)}) - \mathbf{T}_i(z_i^{(0)})$. Let us define

$$\Delta\mathbf{T}^{(i)} := [\mathbf{T}(z^{(1,i)}) - \mathbf{T}(z^{(0)}) \ \ldots \ \mathbf{T}(z^{(k,i)}) - \mathbf{T}(z^{(0)})] \in \mathbb{R}^{kd_z \times k}$$
$$\Delta\hat{\mathbf{T}}^{(i)} := [\mathbf{T}(\mathbf{v}(z^{(1,i)})) - \mathbf{T}(\mathbf{v}(z^{(0)})) \ \ldots \ \mathbf{T}(\mathbf{v}(z^{(k,i)})) - \mathbf{T}(\mathbf{v}(z^{(0)}))] \in \mathbb{R}^{kd_z \times k} \ .$$

Note that the columns of $\Delta\mathbf{T}^{(i)}$ are linearly independent and all rows are filled with zeros except for the block of rows $\{(i-1)k+1, ..., ik\}$. We can thus rewrite (53) in matrix form

$$\Delta\mathbf{T}^{(i)} = L\Delta\hat{\mathbf{T}}^{(i)} \,. \tag{54}$$

We can regroup these equations for every $i$ by doing

$$[\Delta\mathbf{T}^{(1)} \ ... \ \Delta\mathbf{T}^{(d_z)}] = L[\Delta\hat{\mathbf{T}}^{(1)} \ ... \ \Delta\hat{\mathbf{T}}^{(d_z)}] \,. \tag{55}$$

Notice that the newly formed matrix on the left hand side has size $kd_z \times kd_z$ and is block diagonal. Since every block is invertible, the left hand side of (55) is an invertible matrix, which in turn implies that $L$ is invertible.

We can rewrite (49) as

$$\mathbf{T}(\mathbf{f}^{-1}(x)) = L\mathbf{T}(\hat{\mathbf{f}}^{-1}(x)) + b \ \forall x \in \mathcal{X} \,, \tag{56}$$

which completes the proof of the first part of $\theta \sim_L \hat{\theta}$.

**Linear identifiability of natural parameters.** We now want to show the second part of the equivalence which links $\boldsymbol{\lambda}$ and $\hat{\boldsymbol{\lambda}}$. We start from (37) and rewrite it using $\mathbf{T}$ and $\boldsymbol{\lambda}$

$$\mathbf{T}(z^t)^\top \boldsymbol{\lambda}(z^{<t}, a^{<t}) = \mathbf{T}(\mathbf{v}(z^t)))^\top \hat{\boldsymbol{\lambda}}(\mathbf{v}(z^{<t}), a^{<t}) + d(z^{<t}, a^{<t}) + c(z^t) \,, \tag{57}$$

where all terms depending only on $z^t$ are absorbed in $c(z^t)$ and all terms depending only on $z^{<t}$ and $a^{<t}$ are absorbed in $d(z^{<t}, a^{<t})$. Using (49), we can rewrite (57) as

$$\mathbf{T}(\mathbf{v}(z^t))^\top L^\top \boldsymbol{\lambda}(z^{<t}, a^{<t}) + b^\top \boldsymbol{\lambda}(z^{<t}, a^{<t}) = \tag{58}$$
$$\mathbf{T}(\mathbf{v}(z^t))^\top \hat{\boldsymbol{\lambda}}(\mathbf{v}(z^{<t}), a^{<t}) + d(z^{<t}, a^{<t}) + c(z^t)$$
$$\mathbf{T}(\mathbf{v}(z^t))^\top L^\top \boldsymbol{\lambda}(z^{<t}, a^{<t}) = \tag{59}$$
$$\mathbf{T}(\mathbf{v}(z^t))^\top \hat{\boldsymbol{\lambda}}(\mathbf{v}(z^{<t}), a^{<t}) + \bar{d}(z^{<t}, a^{<t}) + c(z^t),$$

where $\bar{d}(z^{<t}, a^{<t})$ absorbs all terms depending only on $z^{<t}$ and $a^{<t}$. Simplifying further we get

$$\mathbf{T}(\mathbf{v}(z^t))^\top (L^\top \boldsymbol{\lambda}(z^{<t}, a^{<t}) - \hat{\boldsymbol{\lambda}}(\mathbf{v}(z^{<t}), a^{<t})) = \bar{d}(z^{<t}, a^{<t}) + c(z^t) \tag{60}$$
$$\mathbf{T}(z^t)^\top (L^\top \boldsymbol{\lambda}(z^{<t}, a^{<t}) - \hat{\boldsymbol{\lambda}}(\mathbf{v}(z^{<t}), a^{<t})) = \bar{d}(z^{<t}, a^{<t}) + c(\mathbf{v}^{-1}(z^t)), \tag{61}$$

where the second equality is obtained by making the change of variable $z^t \leftarrow \mathbf{v}^{-1}(z^t)$. Again, one can take the difference for two distinct values of $z^t$, say $z^t$ and $\bar{z}^t$, while keeping $z^{<t}$ and $a^{<t}$ constant which yields

$$[\mathbf{T}(z^t) - \mathbf{T}(\bar{z}^t)]^\top (L^\top \boldsymbol{\lambda}(z^{<t}, a^{<t}) - \hat{\boldsymbol{\lambda}}(\mathbf{v}(z^{<t}), a^{<t})) = c(\mathbf{v}^{-1}(z^t)) - c(\mathbf{v}^{-1}(\bar{z}^t)). \tag{62}$$

Using an approach analogous to what we did earlier in the "Invertible $L$" step, we can construct an invertible matrix $[\Delta \mathbf{T}^{(1)} ... \Delta \mathbf{T}^{(d_z)}]$ and get

$$[\Delta \mathbf{T}^{(1)} ... \Delta \mathbf{T}^{(d_z)}]^\top (L^\top \boldsymbol{\lambda}(z^{<t}, a^{<t}) - \hat{\boldsymbol{\lambda}}(\mathbf{v}(z^{<t}), a^{<t})) = [\Delta c^{(1)} ... \Delta c^{(d_z)}], \tag{63}$$

where the $\Delta c^{(i)}$ are defined analogously to $\Delta \mathbf{T}^{(i)}$. Since $[\Delta \mathbf{T}^{(1)} ... \Delta \mathbf{T}^{(d_z)}]$ is invertible we can write

$$L^\top \boldsymbol{\lambda}(z^{<t}, a^{<t}) - \hat{\boldsymbol{\lambda}}(\mathbf{v}(z^{<t}), a^{<t}) = -c \tag{64}$$
$$L^\top \boldsymbol{\lambda}(z^{<t}, a^{<t}) + c = \hat{\boldsymbol{\lambda}}(\mathbf{v}(z^{<t}), a^{<t}),$$

where

$$c = -[\Delta \mathbf{T}^{(1)} ... \Delta \mathbf{T}^{(d_z)}]^{-\top} [\Delta c^{(1)} ... \Delta c^{(d_z)}] \tag{65}$$

which can be rewritten as

$$L^\top \boldsymbol{\lambda}(\mathbf{f}^{-1}(x^{<t}), a^{<t}) + c = \hat{\boldsymbol{\lambda}}(\hat{\mathbf{f}}^{-1}(x^{<t}), a^{<t}). \tag{66}$$

This completes the proof.■

### A.3. Theory for disentanglement via mechanism sparsity

To understand the proof of Thm. 5, it will be useful to see it as the combination of two other theorems, Thm. 21 & 22. These theorems can be understood as specialized versions of Thm. 5 for *time-sparsity* and *action-sparsity*, respectively. This section is organised as follows. App. A.3.1 present the lemmas central to all three theorems. App. A.3.2 & A.3.3 present proofs of the specialized time-sparsity theorem (Thm. 21) and the action-sparsity theorem (Thm. 22), respectively. App. A.3.4 finally demonstrates the theorem presented in the main text, Thm. 5. All permutation-identifiability theorems are based on the linear identifiability theorem (Thm. 4). The structure of the proof is summarized in Fig. 4.

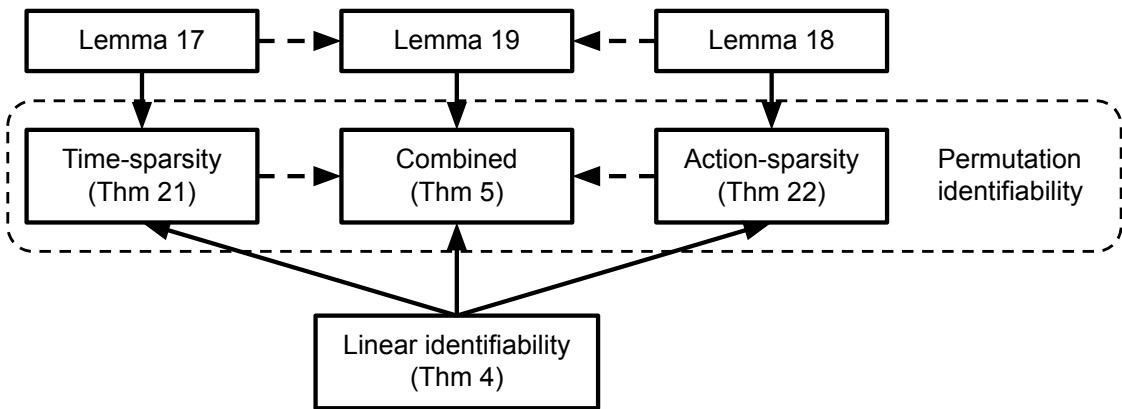

Figure 4: **Proofs structure.** A solid arrow from $A$ to $B$ means $A$ is used in the proof of $B$. A dotted arrow from $A$ to $B$ means the proof of $B$ reuses arguments from the proof of $A$.

### A.3.1. CENTRAL LEMMAS FOR THM. 5, 21 & 22

Throughout this section, we abstract away the details of our problem setting by working with an arbitrary function of the form

$$\Lambda : \Gamma \to \mathbb{R}^{m \times n}, \tag{67}$$

where $\Gamma$ is some arbitrary set. This function will be replaced by the Jacobian of the transition function for temporal sparsity, or by the discrete derivative of the action function for action sparsity. We will study how some notion of sparsity of this function behaves when composed with an invertible linear transformation. The $j$th column of $\Lambda(\gamma)$ and its $i$th row will be denoted as $\Lambda_{\cdot,j}(\gamma)$ and $\Lambda_{i,\cdot}(\gamma)$, respectively. We use analogous notation for subset of indices $S \subset \{1, ..., m\} \times \{1, ..., n\}$:

$$S_{i,\cdot} := \{j \mid (i,j) \in S\} \tag{68}$$
$$S_{\cdot,j} := \{i \mid (i,j) \in S\}. \tag{69}$$

Hence, the above sets correspond to horizontal and vertical slices of $S$, respectively. We introduce further notations in the following definition.

**Definition 11 (Aligned subspaces of $\mathbb{R}^m$)** *Given a subset $S \subset \{1, ..., m\}$, we define*

$$\mathbb{R}^m_S := \{x \in \mathbb{R}^m \mid i \notin S \implies x_i = 0\}. \tag{70}$$

**Definition 12 (Aligned subspaces of $\mathbb{R}^{m \times n}$)** *Given a subset $S \subset \{1, ..., m\} \times \{1, ..., n\}$, we define*

$$\mathbb{R}^{m \times n}_S := \{M \in \mathbb{R}^{m \times n} \mid (i,j) \notin S \implies M_{i,j} = 0\}. \tag{71}$$

*Analogously, given a binary matrix $B \in \{0,1\}^{m \times n}$, we define*

$$\mathbb{R}^{m \times n}_B := \{M \in \mathbb{R}^{m \times n} \mid B_{i,j} = 0 \implies M_{i,j} = 0\}. \tag{72}$$

We can now define what we mean by sparsity:

**Definition 13 (Sparsity pattern of $\Lambda$)** *A sparsity pattern of $\Lambda$ is the smallest subset $S$ of $\{1, ..., m\} \times \{1, ..., n\}$ such that $\Lambda(\Gamma) \subset \mathbb{R}_S^{m \times n}$.*

Thus, the sparsity pattern of $\Lambda$ describes which entries of the matrix $\Lambda(\gamma)$ are not zero for some $\gamma \in \Gamma$. The following two simple lemmas will be key for the following results.

**Lemma 14** *Let $S, S' \subset \{1, ..., m\}^2$ and let $(B_{i,j})_{(i,j) \in S}$ be a basis of $\mathbb{R}_S^{m \times m}$. Let $L$ be a real $m \times m$ matrix. Then*

$$\forall (i, j) \in S, \ L^\top B_{i,j} L \in \mathbb{R}_{S'}^{m \times m}$$
$$\implies \forall (i, j) \in S, \ L_{i,\cdot}^\top L_{j,\cdot} \in \mathbb{R}_{S'}^{m \times m}. \tag{73}$$

*Proof.* Choose $(i_0, j_0) \in S$. We can write the one-hot matrix $E_{i_0, j_0}$ as $\sum_{(i,j) \in S} \alpha_{i,j} B_{i,j}$ for some coefficients $\alpha_{i,j}$. Thus

$$L_{i_0,\cdot}^\top L_{j_0,\cdot} = L^\top e_{i_0} e_{j_0}^\top L \tag{74}$$
$$= L^\top E_{i_0, j_0} L \tag{75}$$
$$= L^\top \left( \sum_{(i,j) \in S} \alpha_{i,j} B_{i,j} \right) L \tag{76}$$
$$= \sum_{(i,j) \in S} \alpha_{i,j} L^\top B_{i,j} L \in \mathbb{R}_{S'}^{m \times m}, \tag{77}$$

where the final "$\in$" holds because each element of the sum is in $\mathbb{R}_{S'}^{m \times m}$. ∎

**Lemma 15** *Let $S, S' \subset \{1, ..., m\}$ and $(b_i)_{i \in S}$ be a basis of $\mathbb{R}_S^m$. Let $L$ be a real $m \times m$ matrix. Then*

$$\forall i \in S, \ Lb_i \in \mathbb{R}_{S'}^m \implies \forall i \in S, \ L_{\cdot, i} \in \mathbb{R}_{S'}^m. \tag{78}$$

*Proof.* Choose $i_0 \in S$. We can write the one-hot vector $e_{i_0}$ as $\sum_{i \in S} \alpha_i b_i$ for some coefficients $\alpha_i$ (since $(b_i)_{i \in S}$ forms a basis). Thus

$$L_{\cdot, i_0} = Le_{i_0} = L \sum_{i \in S} \alpha_i b_i = \sum_{i \in S} \alpha_i Lb_i \in \mathbb{R}_{S'}^m, \tag{79}$$

where the final "$\in$" holds because each element of the sum is in $\mathbb{R}_{S'}^m$. ∎

We also need to define what we are looking for

**Definition 16 (Permutation-Scaling Matrix)** *A matrix is said to be permutation-scaling if every row or column contains exactly one non-zero element.*

Alternatively, permutation-scaling matrices are defined as the matrices that can be written as $PD$ where $P$ is a permutation matrix and $D$ is a full rank diagonal matrix.

We are now ready to show the lemma central to Thm. 21.

**Lemma 17 (Sparsest $L^\top \Lambda(\cdot) L$ implies $L$ is a permutation)** *Let $\Lambda : \Gamma \to \mathbb{R}^{m \times m}$ with sparsity pattern $S$. Let $L \in \mathbb{R}^{m \times m}$ be an invertible matrix and $\hat{S}$ be the sparsity pattern of $\hat{\Lambda}(.) := L^\top \Lambda(\cdot) L$. Assume that*

1. **[Sufficient Variability]** $span(\Lambda(\Gamma)) = \mathbb{R}_S^{m \times m}$.

*Then there exists an $m$-permutation $\sigma$ such that $\sigma(S) \subset \hat{S}$, where $\sigma(S) := \{(\sigma(i), \sigma(j)) \mid (i, j) \in S\}$. Further assume that*

2. **[Sparsity]** $|\hat{S}| \leq |S|$ .

*Then $\sigma(S) = \hat{S}$. Further assume that*

3. **[Graphical Criterion]** *For all $p \in \{1, ..., m\}$, there exist index sets $\mathcal{I}, \mathcal{J} \subset \{1, ..., m\}$ such that*

$$\left( \bigcap_{i \in \mathcal{I}} S_{i, \cdot} \right) \cap \left( \bigcap_{j \in \mathcal{J}} S_{\cdot, j} \right) = \{p\} .$$

*Then $L$ is a permutation-scaling matrix.*

*Proof.* We separate the proof in four steps. The first step leverages the Assumption 1 and Lemma 14 to show that $L$ must contain "many" zeros. The second step leverages the invertibility of $L$ to show that $\sigma(S) \subset \hat{S}$. The thirst step uses Assumption 2 to establish $\sigma(S) = \hat{S}$. The fourth step uses Assumption 3 to show that $L$ must have a permutation structure.

We will denote by $N$ the sparsity pattern of $L$ (Definition 13). $N$ is thus the set of index couples corresponding to nonzero entries of $L$.

**Step 1:** By Assumption 1, there exists $(\gamma_{i,j})_{(i,j) \in S}$ such that $(\Lambda(\gamma_{i,j}))_{(i,j) \in S}$ spans $\mathbb{R}_S^{m \times m}$. Moreover, by the definition of $\hat{S}$ as sparsity pattern of $L^\top \Lambda(.) L$ (Definition 13), we have for all $(i, j) \in S$

$$L^\top \Lambda(\gamma_{i,j}) L \in \mathbb{R}_{\hat{S}}^{m \times m} . \tag{80}$$

Then, by Lemma 14, we must have

$$\forall (i, j) \in S, \ L_{i, \cdot}^\top L_{j, \cdot} \in \mathbb{R}_{\hat{S}}^{m \times m} . \tag{81}$$

We can rewrite our finding as

$$\forall (i, j) \in S, \ N_{i, \cdot} \times N_{j, \cdot} \subset \hat{S} . \tag{82}$$

**Step 2:** Since the matrix $L$ is invertible, its determinant is non-zero, i.e.

$$\det(L) = \sum_{\sigma \in \mathfrak{S}_m} \text{sign}(\sigma) \prod_{j=1}^m L_{\sigma(j), j} \neq 0 , \tag{83}$$

where $\mathfrak{S}_m$ is the set of $m$-permutations. This equation implies that at least one term of the sum is non-zero, meaning

$$\exists \sigma \in \mathfrak{S}_m, \forall j \leq m, L_{\sigma(j), j} \neq 0 . \tag{84}$$

In other words, this $m$-permutation $\sigma$ is included in the sparsity pattern of $L$ – i.e. it is such that for all $i \in \{1, ..., m\}$, $\sigma(i) \in N_{\cdot, i}$.

We thus have for all $(i, j) \in S$ that

$$(\sigma(i), \sigma(j)) \in N_{i,\cdot} \times N_{j,\cdot} \subset \hat{S}, \tag{85}$$

which implies that

$$\sigma(S) \subset \hat{S}, \tag{86}$$

where $\sigma(S) := \{(\sigma(i), \sigma(j)) \mid (i, j) \in S\}$. This proves the first claim of the Thm..

**Step 3:** By Assumption 2, $|\hat{S}| \leq |S| = |\sigma(S)|$, we must have that

$$\sigma(S) = \hat{S}, \tag{87}$$

which proves the second statement of the Thm..

**Step 4:** To show that $L$ is a permutation-scaling matrix we show that any two rows cannot have nonzero entries on the same column. We will proceed by contradiction.

Suppose there are distinct rows $i_1$ and $i_2$ such that $N_{i_1,\cdot} \cap N_{i_2,\cdot} \neq \emptyset$. Choose $i_3$ such that

$$\sigma(i_3) \in N_{i_1,\cdot} \cap N_{i_2,\cdot}. \tag{88}$$

Notice that we must have $i_3 \neq i_1$ or $i_3 \neq i_2$. Without loss of generality, assume the former holds. By Assumption 3, there are sets of indices $\mathcal{I}$ and $\mathcal{J}$ such that $\left(\bigcap_{i \in \mathcal{I}} S_{i,\cdot}\right) \cap \left(\bigcap_{j \in \mathcal{J}} S_{\cdot,j}\right) = \{i_1\}$. Since $i_3 \neq i_1$, one of the two following statement holds:

$$\exists\, i_0 \in \mathcal{I} \text{ s.t. } i_3 \notin S_{i_0,\cdot} \tag{89}$$

$$\exists\, j_0 \in \mathcal{J} \text{ s.t. } i_3 \notin S_{\cdot,j_0}. \tag{90}$$

**Case 1.** Suppose $\exists\, i_0 \in \mathcal{I}$ s.t. $i_3 \notin S_{i_0,\cdot}$. We must have $i_1 \in S_{i_0,\cdot}$, which is equivalent to having

$$(i_0, i_1) \in S. \tag{91}$$

Equations (82) & (91) imply that

$$N_{i_0,\cdot} \times N_{i_1,\cdot} \subset \hat{S}, \tag{92}$$

and since $(\sigma(i_0), \sigma(i_3)) \in N_{i_0,\cdot} \times N_{i_1,\cdot}$ (by (88)), we have that

$$(\sigma(i_0), \sigma(i_3)) \in \hat{S}. \tag{93}$$

But this implies, by (87), that $(i_0, i_3) \in S$, which contradicts (89).

**Case 2.** Suppose $\exists\, j_0 \in \mathcal{J}$ s.t. $i_3 \notin S_{\cdot,j_0}$. We must have that $i_1 \in S_{\cdot,j_0}$ which is equivalent to having

$$(i_1, j_0) \in S. \tag{94}$$

Equations (82) & (94) imply that

$$N_{i_1,\cdot} \times N_{j_0,\cdot} \subset \hat{S}, \tag{95}$$

and since $(\sigma(i_3), \sigma(j_0)) \in N_{i_1, \cdot} \times N_{j_0, \cdot}$ (by (88)), we have that

$$(\sigma(i_3), \sigma(j_0)) \in \hat{S}. \tag{96}$$

But this implies, by (87), that $(i_3, j_0) \in S$, which violates the fact that $i_3 \notin S_{\cdot, j_0}$. ∎

We now present the lemma central to Thm. 22. The statement and its proof are very similar to Lemma 17.

**Lemma 18 (Sparsest $L^T \Lambda(.)$ implies $L$ is a permutation)** *Let $\Lambda : \Gamma \to \mathbb{R}^{m \times n}$ with sparsity pattern $S$. Let $L \in \mathbb{R}^{m \times m}$ be an invertible matrix and $\hat{S}$ be the sparsity pattern of $\hat{\Lambda} := L\Lambda$. Assume that*

1. ***[Sufficient Variability]*** *For all $j \in \{1, ..., n\}$, $span(\Lambda_{\cdot, j}(\Gamma)) = \mathbb{R}^m_{S_{\cdot, j}}$ .*

*Then there exists an $m$-permutation $\sigma$ such that $\sigma(S) \subset \hat{S}$ where $\sigma(S) := \{(\sigma(i), j) \mid (i, j) \in S\}$. Further assume that*

2. ***[Sparsity]*** $|\hat{S}| \leq |S|$ .

*Then $\sigma(S) = \hat{S}$. Further assume that*

3. ***[Graphical Criterion]*** *For all $p \in \{1, ..., m\}$, there exists an index set $\mathcal{J} \subset \{1, ..., n\}$ such that $\bigcap_{j \in \mathcal{J}} S_{\cdot, j} = \{p\}$.*

*Then, $L$ is a permutation-scaling matrix.*

*Proof.* We separate the proof in four steps. The first step leverages the Assumption 1 and Lemma 15 to show that $L$ must contain "many" zeros. The second step leverages the invertibility of $L$ to show that $\sigma(S) \subset \hat{S}$. The third step uses Assumption 2 to show this inclusion is in fact an equality. Finally the fourth step Assumption 3 to show that $L$ must have a permutation structure.

We will denote by $N$ the sparsity pattern of $L$ (Definition 13). $N$ is thus the set of index couples corresponding to nonzero entries of $L$.

**Step 1:** Fix $j \in \{1, ..., n\}$. By Assumption 1, there exists $(\gamma_i)_{i \in S_{\cdot, j}}$ such that $(\Lambda_{\cdot, j}(\gamma_i))_{i \in S_{\cdot, j}}$ spans $\mathbb{R}^m_{S_{\cdot, j}}$. Moreover, by the definition of $\hat{S}$ as sparsity pattern of $L\Lambda(.)$ (Definition 13), we have for all $i \in S_{\cdot, j}$

$$L\Lambda_{\cdot, j}(\gamma_i) \in \mathbb{R}^m_{\hat{S}_{\cdot, j}}. \tag{97}$$

By Lemma 15, we must have

$$\forall i \in S_{\cdot, j}, \ L_{\cdot, i} \in \mathbb{R}^m_{\hat{S}_{\cdot, j}}. \tag{98}$$

Since $j$ was arbitrary, this holds for all $j$. We can thus rewrite our finding with $N$ the sparsity pattern of $L$

$$\forall (i, j) \in S, \ N_{\cdot, i} \times \{j\} \subset \hat{S}. \tag{99}$$

**Step 2:** We know there exists an $m$-permutation $\sigma$ such that for all $i \in \{1, ..., m\}$, $\sigma(i) \in N_{\cdot, i}$ (see Step 2 in Lemma 17).

We thus have for all $(i,j) \in S$ that

$$(\sigma(i), j) \in N_{\cdot,i} \times \{j\} \subset \hat{S}, \tag{100}$$

which implies that

$$\sigma(S) \subset \hat{S}, \tag{101}$$

where $\sigma(S) := \{(\sigma(i), j) \mid (i,j) \in S\}$. This proves the first statement of the Thm..

**Step 3:** By Assumption 2, $|\hat{S}| \leq |S| = |\sigma(S)|$, so the inclusion (101) is actually an equality

$$\sigma(S) = \hat{S}, \tag{102}$$

which proves the second statement.

**Step 4:** To show that $L$ is a permutation-scaling matrix, we must show that any two columns cannot have nonzero entries on the same row. We will proceed by contradiction.

Suppose there are two distinct columns $i_1$ and $i_2$ such that $N_{\cdot,i_1} \cap N_{\cdot,i_2} \neq \emptyset$. Choose $i_3$ such that

$$\sigma(i_3) \in N_{\cdot,i_1} \cap N_{\cdot,i_2}. \tag{103}$$

Notice that we must have $i_3 \neq i_1$ or $i_3 \neq i_2$. Without loss of generality, assume the former holds. By Assumption 3, there is a set of indices $\mathcal{J}$ such that $\bigcap_{j \in \mathcal{J}} S_{\cdot,j} = \{i_1\}$. Since $i_3 \neq i_1$, there must exist some $j_0 \in \mathcal{J}$ such that

$$i_3 \notin S_{\cdot,j_0}, \tag{104}$$

Moreover, we must have $i_1 \in S_{\cdot,j_0}$, which is equivalent to

$$(i_1, j_0) \in S. \tag{105}$$

Equations (99) & (105) imply that

$$N_{\cdot,i_1} \times \{j_0\} \subset \hat{S}, \tag{106}$$

and since $(\sigma(i_3), j_0) \in N_{\cdot,i_1} \times \{j_0\}$ (by (103)), we have that

$$(\sigma(i_3), j_0) \in \hat{S}. \tag{107}$$

But this implies, by (102), that $(i_3, j_0) \in S$, which contradicts (104). ∎

In Sec. A.3.4, we will present Thm. 5 and its proof which is, in some sense, the combination of Thm. 21 & 22. Its proof relies on the following lemma, which is, analogously, the combination of Lemmas 17 & 18.

**Lemma 19 (Combining Lemmas 17 & 18)** *Let $\Lambda^{(1)} : \Gamma^{(1)} \to \mathbb{R}^{m \times m}$ and $\Lambda^{(2)} : \Gamma^{(2)} \to \mathbb{R}^{m \times n}$ with sparsity pattern $S^{(1)}$ and $S^{(2)}$, respectively. Let $L \in \mathbb{R}^{m \times m}$ be an invertible matrix and $\hat{S}^{(1)}$ and $\hat{S}^{(2)}$ be the sparsity patterns of $\hat{\Lambda}^{(1)} := L^\top \Lambda^{(1)}(\cdot)L$ and $\hat{\Lambda}^{(2)} := L^\top \Lambda^{(2)}$, respectively. Assume that*

1. *[**Sufficient Variability 1**] $span(\Lambda^{(1)}(\Gamma^{(1)})) = \mathbb{R}^{m \times m}_{S^{(1)}}$.*

*Then there exists an $m$-permutation $\sigma$ such that $\sigma(S^{(1)}) \subset \hat{S}^{(1)}$ where
$\sigma(S^{(1)}) := \{(\sigma(i), \sigma(j)) \mid (i, j) \in S^{(1)}\}$. Further assume that*

2. ***[Sufficient Variability 2]*** *For all $j \in \{1, ..., n\}$, $span(\Lambda_{\cdot,j}^{(2)}(\Gamma^{(2)})) = \mathbb{R}_{S_{\cdot,j}^{(2)}}^m$.*

*Then $\sigma(S^{(2)}) \subset \hat{S}^{(2)}$ where $\sigma(S^{(2)}) := \{(\sigma(i), j) \mid (i, j) \in S^{(2)}\}$. Further assume that*

3. ***[Sparsity]*** $|\hat{S}^{(1)}| + |\hat{S}^{(2)}| \leq |S^{(1)}| + |S^{(2)}|$.

*Then $\sigma(S^{(1)}) = \hat{S}^{(1)}$ and $\sigma(S^{(2)}) = \hat{S}^{(2)}$. Further assume that*

4. ***[Graphical Criterion]*** *For all $p \in \{1, ..., m\}$, there exists an index sets $\mathcal{I}^{(1)}, \mathcal{J}^{(1)} \subset \{1, ..., m\}$
   and $\mathcal{J}^{(2)} \subset \{1, ..., n\}$ such that*

$$\left( \bigcap_{i \in \mathcal{I}^{(1)}} S_{i,\cdot}^{(1)} \right) \cap \left( \bigcap_{j \in \mathcal{J}^{(1)}} S_{\cdot,j}^{(1)} \right) \cap \left( \bigcap_{j \in \mathcal{J}^{(2)}} S_{\cdot,j}^{(2)} \right) = \{p\} \tag{108}$$

*Then, $L$ is a permutation-scaling matrix.*

*Proof.* Following proofs of Lemma 17 and 18 we separate the proof in four steps. The only real difference with these previous proofs is in step 3 where we leverage Assumption 3. Step1 leverages Assumptions 1 & 2 and Lemmas 15 & 14 to show that $L$ must contain "many" zeros. Step 2 uses the invertibility of $L$ to show $\sigma(S^{(1)}) \subset \hat{S}^{(1)}$ and $\sigma(S^{(2)}) \subset \hat{S}^{(2)}$. Step 3 uses Assumption 3 to conclude that $\sigma(S^{(1)}) = \hat{S}^{(1)}$ and $\sigma(S^{(2)}) = \hat{S}^{(2)}$. Step 4 uses the Assumption 4 to show that $L$ must have a permutation structure.

We will denote by $N$ the sparsity pattern of $L$ (Definition 13). $N$ is thus the set of index couples corresponding to nonzero entries of $L$.

**Step 1:** Here, Assumption 1 is the same as Assumption 1 of Lemma 17. The reasoning of Step 1 of its proof gives equation (82)

$$\forall (i, j) \in S^{(1)}, \ N_{i,\cdot} \times N_{j,\cdot} \subset \hat{S}^{(1)}. \tag{109}$$

Similarly, by Assumption 2, the exact same reasoning as Step 1 of Lemma 18, we reach equation (99)

$$\forall (i, j) \in S^{(2)}, \ N_{i,\cdot} \times \{j\} \subset \hat{S}^{(2)}. \tag{110}$$

**Step 2:** Following step 2 of Lemma 17, we obtain

$$\sigma(S^{(1)}) \subset \hat{S}^{(1)}, \tag{111}$$

where $\sigma(S^{(1)}) := \{(\sigma(i), \sigma(j)) \mid (i, j) \in S^{(1)}\}$, thus proving the first statement.

Similarly, following step 2 of the proof of Lemma 18, we reach

$$\sigma(S^{(2)}) \subset \hat{S}^{(2)}, \tag{112}$$

where $\sigma(S^{(2)}) := \{(\sigma(i), j) \mid (i, j) \in S^{(2)}\}$, thus proving the second statement. Be aware that $\sigma(S^{(2)})$ and $\sigma(S^{(1)})$ carry different meanings.

**Step 3:** By the Assumption 3, we have

$$|\hat{S}^{(1)}| + |\hat{S}^{(2)}| \leq |S^{(1)}| + |S^{(2)}| \tag{113}$$

$$= |\sigma(S^{(1)})| + |\sigma(S^{(2)})| \tag{114}$$

$$[\text{using } (112)] \leq |\hat{S}^{(1)}| + |\sigma(S^{(2)})| \tag{115}$$

$$[\text{using } (111)] \leq |\hat{S}^{(1)}| + |\hat{S}^{(2)}| \tag{116}$$

which implies that all the inequalities we used are actually equalities. In particular $|\sigma(S^{(1)})| = |\hat{S}^{(1)}|$ and $|\sigma(S^{(2)})| = |\hat{S}^{(2)}|$. Thanks to the inclusions (112) and (111), we finally reach $\sigma(S^{(1)}) = \hat{S}^{(1)}$ and $\sigma(S^{(2)}) = \hat{S}^{(2)}$, which proves the third statement of the Thm..

**Step 4:** To show that $L$ is a permutation-scaling matrix, we must show that every row has exactly one nonzero entry. Since $L$ is invertible, it will have at least one nonzero entry per row, so we only need to make sure it does not have more than one nonzero entry. We will proceed by contradiction.

Suppose there is a column such that the $i_1$th and $i_2$th elements are nonzero. In other words, there are two distinct rows $i_1$ and $i_2$ such that $N_{i_1,\cdot} \cap N_{i_2,\cdot} \neq \emptyset$. Choose $i_3$ such that

$$\sigma(i_3) \in N_{i_1,\cdot} \cap N_{i_2,\cdot}. \tag{117}$$

Notice $\sigma(i_3)$ is the problematic column. Since $i_1 \neq i_2$, we must have $i_3 \neq i_1$ or $i_3 \neq i_2$. Without loss of generality, assume $i_3 \neq i_1$.

By the Assumption 4, there are sets of indices $\mathcal{I}^{(1)}, \mathcal{J}^{(1)}$ and $\mathcal{J}^{(2)}$ such that

$$\left( \bigcap_{i \in \mathcal{I}^{(1)}} S_{i,\cdot}^{(1)} \right) \cap \left( \bigcap_{j \in \mathcal{J}^{(1)}} S_{\cdot,j}^{(1)} \right) \cap \left( \bigcap_{j \in \mathcal{J}^{(2)}} S_{\cdot,j}^{(2)} \right) = \{i_1\} \tag{118}$$

Since $i_3 \neq i_1$, there are three possibilities:

1. $\exists i_0 \in \mathcal{I}^{(1)}, i_3 \notin S_{i_0,\cdot}^{(1)}$ which is the same as Step 4 Case 1 of Lemma 17,

2. $\exists j_0 \in \mathcal{J}^{(1)}, i_3 \notin S_{\cdot,j_0}^{(1)}$ which is the same as Step 4 Case 2 of Lemma 17,

3. $\exists j_0 \in \mathcal{J}^{(2)}, i_3 \notin S_{\cdot,j_0}^{(2)}$ which is the same as Step 4 of Lemma 18.

From the proof of previous lemmas, we know each of these possibilities lead to a contradiction. We conclude that $L$ must be a permutation-scaling matrix. ∎

### A.3.2. PROOF OF THE SPECIALIZED TIME-SPARSITY THEOREM (THM. 21)

The following definition will be useful.

**Definition 20 (Inclusion of matrices)** *For two matrices $A$ and $B$ of same size, we write $A \subset B$ to say that $\forall i, j, A_{i,j} \neq 0 \implies B_{i,j} \neq 0$.*

We are now ready to state and prove the specialized time-sparsity theorem.

**Theorem 21 (Permutation-identifiability from time-sparsity)** *Suppose we have two models as described in Sec. 2.1 with parameters $\theta = (\mathbf{f}, \boldsymbol{\lambda}, G)$ and $\hat{\theta} = (\hat{\mathbf{f}}, \hat{\boldsymbol{\lambda}}, \hat{G})$ representing the same distribution, i.e. $\mathbb{P}_{X^{\leq T}|a;\theta} = \mathbb{P}_{X^{\leq T}|a;\hat{\theta}}$ for all $a \in \mathcal{A}^T$. Suppose the assumptions of Thm. 4 hold and that*

1. *The sufficient statistic $\mathbf{T}$ is $d_z$-dimensional ($k = 1$) and is a diffeomorphism from $\mathcal{Z}$ to $\mathbf{T}(\mathcal{Z})$.*

2. *[**Sufficient Variability**] There exist $\{(z_{(p)}, a_{(p)}, \tau_{(p)})\}_{p=1}^{||G^z||_0}$ belonging to their respective support such that*

$$\mathrm{span}\left\{D_z^{\tau_{(p)}} \boldsymbol{\lambda}(z_{(p)}, a_{(p)}) D_z \mathbf{T}(z_{(p)}^{\tau_{(p)}})^{-1}\right\}_{p=1}^{||G^z||_0} = \mathbb{R}_{G^z}^{d_z \times d_z},$$

   *where $D_z^{\tau_{(p)}} \boldsymbol{\lambda}$ and $D_z \mathbf{T}$ are the Jacobian matrices with respect to $z^{\tau_{(p)}}$ and $z$, respectively.*

*Then, there exists a permutation matrix $P$ such that $PG^z P^\top \subset \hat{G}^z$. Further assume that*

3. *[**Sparsity**] $||\hat{G}^z||_0 \leq ||G^z||_0$.*

*Then, $PG^z P^\top = \hat{G}^z$. Further assume that*

4. *[**Graphical criterion**] For all $p \in \{1, ..., d_z\}$, there exist $\mathcal{I}, \mathcal{J} \subset \{1, ..., d_z\}$ such that*

$$\left(\bigcap_{i \in \mathcal{I}} \mathbf{Pa}_i^z\right) \cap \left(\bigcap_{j \in \mathcal{J}} \mathbf{Ch}_j^z\right) = \{p\}.$$

*Then $\theta$ and $\hat{\theta}$ are permutation equivalent, $\theta \sim_P \hat{\theta}$, i.e. the model $\hat{\theta}$ is disentangled.*

*Proof.* In what follows, we drop the superscript $^z$ on $G^z$ and $\hat{G}^z$ to lighten the notation.

First of all, since the assumptions of Thm. 4 holds and the two models represent the same model, the following relations hold

$$\mathbf{T}(\mathbf{f}^{-1}(x)) = L\mathbf{T}(\hat{\mathbf{f}}^{-1}(x)) + b \tag{119}$$

$$L^\top \boldsymbol{\lambda}(\mathbf{f}^{-1}(x^{<t}), a^{<t}) + c = \hat{\boldsymbol{\lambda}}(\hat{\mathbf{f}}^{-1}(x^{<t}), a^{<t}). \tag{120}$$

We can rearrange (119) to obtain

$$\hat{\mathbf{f}}^{-1}(x) = \mathbf{T}^{-1}(L^{-1}(\mathbf{T}(\mathbf{f}^{-1}(x)) - b)) \tag{121}$$

$$\hat{\mathbf{f}}^{-1} \circ \mathbf{f}(z) = \mathbf{T}^{-1}(L^{-1}(\mathbf{T}(z) - b)) \tag{122}$$

$$\mathbf{v}(z) = \mathbf{T}^{-1}(L^{-1}(\mathbf{T}(z) - b)), \tag{123}$$

where we defined $\mathbf{v} := \hat{\mathbf{f}}^{-1} \circ \mathbf{f}$. Taking the derivative of (123) w.r.t. $z$, we obtain

$$D\mathbf{v}(z) = D\mathbf{T}^{-1}(L^{-1}(\mathbf{T}(z) - b))L^{-1}D\mathbf{T}(z) \tag{124}$$

$$= D\mathbf{T}^{-1}(\mathbf{T}(\mathbf{v}(z)))L^{-1}D\mathbf{T}(z) \tag{125}$$

$$= D\mathbf{T}(\mathbf{v}(z))^{-1}L^{-1}D\mathbf{T}(z). \tag{126}$$

We can rewrite (120) as

$$L^\top \boldsymbol{\lambda}(z^{<t}, a^{<t}) + c = \hat{\boldsymbol{\lambda}}(\mathbf{v}(z^{<t}), a^{<t}). \tag{127}$$

By taking the derivative of the above equation w.r.t. $z^\tau$ for some $\tau \in \{1, ..., t-1\}$, we obtain

$$L^\top D_z^\tau \boldsymbol{\lambda}(z^{<t}, a^{<t}) = D_z^\tau \hat{\boldsymbol{\lambda}}(\mathbf{v}(z^{<t}), a^{<t}) D\mathbf{v}(z^\tau), \tag{128}$$

where we use $D_z^\tau$ to make explicit the fact that we are taking the derivative with respect to $z^\tau$. By plugging (126) in the above equation and rearranging the terms, we get the master equation

$$\boxed{L^\top D_z^\tau \boldsymbol{\lambda}(z^{<t}, a^{<t}) D\mathbf{T}(z^\tau)^{-1} L = D_z^\tau \hat{\boldsymbol{\lambda}}(\mathbf{v}(z^{<t}), a^{<t}) D\mathbf{T}(\mathbf{v}(z^\tau))^{-1}.} \tag{129}$$

It is now time to make the connection between the mathematical objects of the present theorem and the more abstract ones of Lemma 17

$$L^\top \underbrace{D_z^\tau \boldsymbol{\lambda}(z^{<t}, a^{<t}) D\mathbf{T}(z^\tau)^{-1}}_{\Lambda(\gamma)} L = \underbrace{D_z^\tau \hat{\boldsymbol{\lambda}}(\mathbf{v}(z^{<t}), a^{<t}) D\mathbf{T}(\mathbf{v}(z^\tau))^{-1}}_{\hat{\Lambda}(\gamma)}. \tag{130}$$

where the argument $\gamma \in \Gamma$ of the abstract function $\Lambda(\gamma)$ corresponds to $(z^{<t}, a^{<t}, \tau)$.

Define $S$ and $\hat{S}$ the sparsity patterns (Def. 13) of $\Lambda$ and $\hat{\Lambda}$ respectively. The key point of this proof is to notice the correspondence between sparsity patterns of Jacobians, and dependency graphs. Assumption 2 of the present theorem guarantees that $D_z^\tau \boldsymbol{\lambda}(z^{<t}, a^{<t}) D\mathbf{T}(z^\tau)^{-1}$ spans $\mathbb{R}_G^{m \times m}$, which implies that the sparsity pattern of this Jacobian is equal to $G$

$$S = G \tag{131}$$

in the sense that $(i, j) \in S \iff G_{i,j} = 1$. By assumption 1, $D\mathbf{T}(\cdot)^{-1}$ is diagonal and full rank, thus the sparsity pattern of $D_z^\tau \hat{\boldsymbol{\lambda}}(\mathbf{v}(z^{<t}), a^{<t}) D\mathbf{T}(\mathbf{v}(z^\tau))^{-1}$ is the same as $D_z^\tau \hat{\boldsymbol{\lambda}}(\mathbf{v}(z^{<t}), a^{<t})$. Notice that if $\hat{G}_{i,j} = 0$, then $D_z^\tau \hat{\boldsymbol{\lambda}}(\mathbf{v}(z^{<t}), a^{<t})_{i,j} = 0$ everywhere, and thus $(i, j) \notin \hat{S}$. Taking the contraposition, we get

$$\hat{S} \subset \hat{G} \tag{132}$$

in the sense that $(i, j) \in \hat{S} \implies \hat{G}_{i,j} = 1$.

We now proceed to demonstrate that all three statements of the present theorem holds. This is done by showing that all three assumptions of Lemma 17 are satisfied by exploiting the correspondence between them and Assumptions 2, 3 & 4 of the present theorem.

**Statement 1:** Assumption 1 of Lemma 17 directly holds for $\Lambda(\gamma) = D_z^\tau \boldsymbol{\lambda}(z^{<t}, a^{<t}) D\mathbf{T}(z^\tau)^{-1}$ by Assumption 2 of the present theorem. This implies that there exists a permutation $\sigma$ such that $\sigma(S) \subset \hat{S}$. Using (132) & (131), we have that

$$PGP^\top = \sigma(S) \subset \hat{S} \subset \hat{G}, \tag{133}$$

where $P$ is the permutation matrix associated with $\sigma$. This proves the first statement.

**Statement 2:** We will now show that Assumption 2 of Lemma 17 holds. Since $S = G$ and $\hat{S} \subset \hat{G}$, we have

$$|S| = ||G||_0 \tag{134}$$

$$|\hat{S}| \leq ||\hat{G}||_0. \tag{135}$$

By Assumption 3, $||\hat{G}||_0 \leq ||G||_0$. Thus, we have

$$|\hat{S}| \leq ||\hat{G}||_0 \leq ||G||_0 = |S|. \tag{136}$$

The above equation is precisely Assumption 2 of Lemma 17. Using $||\hat{G}||_0 \leq ||G||_0$ and (133), we can easily see that

$$PGP^{\top} = \hat{G}, \tag{137}$$

which proves the second statement.

**Statement 3:** We finally show that Assumption 3 of Lemma 17 holds. By Assumption 4 of the present theorem, we have that for all $p \in \{1, ..., d_z\}$, there are subsets $\mathcal{I}, \mathcal{J} \subset \{1, ..., d_a\}$ such that

$$\left( \bigcap_{i \in \mathcal{I}} \mathbf{Pa}_i \right) \cap \left( \bigcap_{j \in \mathcal{J}} \mathbf{Ch}_j \right) = \{p\} \iff \left( \bigcap_{i \in \mathcal{I}} S_{i,\cdot} \right) \cap \left( \bigcap_{j \in \mathcal{J}} S_{\cdot,j} \right) = \{p\}, \tag{138}$$

where the equivalence holds because $S = G$. We can thus apply Lemma 17 to conclude that $L$ is a permutation-scaling matrix, which is the third and final statement ■

A.3.3. PROOF OF THE SPECIALIZED ACTION-SPARSITY THEOREM (THM. 22)

The proof of Thm. 22 is very similar in spirit to the proof of Thm. 21, except we use Lemma 18 instead of Lemma 17.

**Theorem 22 (Permutation-identifiability from action-sparsity)** *Suppose we have two models as described in Sec. 2.1 with parameters $\theta = (\mathbf{f}, \boldsymbol{\lambda}, G^a)$ and $\hat{\theta} = (\hat{\mathbf{f}}, \hat{\boldsymbol{\lambda}}, \hat{G}^a)$ representing the same distribution, i.e. $\mathbb{P}_{X^{\leq T}|a;\theta} = \mathbb{P}_{X^{\leq T}|a;\hat{\theta}}$ for all $a \in \mathcal{A}^T$. Suppose the assumptions of Thm. 4 hold and that*

1. *Each $Z_i^t$ has a 1-dimensional sufficient statistic, i.e. $k = 1$.*

2. *[**Sufficient variability**] For all $\ell \in \{1, ..., d_a\}$, there exist $\{(z_{(p)}, a_{(p)}, \epsilon_{(p)}, \tau_{(p)})\}_{p=1}^{|\mathbf{Ch}_\ell^a|}$ belonging to their respective support such that*

$$\mathrm{span} \left\{ \Delta_\ell^{\tau_{(p)}} \boldsymbol{\lambda}(z_{(p)}, a_{(p)}, \epsilon_{(p)}) \right\}_{p=1}^{|\mathbf{Ch}_\ell^a|} = \mathbb{R}^{d_z}_{\mathbf{Ch}_\ell^a}.$$

*Then, there exists a permutation matrix $P$ such that $PG^a \subset \hat{G}^a$. Further assume that*

3. *[**Sparsity**] $||\hat{G}^a||_0 \leq ||G^a||_0$.*

*Then, $PG^a = \hat{G}^a$. Further assume that*

4. *[**Graphical criterion**] For all $p \in \{1, ..., d_z\}$, there exist $\mathcal{L} \subset \{1, ..., d_a\}$ such that*

$$\bigcap_{\ell \in \mathcal{L}} \mathbf{Ch}_\ell^a = \{p\}.$$

*Then, $\theta$ and $\hat{\theta}$ are permutation-equivalent.*

*Proof.* In what follows, we drop the superscript $^a$ on the the graphs $G^a$ and $\hat{G}^a$ to lighten notation. First of all, since the assumptions of Thm. 4 holds, we must have that the following relations hold

$$\mathbf{T}(\mathbf{f}^{-1}(x)) = L\mathbf{T}(\hat{\mathbf{f}}^{-1}(x)) + b \tag{139}$$

$$L^\top \boldsymbol{\lambda}(\mathbf{f}^{-1}(x^{<t}), a^{<t}) + c = \hat{\boldsymbol{\lambda}}(\hat{\mathbf{f}}^{-1}(x^{<t}), a^{<t}). \tag{140}$$

We can rewrite (140) as

$$L^\top \boldsymbol{\lambda}(z^{<t}, a^{<t}) + c = \hat{\boldsymbol{\lambda}}(\mathbf{v}(z^{<t}), a^{<t}). \tag{141}$$

We can take a partial differences w.r.t. $a_\ell^\tau$ (defined in (6)) on both sides of the equation to obtain

$$L^\top \Delta_\ell^\tau \boldsymbol{\lambda}(z^{<t}, a^{<t}, \epsilon) = \Delta_\ell^\tau \hat{\boldsymbol{\lambda}}(\mathbf{v}(z^{<t}), a^{<t}, \epsilon), \tag{142}$$

where $\epsilon$ is some real number. We can regroup the partial difference for every $\ell \in \{1, ..., d_a\}$ and get

$$\Delta^\tau \boldsymbol{\lambda}(z^{<t}, a^{<t}, \boldsymbol{\epsilon}) := \left[ \Delta_1^\tau \boldsymbol{\lambda}(z^{<t}, a^{<t}, \epsilon_1) \ldots \Delta_{d_a}^\tau \boldsymbol{\lambda}(z^{<t}, a^{<t}, \epsilon_{d_a}) \right] \in \mathbb{R}^{d_z \times d_a}.$$

This allows us to rewrite (142) and obtain the master equation

$$\boxed{L^\top \Delta^\tau \boldsymbol{\lambda}(z^{<t}, a^{<t}, \boldsymbol{\epsilon}) = \Delta^\tau \hat{\boldsymbol{\lambda}}(\mathbf{v}(z^{<t}), a^{<t}, \boldsymbol{\epsilon}).} \tag{143}$$

We now make the connection between the mathematical objects of the present theorem and the more abstract ones of Lemma 18

$$L^\top \underbrace{\Delta^\tau \boldsymbol{\lambda}(z^{<t}, a^{<t}, \boldsymbol{\epsilon})}_{\Lambda(\gamma)} = \underbrace{\Delta^\tau \hat{\boldsymbol{\lambda}}(\mathbf{v}(z^{<t}), a^{<t}, \boldsymbol{\epsilon})}_{\hat{\Lambda}(\gamma)}. \tag{144}$$

where the argument $\gamma \in \Gamma$ of the abstract function $\Lambda(\gamma)$ corresponds to $(z^{<t}, a^{<t}, \boldsymbol{\epsilon}, \tau)$.

Define $S$ and $\hat{S}$ the sparsity patterns of $\Lambda$ and $\hat{\Lambda}$ respectively. The key point of this proof is to notice the correspondence between sparsity patterns of finite difference matrices, and dependency graphs. Assumption 2 of the present theorem guarantees that, for all $\ell \le d_a$, $\Delta_\ell^\tau \boldsymbol{\lambda}(z^{<t}, a^{<t}, \epsilon)$ spans $\mathbb{R}^{d_z}_{\mathbf{Ch}_\ell}$, which implies that the sparsity pattern of this finite difference is equal to $[\mathbf{Ch}_1, \ldots, \mathbf{Ch}_{d_a}] = G$

$$S = G \tag{145}$$

in the sense that $(i, j) \in S \iff G_{i,j} = 1$. Recall that $\hat{S}$ is the sparsity pattern of $\Delta^\tau \hat{\boldsymbol{\lambda}}(\mathbf{v}(z^{<t}), a^{<t}, \boldsymbol{\epsilon})$. Note that if $\hat{G}_{i,\ell} = 0$, then $\Delta^\tau \hat{\boldsymbol{\lambda}}(\mathbf{v}(z^{<t}), a^{<t}, \boldsymbol{\epsilon})_{i,\ell} = 0$ everywhere, and thus $(i, j) \notin \hat{S}$. Taking the contraposition, we get

$$\hat{S} \subset \hat{G} \tag{146}$$

in the sense that $(i, j) \in \hat{S} \implies \hat{G}_{i,j} = 1$.

We now proceed to demonstrate that all three statements of the present theorem holds. This is done by showing that all three assumptions of Lemma 18 are satisfied by exploiting the correspondence between them and Assumptions 2, 3 & 4 of the present theorem.

**Statement 1:** Assumption 1 of Lemma 18 directly holds for $\Lambda_{.,\ell}(\gamma) = \Delta_\ell^\tau \boldsymbol{\lambda}(z^{<t}, a^{<t}, \epsilon)$, for all $\ell$, by Assumption 2 of the present theorem. This implies that there exists a permutation $\sigma$ such that $\sigma(S) \subset \hat{S}$. Using (146) & (145), we have that

$$PG = \sigma(S) \subset \hat{S} \subset \hat{G}, \tag{147}$$

where $P$ is the permutation matrix associated with $\sigma$. This proves the first statement.

**Statement 2:** We will now show that Assumption 2 of Lemma 18 holds. Since $S = G$ and $\hat{S} \subset \hat{G}$, we have

$$|S| = ||G||_0 \tag{148}$$

$$|\hat{S}| \leq ||\hat{G}||_0. \tag{149}$$

By Assumption 3, $||\hat{G}||_0 \leq ||G||_0$. Thus, we have

$$|\hat{S}| \leq ||\hat{G}||_0 \leq ||G||_0 = |S|. \tag{150}$$

The above equation is precisely Assumption 2 of Lemma 18. Using $||\hat{G}||_0 \leq ||G||_0$ and (147), we can easily see that

$$PG = \hat{G}, \tag{151}$$

which proves the second statement.

**Statement 3:** We finally show that Assumption 3 of Lemma 18 holds. By Assumption 4 of the present theorem, we have that for all $p \in \{1, ..., d_z\}$, there is a subset $\mathcal{L} \subset \{1, ..., d_a\}$ such that

$$\bigcap_{\ell \in \mathcal{I}} \mathbf{Ch}_\ell = \{p\} \iff \left(\bigcap_{\ell \in \mathcal{L}} S_{.,\ell}\right) = \{p\}, \tag{152}$$

where the equivalence holds because $S = G$. We can thus apply Lemma 18 to conclude that $L$ is a permutation-scaling matrix, which is the third and final statement. ∎

A.3.4. PROOF OF THE COMBINED THEOREM (THM. 5)

Finally, we can prove Thm. 5, which was presented in the main text.

**Theorem 5 (Disentanglement via mechanism sparsity)** *Suppose we have two models as described in Sec. 2.1 with parameters $\theta = (\mathbf{f}, \boldsymbol{\lambda}, G)$ and $\hat{\theta} = (\hat{\mathbf{f}}, \hat{\boldsymbol{\lambda}}, \hat{G})$ representing the same distribution, i.e. $\mathbb{P}_{X^{\leq T}|a;\theta} = \mathbb{P}_{X^{\leq T}|a;\hat{\theta}}$ for all $a \in \mathcal{A}^T$. Suppose the assumptions of Thm. 4 hold and that*

1. *The sufficient statistic $\mathbf{T}$ is $d_z$-dimensional ($k = 1$) and is a diffeomorphism from $\mathcal{Z}$ to $\mathbf{T}(\mathcal{Z})$.*

2. *[**Sufficient time-variability**] There exist $\{(z_{(p)}, a_{(p)}, \tau_{(p)})\}_{p=1}^{||G^z||_0}$ belonging to their respective support such that*

$$\mathrm{span}\left\{D_z^{\tau_{(p)}} \boldsymbol{\lambda}(z_{(p)}, a_{(p)}) D_z \mathbf{T}(z_{(p)}^{\tau_{(p)}})^{-1}\right\}_{p=1}^{||G^z||_0} = \mathbb{R}_{G^z}^{d_z \times d_z},$$

*where $D_z^{\tau_{(p)}}$ and $D_z$ are the Jacobian operators with respect to $z^{\tau_{(p)}}$ and $z$, respectively.*

*Then, there exists a permutation matrix $P$ such that $PG^z P^\top \subset \hat{G}^z$.[8] Further assume that*

3. ***[Sufficient action-variability]*** *For all $\ell \in \{1, ..., d_a\}$, there exist $\{(z_{(p)}, a_{(p)}, \epsilon_{(p)}, \tau_{(p)})\}_{p=1}^{|\mathbf{Ch}_\ell^a|}$ belonging to their respective support such that*

$$\text{span} \left\{ \Delta_\ell^{\tau_{(p)}} \lambda(z_{(p)}, a_{(p)}, \epsilon_{(p)}) \right)_{p=1}^{|\mathbf{Ch}_\ell^a|} = \mathbb{R}^{d_z}_{\mathbf{Ch}_\ell^a} .$$

*Then $PG^a \subset \hat{G}^a$. Further assume that*

4. ***[Sparsity]*** $||\hat{G}||_0 \leq ||G||_0.$

*Then, $PG^z P^\top = \hat{G}^z$ and $PG^a = \hat{G}^a$. Further assume that*

5. ***[Graphical criterion]*** *For all $p \in \{1, ..., d_z\}$, there exist sets $\mathcal{I}, \mathcal{J} \subset \{1, ..., d_z\}$ and $\mathcal{L} \subset \{1, ..., d_a\}$ such that*

$$\left( \bigcap_{i \in \mathcal{I}} \mathbf{Pa}_i^z \right) \cap \left( \bigcap_{j \in \mathcal{J}} \mathbf{Ch}_j^z \right) \cap \left( \bigcap_{\ell \in \mathcal{L}} \mathbf{Ch}_\ell^a \right) = \{p\} .$$

*Then $\theta$ and $\hat{\theta}$ are permutation-equivalent, i.e. the model $\hat{\theta}$ is disentangled.*

*Proof.* This theorem is a combination of 21 & 22 and as such we are going to re-use most of the proofs content. The idea is to apply Lemma 19, showing the correspondence of assumptions.

**Correspondence of parameters.** First of all, since the assumptions of Thm. 4 hold along with assumption 1 & 2, we can get the master equation of the proof of theorem 21 and map to the corresponding abstract functions

$$L^\top \underbrace{D_z^\tau \lambda(z^{<t}, a^{<t}) D\mathbf{T}(z^\tau)^{-1}}_{\Lambda^{(1)}(\gamma)} L = \underbrace{D_z^\tau \hat{\lambda}(\mathbf{v}(z^{<t}), a^{<t}) D\mathbf{T}(\mathbf{v}(z^\tau))^{-1}}_{\hat{\Lambda}^{(1)}(\gamma)} . \tag{153}$$

Similarly with assumption 3, we get the master equation of the proof of theorem 22

$$L^\top \underbrace{\Delta^\tau \lambda(z^{<t}, a^{<t}, \boldsymbol{\epsilon})}_{\Lambda^{(2)}(\gamma)} = \underbrace{\Delta^\tau \hat{\lambda}(\mathbf{v}(z^{<t}), a^{t-1}, \boldsymbol{\epsilon})}_{\hat{\Lambda}^{(2)}(\gamma)} . \tag{154}$$

Let us introduce $S^{(1)}, \hat{S}^{(1)}, S^{(2)}, \hat{S}^{(2)}$ the sparsity patterns of respectively $\Lambda^{(1)}, \hat{\Lambda}^{(1)}, \Lambda^{(2)}, \hat{\Lambda}^{(2)}$. The same mapping between sparsity patterns and dependency matrices as in theorem 21 & 22 applies

$$S^{(1)} = G^z \tag{155}$$

$$\hat{S}^{(1)} \subset \hat{G}^z \tag{156}$$

$$S^{(2)} = G^a \tag{157}$$

$$\hat{S}^{(2)} \subset \hat{G}^a . \tag{158}$$

---

8. Given two binary matrices $M^1$ and $M^2$ with equal shapes, we say $M^1 \subset M^2$ when $M_{i,j}^1 = 1 \implies M_{i,j}^2 = 1$.

**Correspondence of assumptions.** Now that we identified the relevant parameters, we ready to show the correspondence between the assumptions of the present theorem and the four assumptions of Lemma 19. The assumptions of sufficient time- and action-variability respectively map to the assumptions of sufficient variability 1 and 2 of Lemma 19.

We will now show that the sparsity assumption of Lemma 19 holds. By the sparsity assumption of this theorem $||\hat{G}^z||_0 + ||\hat{G}^a||_0 \leq ||G^z||_0 + ||G^a||_0$. But we know from the previous identification between sparsity patterns and dependency graphs that $|\hat{S}^{(1)}| \leq ||\hat{G}^z||_0$, $|\hat{S}^{(2)}| \leq ||\hat{G}^a||_0$, $|S^{(1)}| = ||G^z||_0$, and $|S^{(2)}| = ||G^a||_0$, thus

$$|\hat{S}^{(1)}| + |\hat{S}^{(2)}| \leq ||\hat{G}^a||_0 + ||\hat{G}^z||_0 \tag{159}$$

$$\leq ||G^a||_0 + ||G^z||_0 \tag{160}$$

$$= |S^{(1)}| + |S^{(2)}|. \tag{161}$$

The above equation is precisely the sparsity assumption of Lemma 19.

Finally, the equality between graphs and sparsity patterns mean that the graphical criterion is the same between this theorem and Lemma 19.

We can thus apply Lemma 19 to conclude that $L$ is a permutation-scaling matrix. ∎

### A.4. Minor extensions of the theory

The experiments presented in Sec. 4 differed in minor ways from the theory presented in the main paper. In what follows, we explain how our theory can be extended to cover a wider range of models, including the one used in our experiments.

### A.4.1. IDENTIFIABILITY WHEN $\sigma^2$ IS LEARNED

In this section, we show how, by adding the extra assumption that $d_z < d_x$, we can adapt the argument in Equations (18) to (23) to work when the variance $\sigma^2$ of the additive noise is learned, i.e., $\theta := (\mathbf{f}, \boldsymbol{\lambda}, G, \sigma^2)$ instead of just $\theta := (\mathbf{f}, \boldsymbol{\lambda}, G)$.

Recall $Y^t := \mathbf{f}(Z^t)$ and that given an arbitrary $a \in \mathcal{A}^T$ and a parameter $\theta = (\mathbf{f}, \boldsymbol{\lambda}, G, \sigma^2)$, $\mathbb{P}_{Y^{\leq T}|a;\theta}$ is the conditional probability distribution of $Y^{\leq T}$ and $\mathbb{P}_{Z^{\leq T}|a;\theta}$ is the conditional probability distribution of $Z^{\leq T}$. Since now $\sigma^2$ is learned, we denote by $\mathbb{P}_{N^{\leq T};\sigma^2}$ the probability distribution of $N^{\leq T}$, the Gaussian noises with covariance $\sigma^2 I$. We now present the modified argument:

$$\mathbb{P}_{X^{\leq T}|a;\theta} = \mathbb{P}_{X^{\leq T}|a;\hat{\theta}} \tag{162}$$

$$\mathbb{P}_{Y^{\leq T}|a;\theta} * \mathbb{P}_{N^{\leq T};\sigma^2} = \mathbb{P}_{Y^{\leq T}|a;\hat{\theta}} * \mathbb{P}_{N^{\leq T};\hat{\sigma}^2} \tag{163}$$

$$\mathcal{F}(\mathbb{P}_{Y^{\leq T}|a;\theta} * \mathbb{P}_{N^{\leq T};\sigma^2}) = \mathcal{F}(\mathbb{P}_{Y^{\leq T}|a;\hat{\theta}} * \mathbb{P}_{N^{\leq T};\hat{\sigma}^2}) \tag{164}$$

$$\mathcal{F}(\mathbb{P}_{Y^{\leq T}|a;\theta})\mathcal{F}(\mathbb{P}_{N^{\leq T};\sigma^2}) = \mathcal{F}(\mathbb{P}_{Y^{\leq T}|a;\hat{\theta}})\mathcal{F}(\mathbb{P}_{N^{\leq T};\hat{\sigma}^2}) \tag{165}$$

$$\forall t \ \mathcal{F}(\mathbb{P}_{Y^{\leq T}|a;\theta})(t)e^{-\frac{\sigma^2}{2}t^\top t} = \mathcal{F}(\mathbb{P}_{Y^{\leq T}|a;\hat{\theta}})(t)e^{-\frac{\hat{\sigma}^2}{2}t^\top t} \tag{166}$$

$$\forall t \ \mathcal{F}(\mathbb{P}_{Y^{\leq T}|a;\theta})(t) = \mathcal{F}(\mathbb{P}_{Y^{\leq T}|a;\hat{\theta}})(t)e^{-\frac{\hat{\sigma}^2-\sigma^2}{2}t^\top t} \tag{167}$$

where (166) leverages the formula for the Fourier transform of a Gaussian measure. We now want to show that $\sigma^2 = \hat{\sigma}^2$ by contradiction. Assuming without loss of generality that $\hat{\sigma}^2 > \sigma^2$ we have

that $e^{-\frac{\hat{\sigma}^2-\sigma^2}{2}t^\top t}$ is the Fourier transform of a Gaussian distribution with mean zero and covariance $(\hat{\sigma}^2 - \sigma^2)I$. Thus, the left hand side is the Fourier transform of a probability measure concentrated on a $Td_z$-manifold while the right hand side is the Fourier transform of a convolution between two distributions, one of which has probability mass over all $\mathbb{R}^{Td_x}$. Since $d_z < d_x$, the left measure is not absolutely continuous with respect to the Lebesgue measure (on $\mathbb{R}^{Td_x}$) while the right one is. This is a contradiction since both measures should be equal. Thus, $\sigma^2 = \hat{\sigma}^2$. The rest of the proof of Thm. 4 follows through without modification.

### A.4.2. IDENTIFIABILITY WHEN SOME LEARNED PARAMETERS ARE INDEPENDENT OF THE PAST

Suppose we slightly modify the model of (2) to be

$$p(z_i^t \mid z^{<t}, a^{<t}) = h_i(z_i^t) \exp\{\mathbf{T}_i(z_i^t)^\top \boldsymbol{\lambda}_i(G_i^z \odot z^{<t}, G_i^a \odot a^{<t}) + \mathbf{T}_i^0(z_i^t)^\top \lambda_i^0 - \psi_i(z^{<t}, a^{<t})\}, \tag{168}$$

where $\mathbf{T}_i^0$ is the piece of the sufficient statistic with learnable natural parameters $\lambda_i^0$ which does not depend on $(z^{<t}, a^{<t})$. In that case, the learnable parameters of the model are $\theta = (\mathbf{f}, \boldsymbol{\lambda}, \lambda^0, G)$. We now show that the proof of Thm. 4 can be adapted to allow for this slightly more general model.

In the proof of Thm. 4, all steps until equation (36) do not depend on the specific form of $p(z^t | z^{<t}, a^{<t})$, thus they also apply to the more general model (168). We can now adapt (37) to get:

$$\sum_{i=1}^{d_z} \log h_i(z_i^t) + \mathbf{T}_i(z_i^t)^\top \boldsymbol{\lambda}_i(G_i^z \odot z^{<t}, G_i^a \odot a^{<t}) + \mathbf{T}_i^0(z_i^t)^\top \lambda_i^0 - \psi_i(z^{<t}, a^{<t}) \tag{169}$$

$$= \sum_{i=1}^{d_z} \log h_i(\mathbf{v}_i(z^t)) + \mathbf{T}_i(\mathbf{v}_i(z^t)))^\top \hat{\boldsymbol{\lambda}}_i(\hat{G}_i^z \odot \mathbf{v}(z^{<t}), \hat{G}_i^a \odot a^{<t}) + \mathbf{T}_i^0(\mathbf{v}_i(z^t))^\top \hat{\lambda}_i^0$$

$$- \hat{\psi}_i(\mathbf{v}(z^{<t}), a^{<t}) + \log |\det D\mathbf{v}(z^t)|.$$

In the following step of the proof, we evaluate the above equation at $(z^t, z_{(p)}, a_{(p)})$ and $(z^t, z_{(0)}, a_{(0)})$ and take their difference which gives

$$\sum_{i=1}^{d_z} \mathbf{T}_i(z_i^t)^\top [\boldsymbol{\lambda}_i(G_i^z \odot z_{(p)}, G_i^a \odot a_{(p)}) - \boldsymbol{\lambda}_i(G_i^z \odot z_{(0)}, G_i^a \odot a_{(0)})] - \psi_i(z_{(p)}, a_{(p)}) + \psi_i(z_{(0)}, a_{(0)})$$

$$= \sum_{i=1}^{d_z} \mathbf{T}_i(\mathbf{v}_i(z^t))^\top [\hat{\boldsymbol{\lambda}}_i(\hat{G}_i^z \odot \mathbf{v}(z_{(p)}), \hat{G}_i^a \odot a_{(p)}) - \hat{\boldsymbol{\lambda}}_i(\hat{G}_i^z \odot \mathbf{v}(z_{(0)}), \hat{G}_i^a \odot a_{(0)})] \tag{170}$$

$$- \hat{\psi}_i(\mathbf{v}(z_{(p)}), a_{(p)}) + \hat{\psi}_i(\mathbf{v}(z_{(0)}), a_{(0)}),$$

where the terms $\mathbf{T}_i^0(z_i^t)^\top \lambda_i^0$ and $\mathbf{T}_i^0(\mathbf{v}_i(z^t))^\top \hat{\lambda}_i^0$ disappear since they do not depend on $(z^{<t}, a^{<t})$, just like $\log h_i(z_i^t)$, $\log h_i(\mathbf{v}_i(z^t))$ and $\log |\det D\mathbf{v}(z^t)|$. Notice that (170) is identical to (38) from the proof of Thm. 4. All following steps are derived from this equation except for (57) which starts from (37), but it can be easily seen that the terms $\mathbf{T}_i^0(z_i^t)^\top \lambda_i^0$ and $\mathbf{T}_i^0(\mathbf{v}_i(z^t))^\top \hat{\lambda}_i^0$ will be absorbed in $c(z^t)$ like the other terms depending only on $z^t$. We, thus, conclude that Thm. 4 holds also for the slightly more general model specified in (168).

Since the proofs of Thm. 5, 21 & 22 rely on Thm. 4, this slight generalization applies to them as well.

In our experiments, we parameterize $p(z^t \mid z^{<t}, a^{<t})$ with the standard $(\mu, \sigma^2)$ and not with the natural parameters. Precisely, $\mu(z^{<t}, a^{<t})$ is modeled using a neural network while $\sigma^2$ is learned but does not depend on the past. The natural parameters $(\lambda_1, \lambda_2)$ of a Normal distribution can be written as a function of $\mu$ and $\sigma^2$:

$$[\lambda_1, \lambda_2] = \left[ \frac{\mu(z^{<t}, a^{<t})}{\sigma^2}, -\frac{1}{2\sigma^2} \right] . \tag{171}$$

We can see that $\lambda_2$ does not depend on $(z^{<t}, a^{<t})$ and, thus, the model we learn in practice is an instance of the slightly more general model (168).

### A.4.3. IDENTIFIABILITY WHEN $\lambda_i^0$ AT $t = 1$ DIFFERS FROM $\lambda_i^0$ AT $t > 1$

The extension of Sec. A.4.2 implicitly assumed that $\lambda_i^0$ is the same for every time steps $t$. Indeed, when taking the difference in (170), if $(z_{(0)}, a_{(0)})$ and $(z_{(p)}, a_{(p)})$ come from different time steps for which $\lambda_i^0$ have different values, they would not cancel each other.

This assumption is somewhat problematic in the case where $\lambda_i^0$ represents the variance, since we might expect the variance of the very first latent $\mathbb{V}[z_i^0]$ to be much larger than the subsequent conditional variances $\mathbb{V}[z_i^t \mid z^{<t}, a^{<t}]$. For example, this would be the case in an environment that is initialized randomly, but that follows nearly deterministic transitions.

To solve this issue, we allow only the very first $\lambda_i^0$ to be different from the subsequent ones, which are assumed to be all equal. In that case, we must modify the assumptions of sufficient variability so that all $(z_{(p)}, a_{(p)})$ cannot be selected from the very first time step.

### A.5. On the invertibility of the mixing function f

Throughout this work as well as many others (Hyvärinen and Morioka, 2016, 2017; Hyvärinen et al., 2019; Khemakhem et al., 2020a; Locatello et al., 2020; Klindt et al., 2021), it is assumed that the mixing function mapping the latent factors to the observation is a diffeomorphism from $\mathcal{Z}$ to $\mathcal{X}$. In this section, we briefly discuss the practical implications of this assumption.

Recall that a diffeomorphism is a differentiable bijective function with a differentiable inverse. We start by adressing the bijective part of the assumption. To understand it, we consider a plausible situation where the mapping $\mathbf{f}$ is not invertible. Consider the minimal example of Fig. 1 consisting of a tree, a robot and a ball. Assume that the ball can be hidden behind either the tree or the robot. Then, the mixing function $\mathbf{f}$ is not invertible because, given only the image, it is impossible to know whether the ball is behind the tree or the robot. Thus, this situation is not covered by our theory. Intuitvely, one could infer, at least approximately, where the ball is hidden based on previous time frames. Allowing for this form of occlusion is left as future work.

We believe the differentiable part of this assumption is only a technicality that could probably be relaxed to being piecewise differentiable. Our experiments were performed with data generated with a piecewise linear $\mathbf{f}$, which in not differentiable only on a set of (Lebesgue) measure zero, but this was not an issue in practice.

### A.6. Contrasting with the assumptions of iVAE

Recall from Sec. 3 that the most significant distinction between the theory of (Khemakhem et al., 2020a) and ours is how *permutation-identifiability* is obtained: Thm. 2 & 3 from iVAE shows that if the assumptions of their Thm. 1 (which is almost the same as our Thm. 4) are satisfied and $\mathbf{T}_i$ has dimension $k > 1$ or is non-monotonic, then the model is not just linearly, but permutation-identifiable. In contrast, our theory covers the case where $k = 1$ and $\mathbf{T}_i$ is monotonic, like in the Gaussian case with fixed variance. Interestingly, Khemakhem et al. (2020a) mentioned this specific case as a counterexample to their theory in their Prop. 3. The extra power of our theory comes from the extra *structure* in the dependencies of the latent factors coupled with sparsity regularization.

We now argue that the assumptions of iVAE for disentanglement are less plausible in an environment such as the one of Fig. 1. Assuming the latent factors are Gaussian, the variability assumption of Thm. 4 combined with $k > 1$ requires the variance to vary sufficiently, which is implausible in such a nearly deterministic environment. Assuming $k = 1$ with non-monotonic $\mathbf{T}_i$ implies the conditional mean of $Z^t$ does not depend on the past (since the sufficient statistic corresponding to the mean of a Gaussian is monotonous), which is also implausible in this environment. On the other hand, the case $k = 1$ with monotonic $\mathbf{T}_i$ of Thm. 5 is well suited for the situation. Indeed, again in the Gaussian case, this would amount to predicting only the mean of the future positions of object. That being said, we also believe practical applications of these ideas will most likely require a combination of different identifiability results. How to formally combined these results is left as future work.

### A.7. Illustrating the sufficient variability assumptions

In this section, we construct a simple example based on the situation depicted in Fig. 1 that illustrates a simple case where the assumption of variability is satisfied. Suppose we have a tree and a robot with position $T^t$ and $R^t$, respectively (there is no ball). Suppose the transition model $p((T^t, R^t) \mid (T^{<t}, R^{<t}), A^{<t})$ is Gaussian with variance fixed to a very small value (nearly deterministic) and a mean given by functions $\mu_T(T^{t-1})$ and $\mu_R(T^{t-1}, R^{t-1}, R^{t-2}, A^{t-1})$ specified by

$$\mu_T(T^{t-1}) := T^{t-1} \tag{172}$$

$$\mu_R(T^{t-1}, R^{t-1}, R^{t-2}, A^{t-1}) := \begin{cases} T^{t-1} - \delta, & \text{if } R^{t-1} < T^{t-1} \text{ and } T^{t-1} - \delta < R^{t-1} + \Delta^{t-1} \\ T^{t-1} + \delta, & \text{if } T^{t-1} < R^{t-1} \text{ and } R^{t-1} + \Delta^{t-1} < T^{t-1} + \delta \\ R^{t-1} + \Delta^{t-1}, & \text{otherwise} \end{cases} \tag{173}$$

where $\Delta^{t-1} := R^{t-1} - R^{t-2} + A^{t-1}$ is the expected change of position of the robot given it does not hit the tree. Note that the first and second case in (173) correspond to when the robot hit the tree from the left and from the right, respectively, and $\delta$ is the distance between the center of the tree and the center of the robot when they both touch each other. The action $A^{t-1}$ thus controls the speed and direction of the robot. Notice that, here, the graph $G^z$ and $G^a$ are given by

$$G^z := \begin{bmatrix} 1 & 0 \\ 1 & 1 \end{bmatrix} \tag{174}$$

$$G^a := \begin{bmatrix} 0 \\ 1 \end{bmatrix} \tag{175}$$

We now show that the assumptions of sufficient variability of Thm. 5 hold in this case (Assumptions 1 & 2). First, recall that for a Gaussian distribution with fixed variance, the natural parameter is given by $\lambda = \frac{\mu}{\sigma}$. We thus have

$$\lambda(T^{t-1}, R^{t-1}, R^{t-2}, A^{t-1}) := \frac{1}{\sigma_z} \begin{bmatrix} \mu_T(T^{t-1}) \\ \mu_R(T^{t-1}, R^{t-1}, R^{t-2}, A^{t-1}) \end{bmatrix} . \tag{176}$$

Without loss of generality, we fix $\sigma_z = 1$. We start by showing time-sufficient variability. Note that taking the Jacobian of (176) with respect to $(T^{t-1}, R^{t-1})$ we obtain

$$D_z^{t-1}\lambda(T^{t-1}, R^{t-1}, R^{t-2}, A^{t-1}) = \begin{cases} \begin{bmatrix} 1 & 0 \\ 1 & 0 \end{bmatrix}, & \text{if } R^{t-1} < T^{t-1} \text{ and } T^{t-1} - \delta < R^{t-1} + \Delta^{t-1} \\[2em] \begin{bmatrix} 1 & 0 \\ 1 & 0 \end{bmatrix}, & \text{if } T^{t-1} < R^{t-1} \text{ and } R^{t-1} + \Delta^{t-1} < T^{t-1} + \delta \\[2em] \begin{bmatrix} 1 & 0 \\ 0 & 2 \end{bmatrix}, & \text{otherwise.} \end{cases} \tag{177}$$

We can do the same thing but differentiating with respect to $(T^{t-2}, R^{t-2})$:

$$D_z^{t-2}\lambda(T^{t-1}, R^{t-1}, R^{t-2}, A^{t-1}) = \begin{cases} \begin{bmatrix} 0 & 0 \\ 0 & 0 \end{bmatrix}, & \text{if } R^{t-1} < T^{t-1} \text{ and } T^{t-1} - \delta < R^{t-1} + \Delta^{t-1} \\[2em] \begin{bmatrix} 0 & 0 \\ 0 & 0 \end{bmatrix}, & \text{if } T^{t-1} < R^{t-1} \text{ and } R^{t-1} + \Delta^{t-1} < T^{t-1} + \delta \\[2em] \begin{bmatrix} 0 & 0 \\ 0 & -1 \end{bmatrix}, & \text{otherwise.} \end{cases} \tag{178}$$

We can see that the first and third matrix of (177) together with the third matrix of (178) form a basis of the space $\mathbb{R}^{2\times2}_{G^z}$, which proves sufficient time-variability.

We now prove sufficient action-variability. We can compute the following partial difference with respect to $A^{t-1}$ (for a sufficiently small step $\epsilon \in \mathbb{R}$):

$$\Delta_1^{t-1}\boldsymbol{\lambda}(T^{t-1}, R^{t-1}, R^{t-2}, A^{t-1}; \epsilon) = \begin{cases} \begin{bmatrix} 0 \\ 0 \end{bmatrix}, & \text{if } R^{t-1} < T^{t-1} \text{ and } T^{t-1} - \delta < R^{t-1} + \Delta^{t-1} \\[2em] \begin{bmatrix} 0 \\ 0 \end{bmatrix}, & \text{if } T^{t-1} < R^{t-1} \text{ and } R^{t-1} + \Delta^{t-1} < T^{t-1} + \delta \\[2em] \begin{bmatrix} 0 \\ \epsilon \end{bmatrix}, & \text{otherwise.} \end{cases} \tag{179}$$

Clearly, the last vector in (179) spans $\mathbb{R}^2_{\mathbf{Ch}_1^a}$ since, recall, $\mathbf{Ch}_1^a = \{2\}$ (the robot position $R$ is the second coordinate).

## A.8. Derivation of the ELBO

In this section, we derive the evidence lower bound presented in Sec. 2.6.

$$\log p(x^{\leq T} \mid a^{<T}) = \tag{180}$$

$$\mathbb{E}_{q(z^{\leq T}|x^{\leq T}, a^{<T})} \left[ \log \frac{q(z^{\leq T} \mid x^{\leq T}, a^{<T})}{p(z^{\leq T} \mid x^{\leq T}, a^{<T})} \right. \tag{181}$$

$$\left. + \log \frac{p(z^{\leq T}, x^{\leq T} \mid a^{<T})}{q(z^{\leq T} \mid x^{\leq T}, a^{<T})} \right] \tag{182}$$

$$\geq \mathbb{E}_{q(z^{\leq T}|x^{\leq T}, a^{<T})} \left[ \log \frac{p(z^{\leq T}, x^{\leq T} \mid a^{<T})}{q(z^{\leq T} \mid x^{\leq T}, a^{<T})} \right] \tag{183}$$

$$= \mathbb{E}_{q(z^{\leq T}|x^{\leq T}, a^{<T})} \left[ \log p(x^{\leq T} \mid z^{\leq T}, a^{<T}) \right] \tag{184}$$

$$- KL(q(z^{\leq T} \mid x^{\leq T}, a^{<T}) || p(z^{\leq T} \mid a^{<T})) \tag{185}$$

where the inequality holds because the term at (181) is a Kullback-Leibler divergence, which is greater or equal to 0. Notice that

$$p(x^{\leq T} \mid z^{\leq T}, a^{<T}) = p(x^{\leq T} \mid z^{\leq T}) = \prod_{t=1}^{T} p(x^t \mid z^t). \tag{186}$$

Recall that we are considering a variational posterior of the following form:

$$q(z^{\leq T} \mid x^{\leq T}, a^{<T}) := \prod_{t=1}^{T} q(z^t \mid x^t). \tag{187}$$

Equations (186) & (187) allow us to rewrite the term in (184) as

$$\sum_{t=1}^{T} \mathbb{E}_{Z^t \sim q(\cdot|x^t)} \left[ \log p(x^t \mid Z^t) \right] \tag{188}$$

Notice further that

$$p(z^{\leq T} \mid a^{<T}) = \prod_{t=1}^{T} p(z^t \mid z^{<t}, a^{<t}). \tag{189}$$

Using (187) & (189), the KL term (185) can be broken down as a sum of KL as:

$$\sum_{t=1}^{T} \mathop{\mathbb{E}}_{Z^{<t} \sim q(\cdot \mid x^{<t})} KL(q(Z^t \mid x^t) || p(Z^t \mid Z^{<t}, a^{<t})) \tag{190}$$

Putting all together yields the desired ELBO:

$$\log p(x^{\leq T} \mid a^{<T}) \geq \sum_{t=1}^{T} \mathop{\mathbb{E}}_{Z^t \sim q(\cdot \mid x^t)} [\log p(x^t \mid Z^t)] \tag{191}$$
$$- \mathop{\mathbb{E}}_{Z^{<t} \sim q(\cdot \mid x^{<t})} KL(q(Z^t \mid x^t) || p(Z^t \mid Z^{<t}, a^{<t})).$$

## Appendix B. Experiments

### B.1. Synthetic datasets

We now provide a detailed description of the synthetic datasets used in experiments of Sec. 4.

For all experiments, the dimensionality of $X^t$ is $d_x = 20$ and the ground-truth $\mathbf{f}$ is a random neural network with three hidden layers of 20 units with Leaky-ReLU activations with negative slope of 0.2. The weight matrices are sampled according to a 0-1 Gaussian distribution and, to make sure $\mathbf{f}$ is injective as assumed in all theorems of this paper, we orthogonalize its columns. Inspired by typical weight initialization in NN (Glorot and Bengio, 2010), we rescale the weight matrices by $\sqrt{\frac{2}{1+0.2^2}} \sqrt{\frac{2}{d_{in}+d_{out}}}$. The standard deviation of the Gaussian noise added to $\mathbf{f}(z^t)$ is set to $\sigma = 10^{-2}$ throughout. All datasets consist of 1 million examples.

We now present the different choices of ground-truth $p(z^t \mid z^{<t}, a^{<t})$ we explored in our experiments. In all cases considered(except the experiment with $k = 2$ of Fig. 9), it is a Gaussian with covariance $0.0001I$ independent of $(z^{<t}, a^{<t})$ and a mean given by some function $\mu(z^{t-1}, a^{t-1})$ carefully chosen to satisfy the assumptions of Thm. 5. Notice that we hence are in the case where $k = 1$ which is not covered by the theory of Khemakhem et al. (2020a). We suppose throughout that $d_z = d_a = 10$. In all *time-sparsity* experiments, sequences have length $T = 2$. In *action-sparsity* experiments, the value of $T$ has no consequence since we assume there is no time dependence.

**Temporal sparsity with diagonal dependencies (Fig. 3).** In this dataset, each $Z_i^t$ has only $Z_i^{t-1}$ as parent. This trivially satisfies the graphical criterion of Thm. 5. The mean function is given by

$$\mu(z^{t-1}, a^{t-1}) := z^{t-1} + 0.5 \sin(z^{t-1}),$$

where the sin function is applied element-wise. Notice that no auxiliary variables are required.

**Temporal sparsity with triangular dependencies (Fig. 3).** We consider a case where the graphical criterion of Thm. 5 is satisfied non-trivially. Let

$$G^z := \begin{pmatrix} 1 & & & & \\ 1 & 1 & & & \\ \vdots & & \ddots & & \\ 1 & & & 1 & \\ 1 & 1 & \dots & 1 & 1 \end{pmatrix} \tag{192}$$

be the adjacency matrix between $Z^t$ and $Z^{t-1}$. The $i$th row of $G^z$, denoted by $G_i^z$, corresponds to the parents of $Z_i^t$. Notice that this connectivity matrix has no 2-cycles and all self-loops are present. Thus, by Prop. 6, it satisfies the graphical criterion. The mean function in this case is given by

$$\mu(z^{t-1}, a^{t-1}) := z^{t-1} + 0.5 \begin{bmatrix} G_1^z \cdot \sin(\frac{3}{\pi} z^{t-1}) \\ G_2^z \cdot \sin(\frac{4}{\pi} z^{t-1} + 1) \\ \vdots \\ G_{d_z}^z \cdot \sin(\frac{d_z+2}{\pi} z^{t-1} + d_z - 1) \end{bmatrix}, \tag{193}$$

where, the $\sin$ function is applied element-wise, the $\cdot$ is the dot product between two vectors and the summation in the $\sin$ function is broadcasted. Once again, the various frequencies and phases in the $\sin$ functions ensures the sufficient time-variability assumption of Thm. 5 is satisfied.

**Temporal sparsity with triangular dependencies and insufficient variability (Fig 8).** This dataset has the same ground truth adjacency matrix as in (192), but a different transition function that does not satisfy the assumption of sufficient time-variability. We sampled a transition matrix $W$ with independent Normal 0-1 entries. The transition function is thus

$$\mu(z^{t-1}, a^{t-1}) := z^{t-1} + 0.5(G^z \odot W)z^{t-1}. \tag{194}$$

**Temporal sparsity with graphical criterion violation (Fig. 8).** In this dataset, the mean function is the same as the one given in (193) except for the adjacency matrix which does not satisfy the graphical criterion and is given by

$$G^z := \begin{pmatrix} \mathbb{I}_{\frac{1}{2}d_z \times \frac{1}{2}d_z} & \\ & \mathbb{I}_{\frac{1}{2}d_z \times \frac{1}{2}d_z} \end{pmatrix}, \tag{195}$$

where $\mathbb{I}_{\frac{1}{2}d_z \times \frac{1}{2}d_z}$ is the $\frac{1}{2}d_z \times \frac{1}{2}d_z$ matrix filled with ones.

**Temporal sparsity with $k = 2$ (Fig. 9).** This dataset has the lower triangular adjacency matrix of (192) and the same mean function of (193), but the variance of $z^t$ (we assume diagonal covariance) depends on $z^{t-1}$ via

$$\sigma^2(z^{t-1}, a^{t-1}) := \frac{1}{10d_z} \begin{bmatrix} \exp\left(G_1^z \cdot \cos(\frac{3}{\pi} z^{t-1})\right) \\ \exp(G_2^z \cdot \cos(\frac{4}{\pi} z^{t-1} + 1)) \\ \vdots \\ \exp(G_{d_z}^z \cdot \cos(\frac{d_z+2}{\pi} z^{t-1} + d_z - 1)) \end{bmatrix}. \tag{196}$$

**Action sparsity with diagonal dependencies (Fig. 3).** In this setting, $d_a = d_x$ and the connectivity matrix between $A^{t-1}$ and $Z^t$ is diagonal, which trivially implies that the graphical criterion of Thm. 5 is satisfied. The mean function is given by

$$\mu(z^{t-1}, a^{t-1}) := \sin(a^{t-1}),$$

where $\sin$ is applied element-wise. Moreover, the components of the action vector $a^{t-1}$ are sampled independently and uniformly between $-2$ and $2$. The same sampling scheme is used for all following datasets.

**Action sparsity with double diagonal dependencies (Fig. 3).** We consider a case where the graphical criterion of Thm. 5 is satisfied non-trivially. Let

$$G^a := \begin{pmatrix} 1 & & & & 1 \\ 1 & 1 & & & \\ & 1 & \ddots & & \\ & & \ddots & 1 & \\ & & & 1 & 1 \end{pmatrix} \tag{197}$$

be the adjacency matrix between $A^{t-1}$ and $Z^t$. The $i$th row, denoted by $G_i^a$, corresponds to parents of $Z_i^t$ in $A^{t-1}$. Note that it is analogous to graph depicted in Fig. 2, which satisfies the graphical criterion. The mean function is given by

$$\mu(z^{t-1}, a^{t-1}) := \begin{bmatrix} G_1^a \cdot \sin(\frac{3}{\pi} a^{t-1}) \\ G_2^a \cdot \sin(\frac{4}{\pi} a^{t-1} + 1) \\ \vdots \\ G_{d_z}^a \cdot \sin(\frac{d_z+2}{\pi} a^{t-1} + d_z - 1) \end{bmatrix}, \tag{198}$$

which is analogous to (193).

**Action sparsity with double diagonal dependencies and insufficient variability (Fig. 8).** This dataset has the same ground truth adjacency matrix as the above dataset (197), but a different transition function which does not satisfy the assumption of sufficient variability. We sampled a matrix $W$ with independent Normal 0-1 entries. The mean function is thus

$$\mu(z^{t-1}, a^{t-1}) := (G^a \odot W) a^{t-1}. \tag{199}$$

**Action sparsity with graphical criterion violation (Fig 8).** This dataset does not satisfy the graphical criterion. The mean function is the same as (198), but its ground-truth graph $G^a$ is given by

$$G^a := \begin{pmatrix} \mathbb{1}_{2\times 2} & & & \\ & \mathbb{1}_{2\times 2} & & \\ & & \ddots & \\ & & & \mathbb{1}_{2\times 2} \end{pmatrix}, \tag{200}$$

where $\mathbb{1}_{2\times 2}$ is the $2 \times 2$ matrix filled with ones.

**Action sparsity with** $k = 2$ **(Fig. 9).** This dataset has the "double diagonal" adjacency matrix of (197) and the same mean function of (198), but the variance of $z^t$ (we assume diagonal covariance) depends on $a^{t-1}$ via

$$\sigma^2(z^{t-1}, a^{t-1}) := \frac{1}{10d_a} \begin{bmatrix} \exp\left(G_1^a \cdot \cos(\frac{3}{\pi} a^{t-1})\right) \\ \exp(G_2^a \cdot \cos(\frac{4}{\pi} a^{t-1} + 1)) \\ \vdots \\ \exp(G_{d_z}^a \cdot \cos(\frac{d_z+2}{\pi} a^{t-1} + d_z - 1)) \end{bmatrix}. \tag{201}$$

**B.2. Implementation details of our regularized VAE approach**

**Learned mechanisms.** Every coordinate $z_i$ of the latent vector has its own mechanism $\hat{p}(z_i^t \mid z^{<t}, a^{<t})$ that is Gaussian with mean outputted by $\hat{\mu}_i(z^{t-1}, a^{t-1})$ (a multilayer perceptron with 5 layers of 512) and a learned variance which does not depend on the previous time steps. Strictly speaking, having a learned variance that does not depend on the pas is not covered by the theory presented in the main paper, but App. A.4.2 extends it to this slightly more general case. For learning, we use the typical parameterization of the Gaussian distribution with $\mu$ and $\sigma^2$ and not its exponential family parameterization. Details about how this interacts with our theory can be found in Sec. A.4.2. Throughout, the dimensionality of $Z^t$ in the learned model always match the dimensionality of the ground-truth (same for baselines). Learning the dimensionality of $Z^t$ is left for future work.

**Prior of $Z^1$ in time-sparsity experiments.** In *time-sparsity* experiments, the prior of the first latent $\hat{p}(Z^1)$ (when $t = 1$) is modelled separately as a Gaussian with learned mean and learned diagonal covariance. Note that this learned covariance at time $t = 1$ is different from the subsequent learned conditional covariance at time $t > 1$. How this subtle point interacts with our theory is discussed in App. A.4.3.

**Learned graphs $\hat{G}^z$ and $\hat{G}^a$.** As explained in Sec. 2.6, to allow for gradient-based optimization, each edge $\hat{G}_{i,j}$ is viewed as a Bernoulli random variable with probability of success sigmoid$(\gamma_{i,j})$, where $\gamma_{i,j}$ is a learned parameter. The gradient of the loss with respect to the parameter $\gamma_{i,j}$ is estimated using the Gumbel-Softmax Gradient estimator (Jang et al., 2017; Maddison et al., 2017). We found that initializing the parameters $\gamma_{i,j}$ to a large value such that the probability of sampling all edge is almost one improved performance. In *time-sparsity* experiments, there is no action so $\hat{G}^a$ is fixed to 0, i.e. it is not learned. Analogously, in *action-sparsity* experiments, there is no temporal dependence so $\hat{G}^z$ is fixed to 0. In all figures, whenever the regularization coefficient is set to zero, the corresponding adjacency matrix is frozen so that all edges remain active.

**Encoder/Decoder.** In all experiments, including baselines, both the encoder and the decoder is modelled given by a neural network with 6 fully connected hidden layers of 512 units with LeakyReLU activation with negative slope 0.2. For all VAE-based methods, the encoder outputs the mean and a diagonal covariance. Moreover, $p(x|z)$ has a *learned* isotropic covariance $\sigma^2 I$. Note that $\sigma^2 I$ corresponds to the covariance of the independent noise $N^t$ in the equation $X^t = \mathbf{f}(Z^t) + N^t$. The theory presented in the main paper assumes $\sigma^2$ fixed, but Sec. A.4.1 shows how our theory can be adapted to deal with a learned $\sigma^2$, assuming $d_z < d_x$.

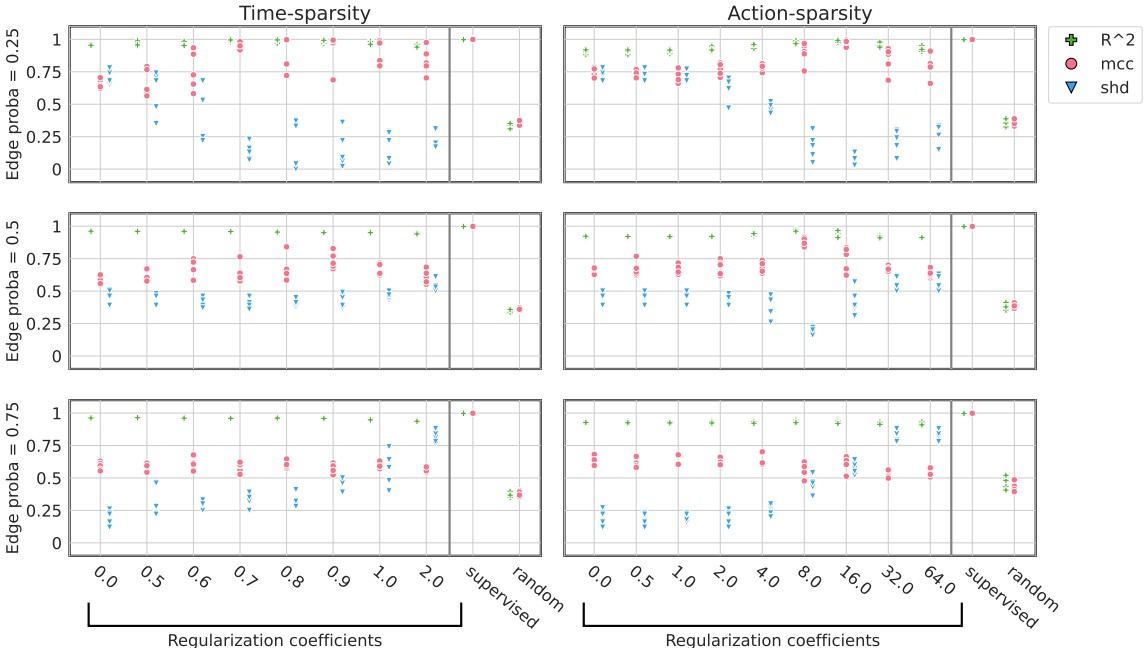

Figure 5: **Randomly sampled graphs.** The data generating process is the same as the one used for the datasets of Fig. 3 with non-diagonal graphs, except that, here, the graphs are sampled randomly. The different rows correspond to different probability of sampling an edge (first row = sparsest graphs). The edges are sampled independently and the edges on the diagonal of the adjacency matrix are included with probability one. Since the graphs are sampled randomly, we do not know if they satisfy the graphical criterion of Thm. 5. Nevertheless, regularization improves MCC (sometimes even more so than in Fig. 3), except for the denser graphs (with edge probability of 0.75), which is explained by the fact that the graphical criterion is less likely to be satisfied for denser graphs.

### B.3. Additional experiments

This section presents additional experiments with (i) diverse randomly sampled graphs (Fig. 5), (ii) different levels of noise on the latents (Fig. 6) and (iii) different levels of noise on the observations (Fig. 7).

### B.4. Experiments that violate assumptions

Fig. 8 & 9 show experiments on datasets that do not satisfy the assumptions of our theory. Fig. 8 shows data violating either the sufficient variability assumption or the graphical criterion. Fig. 9 shows data with a sufficient statistic $\mathbf{T}_i$ of dimension $k = 2$, thus violating the first assumption of Thm. 5. The only dataset that does not show an improved performance with regularization is the time-sparsity data that has insufficient variability. In all other datasets, regularization improves MCC, although by a smaller margin than when assumptions are met. As suggested by Thm. 5, when sufficient variability holds we can learn the graph, even if the graphical criterion does not hold.

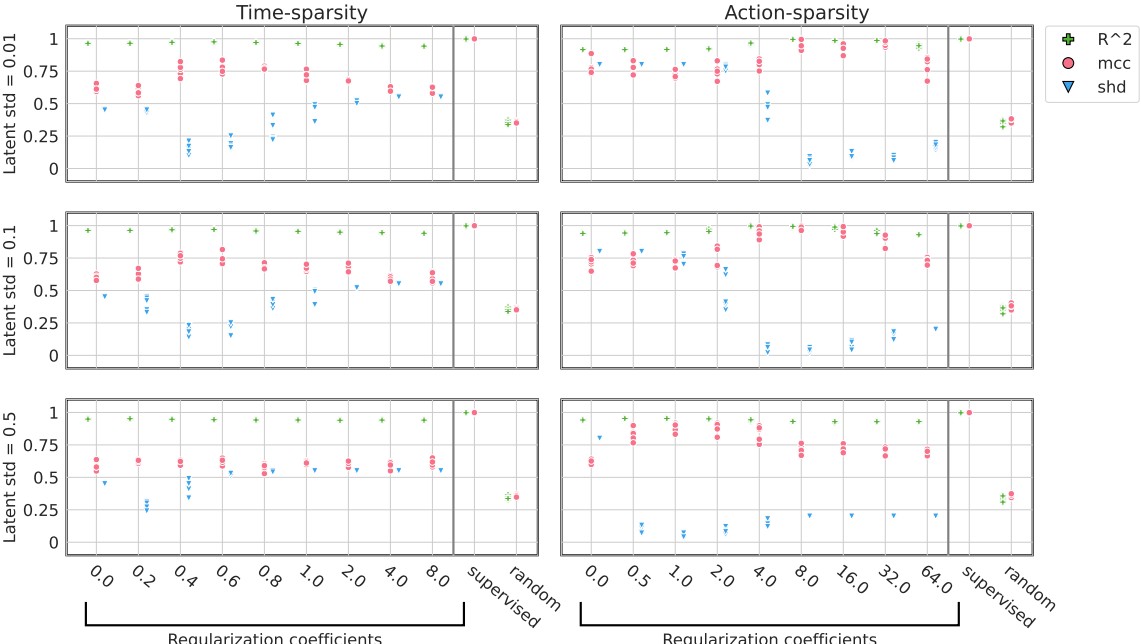

Figure 6: **Varying the variance of** $Z^t \mid Z^{t-1}$**.** The identifiability theory applies for any value of noise level on $Z^t$, but we want to investigate how this parameter affects learning. We consider standard deviations of 0.01, 0.1 and 0.5; the former being the noise-level used throughout our experiments. Our approach performs well everywhere except on the time-sparsity dataset for the maximal standard deviation of 0.5. Given that the initial latent is sampled from a Normal$(0, I)$, we consider a std of 0.5 to be very high.

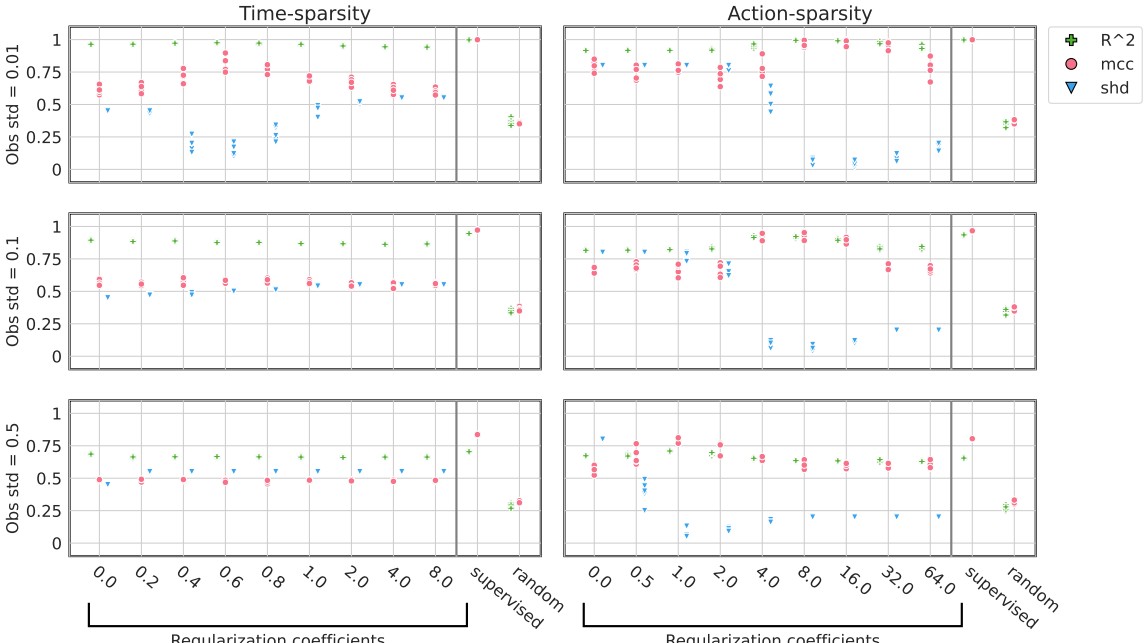

Figure 7: **Varying the variance of $X^t \mid Z^t$, i.e. $\sigma^2$.** Again, the identifiability theory applies for any value of noise level on $X^t$, but we want to investigate how this parameter affects learning. We consider standard deviations of 0.01, 0.1 and 0.5; the former being the noise-level used through-out our experiments. We see that the time-sparsity experiment suffers from higher noise level but not the action-sparsity dataset. This gap in performance might be explained by a worse sample complexity due to noisier data. It is also possible that our simple choice of approximate posterior $q(z^{\leq T} \mid x^{\leq T}, a^{<T})$ is a bad one when the noise is greater. Investigating these questions is left as future work.

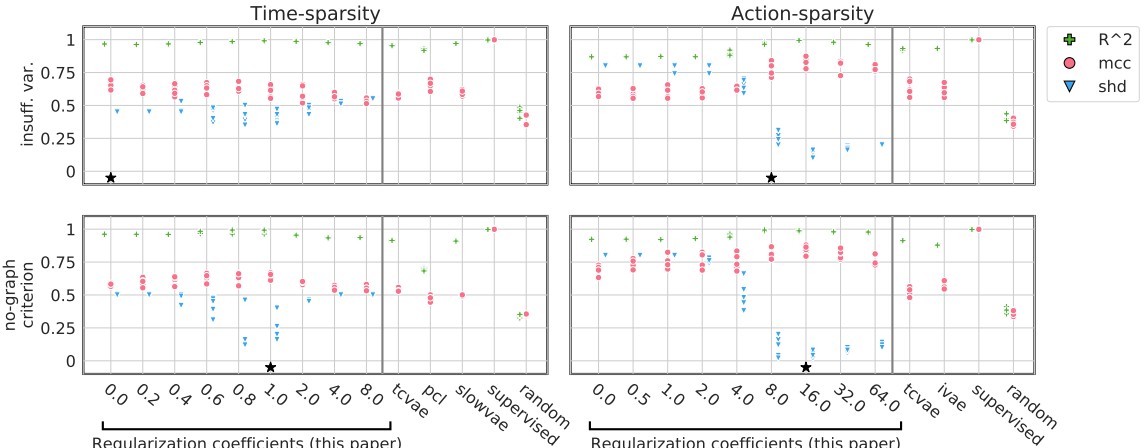

Figure 8: **Violating sufficient variability or graphical criterion.** The first row corresponds to a dataset that does not satisfy the sufficient variability assumption ($\mu(z^{t-1}, a^{t-1})$ is linear) while the second row does not satisfy the graphical criterion (the graphs have a block-diagonal structure). For more details on the synthetic datasets, see App. B.1. The black star indicates which regularization parameter is selected by our filtered UDR procedure (see App. B.7). For $R^2$ and MCC, higher is better. For SHD, lower is better.

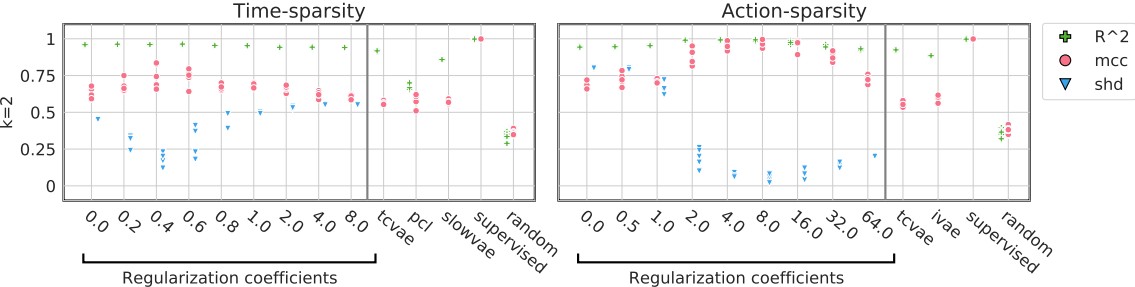

Figure 9: **Dataset with sufficient statistics $\mathbf{T}_i$ of dimension $k = 2$.** This is a violation of the first assumption of Theorem 5. The dataset is similar to Fig. 3, but the variance of $Z^t$ depends on $Z^{t-1}$. The adjacency matrix of the causal graph is non-diagonal. For more details about this dataset, see App. B.1. For $R^2$ and MCC, higher is better. For SHD, lower is better.

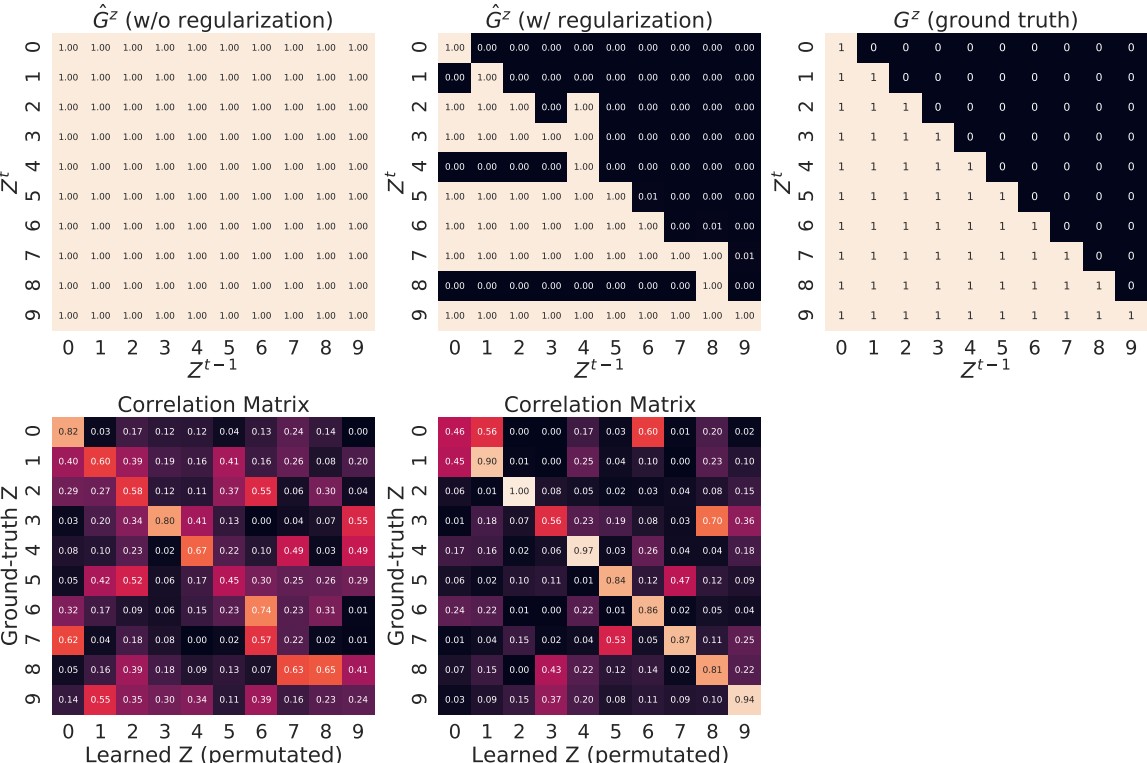

Figure 10: Example of a learned matrix on the "Time-sparsity with non-diagonal graph" dataset. Top row are adjacency matrices $G^z$ and bottom row are Pearson correlation matrices between the ground-truth and the learned representation. Left column corresponds to our approach without regularization. Middle column is our approach with the regularization coefficient selected by our filtered UDR procedure (App. B.7). The top right is the ground-truth adjacency matrix. Note that both the learned graphs and correlation matrices have been permutated to maximize MCC.

### B.5. Visualizing learned graphs

Fig. 10 & 11 shows examples of learned adjacency matrix $\hat{G}^z$ and $\hat{G}^a$ together with their Pearson correlation matrices between ground-truth and learned representations. We can see how adding mechanism sparsity regularization improves disentanglement. The learned adjacency matrix is not is not exactly equal to the ground-truth, but is reasonably close.

### B.6. Baselines

In synthetic experiments of Sec. 4, all methods used a minibatch size of 1024 and the same encoder and decoder architecture: A MLP with 6 layers of 512 units with LeakyReLU activations (negative slope of 0.2). We tuned manually the learning rate of each method to ensure proper convergence. For VAE-based methods, i.e. TCVAE, SlowVAE and iVAE, we are always choosing $p(x|z)$ Gaussian with a covariance $\sigma^2 I$ and learn $\sigma^2$.

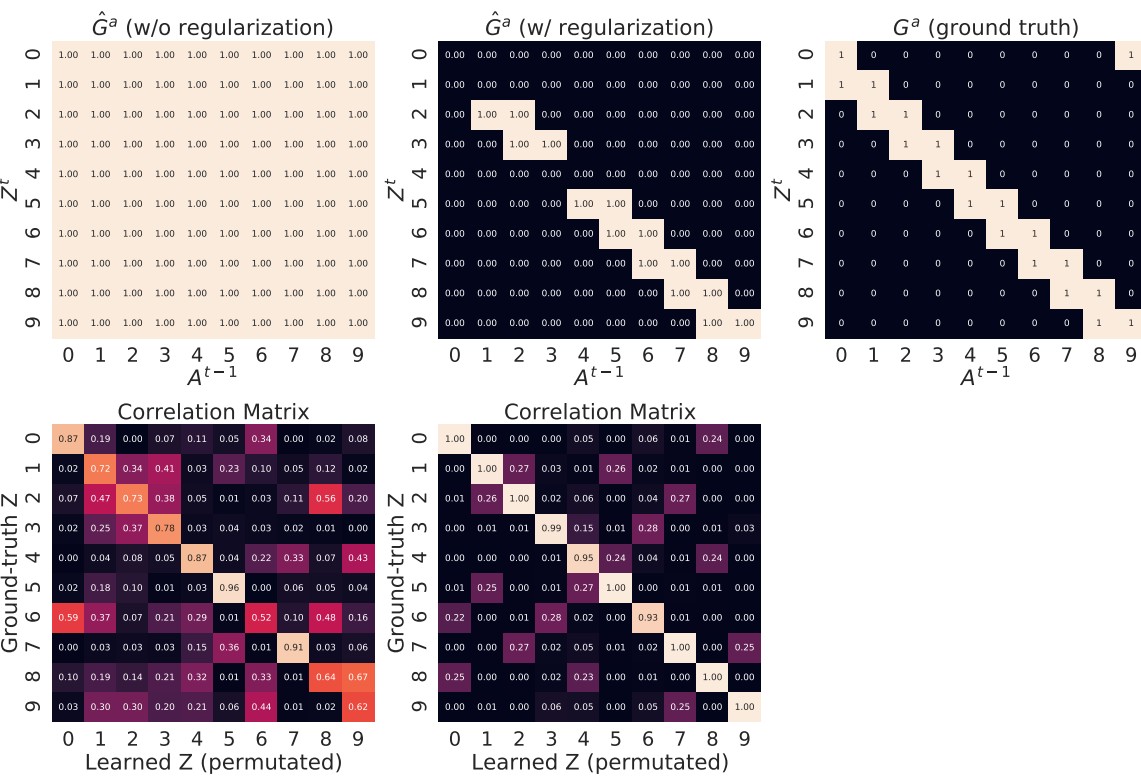

Figure 11: Example of a learned matrix on the "Action-sparsity with non-diagonal graph" dataset. Top row are adjacency matrices $G^a$ and bottom row are Pearson correlation matrices between the ground-truth and the learned representation. Left column corresponds to our approach without regularization. Middle column is our approach with the regularization coefficient selected by our filtered UDR procedure (App. B.7). The top right is the ground-truth adjacency matrix. Note that both the learned graphs and correlation matrices have been permutated to maximize MCC.

$\beta$**-TCVAE.**  We used the implementation provided in the original paper by Chen et al. (2018) which is available at `https://github.com/rtqichen/beta-tcvae`. We used a learning rate of 1e-4.

**iVAE.**  We used the implementation available at `https://github.com/ilkhem/icebeem` from Khemakhem et al. (2020a). In it, the mean of the prior $p(z|a)$ is fixed to zero while its diagonal covariance is allowed to depend on $a$ through an MLP. We change this to allow the mean to also depend on $a$ through the neural network (with 5 layers and width 512). We also lower bounded its variance as well as the variance of $q(z \mid x, a)$ to improve the stability of learning. In the original implementation, the covariance of $p(x|z)$ was not learned. We found that learning it (analogously to what we do in our method) improved performance. We used a learning rate of 1e-4.

**SlowVAE.**  We used the implementation provided in `https://github.com/bethgelab/slow_disentanglement` (Klindt et al., 2021). Like for other VAE-based methods, we modelled $p(x|z)$ as a Gaussian with covariance $\sigma^2 I$ and learned $\sigma^2$.

**PCL.**  We used the implementation provided here: `https://github.com/bethgelab/slow_disentanglement/tree/baselines`. PCL (Hyvarinen and Morioka, 2017) stands for "permutation contrastive learning" and works as follows: Given sequential data $\{X^t\}_{t=1}^T$, PCL trains a regression function $r((x', x))$ to discriminate between pairs of adjacent observations (positive pairs) and randomly matched pairs (negative pairs). The regression function has the form

$$r((x, x')) = \sum_{i=1}^{d_z} B_i(h_i(x), h_i(x')),\tag{202}$$

where $h : \mathbb{R}^{d_x} \to \mathbb{R}^{d_z}$ is the encoder and $B_i : \mathbb{R}^2 \to \mathbb{R}$ are learned functions. In our implementation, the $B_i$ functions are fully connected neural networks with 5 layers and 512 hidden units. We experimented with the less expressive function suggested in the original work, but found that the extra capacity improved performance across all datasets we considered.

### B.7. Unsupervised hyperparameter selection

In practice, one cannot measure MCC since the ground-truth latent variables are not observed. Unlike in standard machine learning setting, hyperparameter selection for disentanglement cannot be performed simply by evaluating goodness of fit on a validation set and selecting the highest scoring model since there is usually a trade-off between goodness of fit and disentanglement (Locatello et al., 2019, Sec. 5.4). To circumvent this problem, Duan et al. (2020) introduced *unsupervised disentanglement ranking* (UDR) which, for every hyperparameter combinations, measures how consistent are different random intializations of the algorithm. The authors argue that hyperparameters yielding disentangled representation typically yields consistent representations. In our experiments, the consistency of a given hyperparameter combination is measured as follows: for every pair of models, we compute the MCC between their representations. Then, we report the median of all pairwise MCC. This gives a UDR score for every hyperparameter values considered. Fig. 12 report the ELBO (normalized between zero and one), the MCC and the UDR score for the experiments of Fig. 3. We can visualize the trade-off between ELBO and MCC. However, MCC and UDR correlates nicely except for the non-diagonal time-sparsity dataset, where, for larger regularization values, UDR indicates highly consistent representations despite the bad MCC. We noticed that these

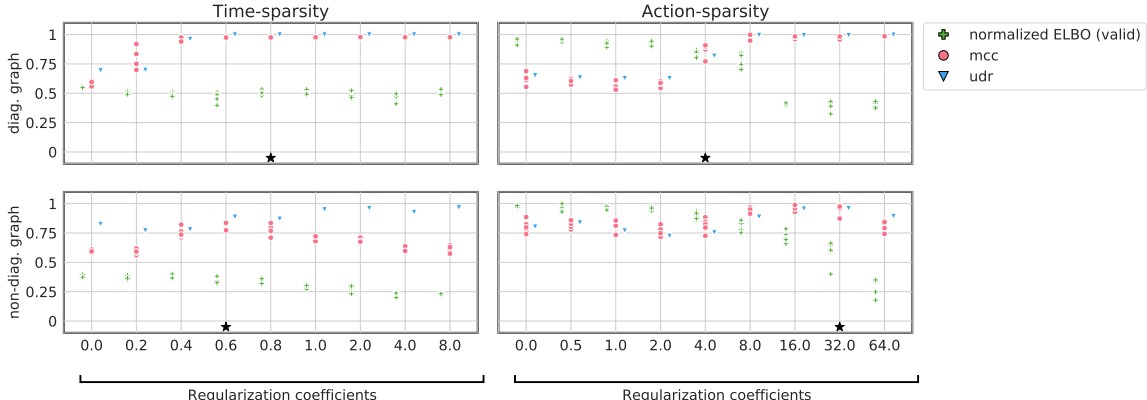

Figure 12: Investigating the link between goodness of fit (ELBO), disentanglement (MCC) and UDR. The ELBO is normalized so that it remains between 0 and 1.

specific runs correspond to excessively sparse graph, with fewer than 10 edges (out of 100 possible edges). The black star indicates the hyperparameter selected by UDR when excluding coefficient values which yields graphs with less than 10 edges (on average). This makes sense since the graphical criterion cannot be satisfied in these cases.

**Baselines.** Two of the baselines considered had hyperparameters to tune, SlowVAE (Klindt et al., 2021) and TCVAE (Chen et al., 2018). For SlowVAE, we did a grid search on the following values, $\gamma \in \{1.0, 2.0, 4.0, 8.0, 16.0\}$ and $\alpha \in \{1, 3, 6, 10\}$. For TCVAE, we explored $\beta \in \{1, 2, 3, 4, 5\}$ but the optimal value in terms of disentanglement was almost always 1. Values of $\beta$ larger than 5 led to instabilities during training. The hyperparameters were selected using UDR, as described in the paragraph above.

## Appendix C. Author contributions

**Sébastien Lachapelle** developed the idea, the theory and proofs behind mechanism sparsity regularization for disentanglement, wrote the first draft of the paper, and designed and implemented the regularized VAE-based method. **Pau Rodríguez López** ran all experiments appearing in the paper, produced associated figures and ran experiments with image data that are still work in progress. **Yash Sharma** contributed to the research process, the experimental design in particular, implemented and started running experiments on image data that are still work in progress, and contributed to the writing and the literature review. **Katie Everett** implemented and started running experiments on image data that are still work in progress and contributed to the writing and figures. **Rémi Le Priol** reviewed the proofs of main theorems, simplified some arguments and the overall proof presentation and contributed to the writing and figures. **Alexandre Lacoste** produced image datasets that are still being investigated and provided supervision. **Simon Lacoste-Julien** helped with overall paper presentation, clarified the conceptual framework and the motivation and provided supervision.

