# OpenReview forum: "Disentanglement via Mechanism Sparsity Regularization: A New Principle for Nonlinear ICA"
_cclear.cc/CLeaR/2022/Conference — CLeaR 2022 Poster_

### Official Review · Reviewer_vHyw · 2021-11-08

**Confidence:** 4
**Overall Score:** 7

**Main Review:**

The paper is very well written and easy to follow, despite several more complex statements. The initial research statement is well motivated, and the examples given throughout the paper makes it accessible, on a higher level, to a wider audience (i.e. what the paper is about in general, and what it tries to solve). The approach is novel and concerns an important problem. The idea of using the sparsity of mechanisms to find the true mechanisms seems fair, since it should intuitively hold in many real-world situations. The assumptions of the method, such as that the conditional density belongs to an exponential family, the Jacobian varying sufficiently etc., might limit it in practical applications and can make it hard to verify whether they hold in a real-world dataset or not, but are necessary considering the goal of provably finding the mechanisms. Overall, I would consider the paper as a good contribution to the ICA and causality research field.

The only major drawback of this paper is the limited experimental evaluation. The proposed method is evaluated on two synthetic datasets, one with a time-sparsity setting, and another for action-sparsity. While the proposed method outperforms the shown baselines, it is not clear how robust the method is, and whether it would work on more realistic data. For instance, the noise added to the observation $X$ was chosen to have a small standard deviation of 0.01. What if this noise is increased? How does the method handle higher noise levels? Similarly, what if the noise level is increased for the ground truth transition function (covariance is chosen to be $0.0001I$). In terms of the adjacency matrices, more diverse graphs with random connections can be tested.

#### Minor comments
* Page 5, last paragraph: typo 'which we which we'
* Appendix page 43, 4th line in B.1: double punctuation 'with negative slope of 0.2. .'
* Page 7, paragraph _Sparsity_: It is stated that the minimal graph is found that 'properly fit[s] the data'. What does 'properly' mean in this case? Is it that the data is fit up to a small error $\epsilon$ which is supposed to be lower than the regularization?
* Page 45, paragraph _Learned mechanism_: The MLP for the mean layer is said to use 5 layers with each 512 hidden units. This seems quite large compared to the ground truth functions (f being a 3 layer MLP with 20 hidden units, mean functions often sin/cos-based). Is such a large network necessary to learn the sin/cos functions in the synthetic datasets, and get stable results?
* Page 12, Figure 3: The chosen plot style is not very intuitive to understand at first. The y-axis represents a 0-1 scaling of three metrics which have different trends (SHD - lower is better, R/MCC - higher is better). The label 'diag. graph' and 'non-diag. graph' on this axis add to the confusion since it is a label for the row, not the y-axis itself. It also wasn't clear that the multiple points/crosses represent multiple runs, because several overlay each other. I would recommend splitting this plot into multiple ones, which makes it much easier to grasp for the reader.

**Summary:**

This paper proposes a method for finding the true, latent factors and their causal relations by enforcing a sparsity among mechanisms. The method is build upon works around ICA and includes proofs under which settings it can correctly disentangle the factors. The paper also discusses a VAE-based approach to implement the proposed method.

---

> ### Author Response · Authors · 2021-12-04
> **Response**
>
> We would like to thank the reviewer for the work they have put into reviewing this work.
>
> **Limited experimental evaluation:** We would like to clarify that we tested our approach on more than two datasets, more precisely 2 * 4 = 8 datasets. For both time and action-sparsity, we have a dataset with diagonal graph, non-diagonal graph, a graph violating the “graph criterion” of Thm. 5 and a dataset violating sufficient variability. The experiments on datasets violating assumptions can be found in Appendix B.3. That being said, we agree with the reviewer that more experiments would further confirm the viability of the proposed approach. Based on the reviewer’s suggestion, we performed a few experiments to verify how robust our approach is to (i) higher observation noise levels, (ii) higher latent noise levels, and (iii) more diverse (random) graphs. We will add these experiments to the camera-ready version.
>
> **(i)** Higher observation noise levels: https://anonymous.4open.science/r/anon_disentanglement_via_mechanism_sparsity-C43C/additional_figure/plot_obs_noise.pdf
>
> We considered the same data generating process with the same non-diagonal graphs as in Fig. 3 on p.12, but varying the noise level on the observation. We consider a standard deviation of 0.01, 0.1 and 0.5. We can see that the time-sparsity experiment suffers from higher noise level but not the action-sparsity dataset. We emphasize that the theory developed in this work applies for any noise level. That being said, the gap in performance we see in the time-sparsity experiment might be explained by a worse sample complexity due to noisier data. It is also possible that our simple choice of approximate posterior $q(z^{\leq T} \mid x^{\leq T}, a^{< T})$ is a worse approximation when the noise is greater. Investigating these questions is left as future work.
>
> **(ii)** Higher latent noise levels: https://anonymous.4open.science/r/anon_disentanglement_via_mechanism_sparsity-C43C/additional_figure/plot_latent_noise.pdf
>
> As above, we consider the same data generating process with the same non-diagonal graphs as in Fig. 3 on p.12, but, this time, varying the noise level on the latent variables. We consider standard deviations of 0.01, 0.1 and 0.5. Again, our theory applies for any noise level on the latents. The approach performs well everywhere except on the time-sparsity dataset for the maximal standard deviation of 0.5 (the method yields good performance with std = 0.01 and 0.1). Given that the initial latent is sampled from a Nomal(0,I), we consider a std of 0.5 to be very high.
>
> **(iii)** Diverse random graphs: https://anonymous.4open.science/r/anon_disentanglement_via_mechanism_sparsity-C43C/additional_figure/plot_random_graphs.pdf
>
> In this experiment, we used the same sin/cos data generating process, but we sampled the graphs randomly using independent Bernoulli variables with probability of sampling an edge of 0.25, 0.5 and 0.75. In this case, we do not know if the sampled graph satisfies the graphical criterion on Thm. 5. What we observe is that we obtain very good MCC (sometimes better than the experiments of Fig. 3) with probability 0.25 and 0.5, and less so with 0.75. We believe this is because a denser graph is less likely to satisfy the graphical criterion.
>
> **Page 7, paragraph “Sparsity”:** This statement was intended to be informal. We adjusted it to make it clearer and more rigorous: “This assumption is satisfied if $\hat{G}$ is a minimal graph among all graphs that allow the model to exactly match the ground-truth generative distribution.”
>
> **Is such a large network necessary to learn the sin/cos functions in the synthetic datasets, and get stable results?** Clarification: it is true that the _ground truth_ decoder function has only 3 layers of 20 units. However, the learned decoder has 6 layers of 512 units. Preliminary experiments showed that overparameterizing both the transition network and the decoder achieved better performance. This is in line with recent observations in the deep learning community that overparameterization is beneficial for optimization, see for example [1].
>
> [1] Sanjeev Arora, Nadav Cohen, Elad Hazan. On the Optimization of Deep Networks: Implicit Acceleration by Overparameterization. ICML (2018).

---

> > ### Comment · Reviewer_vHyw · 2021-12-20
> > **Response to rebuttal**
> >
> > Thank you for your response, it clarifies all my questions.

---

### Official Review · Reviewer_fiBq · 2021-11-17

**Confidence:** 2
**Overall Score:** 7

**Main Review:**

This paper proposes to achieve disentanglement of latent factors by simultaneously learning them with the sparse causal graph that relates them. Building on recent results in the nonlinear ICA literature, permutation-identifiability is proven under a number of assumptions. The theory makes use of the intuition that entangled representations will require a denser adjancency matrix and that inducing sparsity on this matrix will push the latent factors to disentangle.


## Strengths

The paper is written and structured clearly and easy to follow. It provides examples throughout to illustrate the different concepts, which further helps understanding. Additionally, the theory presented is well-motivated throughout.

While the experiments are limited to (rather unrealistic) small-scale simulated data, they show the improved disentanglement performance under the proposed approach, as well as the effect of violated assumptions consolidating the presented theory.


## Weaknesses

While most assumptions are well motivated, for some this motivation is missing and I would be curious to know how the authors expect these assumptions to restrict the applicability of their method to different scenarios. For example:
1) What does it mean in practice that $\mathbf{f}$ is assumed to be a diffeomorphism?
2) What does assumption 1 in theorem 5 mean in practice?

Additionally, some symbols are not well-defined. Most importantly, $G$ is described to be causal, but no explanation is given for why that would be the case. Additionally, I find the usage of $\mathbf{T}$ slightly confusing. While it generally describes the sufficient statistic, on page 5 it is also described as an element-wise nonlinearity. On the next line, an example for the Gaussian case is given, where $\mathbf{T}(z) := z$, which is not nonlinear. Other symbols for which I would appreciate a clearer definition or separate description are $y$ and $Py$ (Definition 2), $p$ (starting in Theorem 4) and $\tau$ (section 2.4.1).


### Disclaimer
I did not check the proofs provided in the Appendix in detail.

**Summary:**

Interesting paper achieving disentanglement by sparsity regularisation

---

> ### Author Response · Authors · 2021-12-04
> **Response**
>
> We thank the reviewer for the work they have put into reviewing our work.
>
> **Practical implications that f must be a diffeomorphism?** Thank you for this very important and interesting question. We answer it with an example that violates this assumption. Choose any vision task presenting occlusion, i.e. when an object can be completely hidden behind another one. Continuing with the tree-robot-ball example, occlusion could happen for instance when the ball is hidden behind either the tree or the robot. This situation violates the invertibility of f since given the pixel representation of the situation, we cannot infer the position of the ball, since it can either be behind the tree or behind the robot. Relaxing this invertibility assumption is a very interesting question left for future work. Diffeomorphism also means $f$ and its inverse must be differentiable, however we believe this assumption is purely technical and could probably be relaxed. Our experiments were performed with a piecewise linear $f$ (because of the ReLU activations) which are not differentiable on a set of (Lebesgue) measure 0, but this was not an issue in practice. We will happily add a discussion on these points to our manuscript.
>
> **What does assumption 1 in theorem 5 mean in practice?** This assumption says two things: First, that $k = 1$ (dimensionality of the sufficient statistic T) and, second, that T is invertible (and differentiable, but this is a purely technical assumption that could probably be relaxed). For instance if both the mean and the variance of $p(z^t \mid z^{<t}, a^{<t})$ in the ground-truth model are assumed to vary a lot with $z^{<t}$ and $a^{<t}$, then Thm. 5 does not provide any guarantee since this situation is better modelled when $k=2$. Another counterexample, which we gave in the paper, is if the mean remains fixed and only the variance changes, i.e. $T(z) = z \odot z$, which is not invertible.
>
> To verify how important the $k=1$ assumption is, we decided to apply our method on a dataset with $k=2$. The data generating process is very similar to the one considered in Fig. 3 with a non-diagonal adjacency matrix, but here the variance of $z^t$ depends on $z^{t-1}$ or $a^{t-1}$. Note that this means the model is misspecified (since we do not model this dependency). The results of these experiments are presented here: https://anonymous.4open.science/r/anon_disentanglement_via_mechanism_sparsity-C43C/additional_figure/plot_k=2.pdf.
>
> This experiment indicates that Assumption 1 of Thm. 5 might not be crucial in practice and that our approach is robust to model misspecification to some extent. We will add these experiments to the manuscript.
>
> **Why is G causal?** Thank you for asking for clarifications. Strictly speaking, our work does not require $G$ to be causal, in the sense that it can be used to predict the effect of interventions targeting the latent variables. But when the index $t$ refers to time, which is what we assume throughout, interpreting $G$ as causal makes a lot of sense, since the future cannot affect the past. The argument is analogous to why Granger causality is actual causality under some assumptions (See Thm. 10.3 in [1] for instance). We will add a quick clarification in Section 2.
>
> **Confusion around T being a sufficient statistic and a nonlinearity.** We clarified the problematic sentence just below Definition 1 by changing “element-wise nonlinearity” to “element-wise sufficient statistic”, since, indeed, T will not necessarily be nonlinear.
>
> [1] Jonas Peters, Dominik Janzing, and Bernhard Scholkopf. Elements of Causal Inference. MIT Press (2017).

---

> > ### Comment · Reviewer_fiBq · 2021-12-14
> > **Reply**
> >
> > Thank you for the clarifications and additional details. I would be happy to see this paper accepted.

---

### Official Review · Reviewer_h9au · 2021-11-23

**Confidence:** 3
**Overall Score:** 7

**Main Review:**

# Review
The authors present a novel approach to discovering disentangled latent variables from observations and auxiliary data generated by a latent variable model. They present identifiability results for their approach (up to permutation), and present empirical comparisons with related work on synthetic data. Both the motivation and the presentation is very clear and easy to follow. The comparison with previous literature is sufficient (at least at the conceptual level, see below). The findings are novel and of interest to the conference audience, and overall topic of the paper is very timely.

## Questions and suggestions

Below I present some questions and suggestions to the authors:

- Possibly the only obvious shortcoming of the paper is comparison on real data with similar works, especially with iVAE. Do the authors plan to add such results?
- It would be a good idea to mention the dependency of the method on the observed auxiliary variable (e.g. time) is a necessary condition for the application of this approach in the abstract of the paper.
- Given that sparsity is a concept that can be understood differently according to the context (e.g. in compressed sensing vs. scientific computation), I think it would be a good idea to delineate what sparse means in this context early on in the paper.
- Can the regularization on G_a and G_z undermine sufficient (time or action) variability if taken to large levels?

## Minor points
- In Thm 4, the explanation for the first assumption is given below at least partly. So in the theorem you could refer the author to the lines below, and from there to Def. 9.
- "This can be interpreted as having a sufficiently nonlinear transition model." Pg. 6. Can any linear transition model meet this criterion?
- For an even more lively exposition (e.g. in Pg. 10), demonstrating how differently results obtained by iVAE would be in a similar example (e.g. that in Fig. 1) could be beneficial.

**Summary:**

The authors present a novel approach for discovering disentangled representations in data sets with auxiliary variables.

---

> ### Author Response · Authors · 2021-12-04
> **Response**
>
> We thank the reviewer again for the work they have put into reviewing our work.
>
> **Experiments on real data:** We agree that experiments demonstrating the usefulness of mechanism sparsity for disentanglement on realistic data would be very interesting. We believe the rigorous theory, the algorithm contribution, the careful writing to make non-trivial ideas approachable, and our existing experiments already constitute a novel and significant contribution that will be of interest to the community, and that publishing the paper and code as it stands now allows the research community to build on these ideas. We note that the paper is already 51 pages, half of which are careful proofs of our theory in the appendix. We are very excited to explore more realistic applications of mechanism sparsity, but leave it as future work.
>
> **Mentioning dependency on auxiliary variables in abstract:** We agree this point should be more explicit in the abstract and will adjust it in camera-ready version (see proposed abstract modification below). We would like to make a clarification. Whether our method requires auxiliary variables depends on how you define “auxiliary variables”. If you consider “previous observations” as an auxiliary variable (like in [1]), then yes our method requires it. If you only consider the random variable $A^t$ to be the auxiliary variable, then, no, our method does not require observing some $A^t$ (previous observations are sufficient).
>
> We have changed the first sentence of the abstract for:
> “This work introduces a novel principle we call disentanglement via mechanism sparsity regularization, which can be applied when the latent factors of interest depend sparsely on past latent factors and/or observed auxiliary variables.”
>
> **Effect of excessive regularization on sufficient variability:** Thank you for this question. It is important to understand that the sufficient variability assumptions are assumptions about the ground-truth data generative process, and not the learned model itself. Thus, regularization will never affect these assumptions. That being said, excessive regularization can indeed be problematic by hindering the ability of the learned model to fit the data properly, which would prevent you from applying Thm. 5, which assumes the learned distribution is exactly equal to the ground-truth distribution. We will add a quick discussion on this phenomenon in the experimental section.
>
> **Can any linear transition model satisfy the sufficient variability assumption of Thm. 4 (linear identifiability)?** Great question. It turns out interpreting this assumption as having a sufficiently nonlinear transition model was our mistake since there are linear models that satisfy this assumption. For example, $\lambda(z^{<t}, a^{< t}) := Wz^{t-1}$ with $W$ an invertible matrix would satisfy the sufficient variability assumption of Thm. 4. We will fix this in the text.
>
> [1] Aapo Hyvarinen, Hiroaki Sasaki, Richard E. Turner. Nonlinear ICA Using Auxiliary Variables and Generalized Contrastive Learning. AISTATS (2019).

---

> > ### Comment · Reviewer_h9au · 2021-12-17
> > **Thanks for the response**
> >
> > I thank the authors for their further clarifications, and welcome the changes they plan to make to improve the exposition and presentation. I am looking forward to reading the published version of their paper.

---

### Author Response · Authors · 2021-12-04
**General Response**

We thank the reviewers for their valuable feedback. *Reviewer fiBq* concisely conveyed the intuition of our contribution: “The theory makes use of the intuition that entangled representations will require a denser adjacency matrix and that inducing sparsity on this matrix will push the latent factors to disentangle.” All three reviewers found the paper well-motivated, well-written and easy to follow. For example, *reviewer vHyw* writes “The paper is very well written and easy to follow, despite several more complex statements. The initial research statement is well motivated, and the examples given throughout the paper makes it accessible, on a higher level, to a wider audience (i.e. what the paper is about in general, and what it tries to solve).” In addition, all three reviewers found the paper of interest. For example, *reviewer h9au* mentioned “The findings are novel and of interest to the conference audience, and overall topic of the paper is very timely.”

We will make our code publicly available. The reviewers can find an anonymized version here: https://anonymous.4open.science/r/anon_disentanglement_via_mechanism_sparsity-C43C/

We will address the concerns of each reviewer individually. Note that some answers provide novel experiments.

---

### Decision · Program_Chairs · 2022-01-12

**Decision:**

Accept (Poster)

**Comment:**

The reviewers unanimously acknowledge the clarity of the paper and the relevance of the theoretical contribution. One weakness pointed out by reviewers is the limited experimental evaluation. Overall, this constitutes a good theoretical contribution at the intersection of causal representation learning and nonlinear ICA, which we recommend for acceptance.